# PROBABILISTIC MODELING THE HIDDEN LAYERS OF DEEP NEURAL NETWORKS

## ABSTRACT

In this paper, we demonstrate that the parameters of Deep Neural Networks (DNNs) do not satisfy the i.i.d. prior assumption and activations being i.i.d. is not valid for all the hidden layers of DNNs, thus the Gaussian process cannot correctly model the distribution of the hidden layers. Alternatively, we propose a novel probabilistic representation for the hidden layers of DNNs in two aspects: (i) a hidden layer formulates a Gibbs distribution, of which the energy function is determined by the activations, and (ii) the statistical connection between two adjacent layers can be formulated as a product of experts model. Based on the proposed probabilistic representation, we further confirm the equivalence between the variational inference and the stochastic gradient descent, and prove the later is also equivalent to the energy minimization optimization. As a result, the entire architecture of DNNs can be viewed as a Bayesian network, in which the hidden layers close to the input formulate the prior distributions of the training dataset. In addition, we propose a new regularization approach to improve the generalization performance of DNNs by pre-training the hidden layers corresponding to the prior distributions. Simulation results validate the proposed theories.

## 1 INTRODUCTION

Recently, interpreting the hidden layers of Deep Neural Networks (DNNs) as the Gaussian Process (GP) has attracted a great deal of attention because it provides a novel probabilistic perspective to clarify the working mechanism of deep learning. Neal (1994) initially demonstrates the equivalence between GP and neural networks with a single fully connected layer in the limit of infinite neurons. Following this seminal work, Lee et al. (2018); Matthews et al. (2018) further extend the equivalence to neural networks with multiple hidden layers and Garriga-Alonso et al. (2018); Novak et al. (2018) establish the correspondence between the convolutional layers and GP.

It is important to note that the GP explanations rely on a fundamental probabilistic premise that the activations of a hidden layer are independent and identically distributed (i.i.d.). More specifically, based on the classical Central Limit Theorem (CLT), the prerequisite of an activation of a hidden layer approaching a Gaussian distribution is that all the activations of the previous layer are i.i.d.. Though the Lyapunov CLT could relax the probabilistic premise to be independence only, all the previous works do not thoroughly discuss the probabilistic premise except assuming an i.i.d. prior for all the parameters of DNNs. Due to its extreme importance but unclear status, it is necessary to thoroughly investigate the probabilistic premise for improving the interpretability of DNNs.

In this work, we demonstrate that the parameters of DNNs are correlated and depend on the training dataset, i.e., they do not satisfy the i.i.d. prior. Moreover, we demonstrate that activations being i.i.d. cannot hold for all the hidden layers of DNNs. In the context of Bayesian probability, we theoretically derive the necessary conditions for the activations of a hidden layer being i.i.d. given the assumption that the activations of the previous layer are i.i.d.. Subsequently, we experimentally show that typical DNNs, such as the Multilayer Perceptron (MLP) and the Convolutional Neural Networks (CNNs), cannot satisfy the necessary conditions. As a result, activations being i.i.d. is not valid for all hidden layers. In other words, GP with the i.i.d. assumption for the parameters of DNNs cannot correctly explain all the hidden layers of DNNs. In addition, some previous works show that GP is sensitive to the curse of dimensionality (Bengio et al., 2005; Hinton et al., 2012) and cannot clarify the hierarchical representation, an essential of deep learning (Matthews et al., 2018).

Alternatively, we propose a novel probabilistic representation for the hidden layers based on the Gibbs distribution (LeCun et al., 2006) and the Product of Experts (PoE) model (Hinton, 2002). The probabilistic representation explains a hidden layer in two aspects: (i) the distribution of a hidden layer can be formulated as a Gibbs distribution, of which the energy function is determined by the activations, and (ii) the connection between two adjacent layers can be modeled by a PoE model. Specifically, the distribution of a single neuron or hidden unit (e.g., convolutional channel) can be expressed as a PoE, in which all the experts are defined by the neurons or hidden units in the previous hidden layer. Compared to the GP explanations, the proposed probabilistic representation provides a more specific explanation for the hidden layers of DNNs without any assumption.

Since the output of a hidden layer is commonly the input of the next layer, DNNs form a Markov chain, thereby corresponding to a joint distribution. Moreover, we confirm the equivalence between the Stochastic Gradient Descent (SGD) and the variational inference (Mandt et al., 2017; Chaudhari & Soatto, 2018) and prove the former is also equivalent to the energy minimization optimization based on the proposed probabilistic representation. As a result, the entire architecture of DNNs can be explained as a Bayesian network (Nielsen & Jensen, 2009; Koski & Noble, 2011). In particular, we demonstrate that the hidden layers close to the input formulate prior distributions of the training dataset and propose a novel regularization approach to improve the generalization performance of DNNs through pre-training the hidden layers corresponding to prior distributions.

## 2 Preliminaries

We assume $\boldsymbol{X}$ and $\boldsymbol{Y}$ are two random variables and $P_{\boldsymbol{\theta}}(\boldsymbol{X}, \boldsymbol{Y}) = P(\boldsymbol{Y}|\boldsymbol{X})P(\boldsymbol{X})$ is an unknown joint distribution between $\boldsymbol{X}$ and $\boldsymbol{Y}$, where $P(\boldsymbol{X})$ describes the prior knowledge of $\boldsymbol{X}$, $P(\boldsymbol{Y}|\boldsymbol{X})$ describes the statistical connection between $\boldsymbol{X}$ and $\boldsymbol{Y}$, and $\boldsymbol{\theta}$ indicate the parameters of $P_{\boldsymbol{\theta}}(\boldsymbol{X}, \boldsymbol{Y})$.

A dataset $\boldsymbol{\mathcal{D}} = \{(\boldsymbol{x}^j, \boldsymbol{y}^j)|\boldsymbol{x}^j \in \mathbb{R}^M, \boldsymbol{y}^j \in \mathbb{R}^L\}_{j=1}^J$ is composed of i.i.d. samples generated from $P_{\boldsymbol{\theta}}(\boldsymbol{X}, \boldsymbol{Y})$. A neural network with $I$ hidden layers is denoted as DNN $= \{\boldsymbol{x}; \boldsymbol{f_1}; ...; \boldsymbol{f_I}; \boldsymbol{f_Y}\}$ and trained by $\boldsymbol{\mathcal{D}}$, where $(\boldsymbol{x}, \boldsymbol{y}) \in \boldsymbol{\mathcal{D}}$ are the input of the DNN and the corresponding training label, and $\boldsymbol{f_Y}$ is an estimation of the true distribution $P(\boldsymbol{Y}|\boldsymbol{X})$. As a result, $\boldsymbol{x} \sim P(\boldsymbol{X})$ and $\boldsymbol{f_Y} \approx P(\boldsymbol{Y}|\boldsymbol{X})$. The meaning of $\boldsymbol{f_i}$ is two-fold: (i) denoting all the neurons in the $i$th hidden layer, and (ii) representing the activations of the $i$th hidden layer. In addition, $\boldsymbol{F_i}$ and $\boldsymbol{F_Y}$ are the random variables corresponding to $\boldsymbol{f_i}$ and $\boldsymbol{f_Y}$, respectively. In this work, we use the MLP (Figure 1) to derive most theoretical conclusions unless otherwise specified.

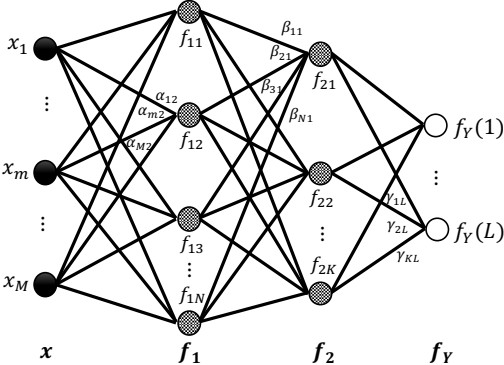

Figure 1: A MLP $= \{\boldsymbol{x}; \boldsymbol{f_1}; \boldsymbol{f_2}; \boldsymbol{f_Y}\}$, in which $\boldsymbol{f_1}$ has $N$ neurons, i.e., $\boldsymbol{f_1} = \{f_{1n} = \sigma_1[g_{1n}(\boldsymbol{x})]\}_{n=1}^N$, where $g_{1n}(\boldsymbol{x}) = \sum_{m=1}^M \alpha_{mn} \cdot x_m + b_{1n}$ is the $n$th linear filter, $\alpha_{mn}$ is the weight of the edge between $x_m$ and $f_{1n}$, $b_{1n}$ is the bias, and $\sigma_1$ is a non-linear activation function. Similarly, $\boldsymbol{f_2} = \{\sigma_2[g_{2k}(\boldsymbol{f_1})]\}_{k=1}^K$ where $g_{2k}(\boldsymbol{f_1}) = \sum_{n=1}^N \beta_{nk} \cdot f_{1n} + b_{2k}$. In addition, since $\boldsymbol{f_Y}$ is the softmax, $\boldsymbol{f_Y} = \{\frac{1}{\boldsymbol{Z_Y}}\exp[g_{yl}(\boldsymbol{f_2})]\}_{l=1}^L$ where $g_{yl}(\boldsymbol{f_2}) = \sum_{k=1}^K \gamma_{kl} \cdot f_{2k} + b_{yl}$, and the partition function is $\boldsymbol{Z_Y} = \sum_{l=1}^L \exp[g_{yl}(\boldsymbol{f_2})]$.

## 3 BACKGROUND

As a fundamental probabilistic model, Gibbs distribution (a.k.a., energy based model, Boltzmann distribution, or renormalization group) formulates the dependence of random variables $\boldsymbol{X}$ through associating an energy to each dependence structure (Geman & Geman, 1984).

$$P(\boldsymbol{X}; \boldsymbol{\theta}) = \frac{1}{Z(\boldsymbol{\theta})} \exp[-E(\boldsymbol{x}; \boldsymbol{\theta})] \qquad (1)$$

where $E(\boldsymbol{x}; \boldsymbol{\theta})$ is the energy function, $\boldsymbol{\theta}$ denotes the parameters, and $Z(\boldsymbol{\theta}) = \int_{\boldsymbol{x}} \exp[-E(\boldsymbol{x}; \boldsymbol{\theta})]d\boldsymbol{x}$ is the partition function. Especially, a Gibbs distribution can be easily reformulated as various probabilistic models by redefining $E(\boldsymbol{x}; \boldsymbol{\theta})$, e.g., it would become Markov Random Fields (MRFs) model if the energy function is defined as the summation of multiple potential functions, i.e., $E(\boldsymbol{x}; \boldsymbol{\theta}) = -\sum_k f_k(\boldsymbol{x}; \boldsymbol{\theta}_k)$ (Wang et al., 2013; Goodfellow et al., 2016). Moreover, assuming a single potential function defines an expert, i.e., $\boldsymbol{F}_k = \exp[-f_k(\boldsymbol{x}; \boldsymbol{\theta}_k)]$, we can regard MRFs as a PoE model, i.e., $P(\boldsymbol{x}; \boldsymbol{\theta}) = \frac{1}{Z(\boldsymbol{\theta})} \prod_k \boldsymbol{F}_k$, where $Z(\boldsymbol{\theta}) = \prod_k Z(\boldsymbol{\theta}_k)$ (Hinton, 2002).

A Gibbs distribution has two appealing properties. First, $E(\boldsymbol{x}; \boldsymbol{\theta})$ is the sufficient statistics of $P(\boldsymbol{x}; \boldsymbol{\theta})$ because $Z(\boldsymbol{\theta})$ only relies on $\boldsymbol{\theta}$, i.e., $P(\boldsymbol{x}; \boldsymbol{\theta})$ entirely depends on $E(\boldsymbol{x}; \boldsymbol{\theta})$. The property establishes a connection between $P(\boldsymbol{X}; \boldsymbol{\theta})$ and the deterministic function $E(\boldsymbol{x}; \boldsymbol{\theta})$, which allow us to explain a deterministic hidden layer in a probabilistic way. Second, the energy minimization is the commonly used optimization for learning $\boldsymbol{\theta}$, i.e., $\boldsymbol{\theta}^* = \arg\min_{\boldsymbol{\theta}} E(\boldsymbol{x}; \boldsymbol{\theta})$ (LeCun et al., 2006), which is helpful to explain the training procedure of DNNs, because the energy minimization can be implemented by the gradient descent algorithm as long as $E(\boldsymbol{x}; \boldsymbol{\theta})$ is differentiable.

Numerous efforts have been devoted to explain DNNs from the viewpoint of Gibbs distribution. The classical work is the Restricted Boltzmann Machine (RBM) (Salakhutdinov & Hinton, 2009), which is a simplified Gibbs distribution. Specifically, RBM only considers the connections between visible units $\boldsymbol{x}$ and hidden units $\boldsymbol{h}$ but overlooks the internal connections within $\boldsymbol{x}$ and $\boldsymbol{h}$, thus it cannot explain complex DNNs, e.g., CNNs. To the best of our knowledge, Mehta & Schwab (2014) first explain the distribution of hidden layers as a Gibbs distribution, and Yaida (2019) indirectly proves the distribution of fully connected layers is a Gibbs distribution. Lin et al. (2017) clarify some properties of DNNs based on the Gibbs distribution. However, all previous works still cannot derive an explicitly Gibbs explanation for complex hidden layers, e.g., convolutional layers.

## 4 MAIN RESULTS

### 4.1 THE PARAMETERS OF DNNS DO NOT SATISFY THE I.I.D. ASSUMPTION

We examine the statistical properties of the parameters of DNNs and show they are correlated. First, we derive a necessary condition for the parameters of DNNs satisfying the i.i.d. prior, Second, we experimentally show that typical DNNs cannot satisfy the necessary condition on the MNIST and CIFAR-10 datasets. Hence, the parameters of DNNs do not satisfy the i.i.d. assumption.

If all the parameters of DNNs are i.i.d., the parameters of each neuron are identically distributed, e.g., $\{\alpha_{mn}\}_{m=1}^M \sim P(\boldsymbol{\alpha}_n)$ in the MLP (Figure 1). Since all the parameters are i.i.d., different neurons should be uncorrelated, i.e., $\forall n \neq n', Corr(\boldsymbol{\alpha}_n, \boldsymbol{\alpha}_{n'}) = 0$. In particular, the i.i.d. assumption enables us to use the sample correlation $r(\boldsymbol{\alpha}_n, \boldsymbol{\alpha}_{n'})$ to estimate $Corr(\boldsymbol{\alpha}_n, \boldsymbol{\alpha}_{n'})$.

$$r(\boldsymbol{\alpha}_n, \boldsymbol{\alpha}_{n'}) = \frac{\sum_{m=1}^M (\alpha_{mn} - \overline{\alpha}_n)(\alpha_{mn'} - \overline{\alpha}_{n'})}{\sqrt{\sum_{m=1}^M (\alpha_{mn} - \overline{\alpha}_n)^2 \sum_{m=1}^M (\alpha_{mn'} - \overline{\alpha}_{n'})^2}} \approx Corr(\boldsymbol{\alpha}_n, \boldsymbol{\alpha}_{n'}) \qquad (2)$$

where $\overline{\alpha}_n = \frac{1}{M} \sum_{m=1}^M \alpha_{mn}$ is the sample mean of $\{\alpha_{mn}\}_{m=1}^M$. Therefore, taking into account the estimation error, $|r(\boldsymbol{\alpha}_n, \boldsymbol{\alpha}_{n'})|$ close to zero is the necessary condition for all parameters being i.i.d..

We examine the necessary condition for all parameters being i.i.d. in various DNNs. First, we use the MLP to classify the benchmark MNIST dataset. The information about the network architecture and training methods is included in Appendix A.1. After training the MLP well, we calculate the absolute sample correlations, namely, $|r(\boldsymbol{\alpha}_n, \boldsymbol{\alpha}_{n'})|$, $|r(\boldsymbol{\beta}_k, \boldsymbol{\beta}_{k'})|$, and $|r(\boldsymbol{\gamma}_l, \boldsymbol{\gamma}_{l'})|$, and derive three absolute correlation matrices, i.e., $A'_{N \times N}$, $B'_{K \times K}$, and $C'_{L \times L}$, shown in Figure 2. We find that most $|r(\boldsymbol{\alpha}_n, \boldsymbol{\alpha}_{n'})|$ are close to zero, but most $|r(\boldsymbol{\beta}_k, \boldsymbol{\beta}_{k'})|$ and $|r(\boldsymbol{\gamma}_l, \boldsymbol{\gamma}_{l'})|$ are non-zero. That indicates the necessary condition cannot hold for all the parameters of the MLP.

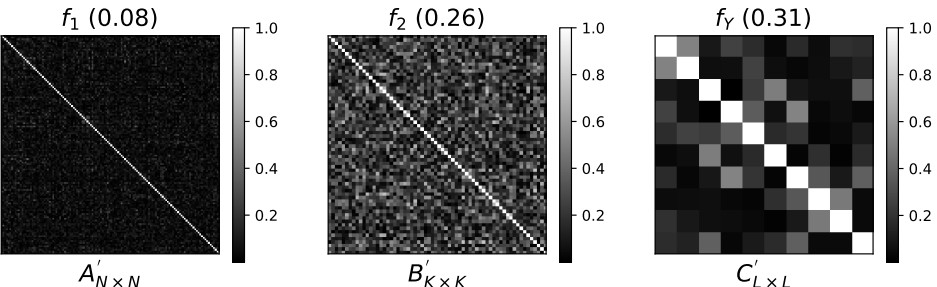

Figure 2: Three absolute correlation matrices for the weights in $\boldsymbol{f_1}$, $\boldsymbol{f_2}$, and $\boldsymbol{f_Y}$. In addition, 0.08, 0.26, and 0.31 are the average of the absolute correlation coefficients for all the weights in $\boldsymbol{f_1}$, $\boldsymbol{f_2}$, and $\boldsymbol{f_Y}$, respectively.

Second, we construct a shallow neural network with a single fully connected hidden layer, namely, $\text{NN} = \{\boldsymbol{x}; \boldsymbol{f}; \boldsymbol{f_Y}\}$, for classifying the MNIST dataset based on the seminal work (Neal, 1994) in Appendix A.2, and construct a CNN for classifying the CIFAR-10 dataset in Appendix A.3. Using the same method in the MLP, we obtain the absolute sample correlations between the weights of different nodes in the both neural networks. The results in Appendix A.2 and Appendix A.3 show that the weights of DNNs do not satisfy the necessary condition.

Third, we visualize all the parameters of the $\text{NN} = \{\boldsymbol{x}; \boldsymbol{f}; \boldsymbol{f_Y}\}$ for classifying a synthetic dataset in Appendix A.4. Since the synthetic dataset is much simpler than MNIST, we can directly show the correlation of different parameters and their dependences on the dataset by visualizing them. Overall, these simulations demonstrate that the parameters of DNNs are not i.i.d..

## 4.2 ACTIVATIONS BEING I.I.D. IS NOT VALID FOR ALL THE HIDDEN LAYERS OF DNNS

Although we demonstrate that the parameters of DNNs do not satisfy the i.i.d. assumption, we are still unclear whether the activations of a hidden layer are i.i.d.. In this section, we demonstrate that activations being i.i.d. is not valid for all the hidden layers of DNNs.

In the context of Bayesian probability, all the weights and biases of the MLP (Figure 1) have prior distributions, i.e., $\alpha_{mn} \sim P(A_{mn})$, $\beta_{nk} \sim P(W_{nk})$, $b_{1n} \sim P(B_{1n})$, and $b_{2k} \sim P(B_{2k})$, thus we regard $G_{1n} = \sum_{m=1}^{M} A_{mn} X_m + B_{1n}$ as the random variable of $g_{1n}(\boldsymbol{x}) = \sum_{m=1}^{M} \alpha_{mn} \cdot x_m + b_{1n}$, and $G_{2k} = \sum_{n=1}^{N} W_{nk} F_{1n} + B_{2k}$ as the random variable of $g_{2k}(\boldsymbol{f_1}) = \sum_{n=1}^{N} \beta_{nk} \cdot f_{1n} + b_{2k}$. Hence, the random variables of two arbitrary activations in $\boldsymbol{f_1}$ and $\boldsymbol{f_2}$, e.g., $f_{1n}$ and $f_{2k}$, can be expressed as $F_{1n} = \sigma_1(G_{1n})$ and $F_{2k} = \sigma_2(G_{2k})$, respectively.

Given two random variables $A$ and $B$, the necessary conditions for them being i.i.d. are that they are uncorrelated (i.e., $Cov(A, B) = 0$) and have the same expectation (i.e., $E(A) = E(B)$). Similarly, assuming $\{F_{1n}\}_{n=1}^{N}$ are i.i.d. and $\{f_{1n}\}_{n=1}^{N} \sim P(F_1)$, we derive the following necessary conditions for the activations of the second hidden layer, i.e. ,$\{F_{2k}\}_{k=1}^{K}$, being i.i.d..

$$\forall (k, k') \in S_1 = \{(k, k') \in \mathbb{Z}^2 | k \neq k', 1 \leq k \leq K, 1 \leq k' \leq K\}, \text{ we must have}$$

$$Cov(G_{2k}, G_{2k'}) = Cov(W_k, W_{k'})E^2(F_1) + Cov(W_k, B_{2k'})E(F_1) \qquad (3)$$
$$+ Cov(W_{k'}, B_{2k})E(F_1) + Cov(B_{2k}, B_{2k'}) = 0$$

$$E(F_1)E(W_k) + E(B_{2k}) = E(F_1)E(W_{k'}) + E(B_{2k'}) \qquad (4)$$

$$\sigma_2(\cdot) \text{ is strictly increasing, differentiable, and invertible.} \qquad (5)$$

where $W_k = \sum_{n=1}^{N} W_{nk}$. Equation 3 and 4 are the necessary conditions for $\{G_{2k}\}_{k=1}^{K}$ being i.i.d.. Equation 5 constrains the activation function $\sigma_2(\cdot)$ to guarantee that if $\{G_{2k}\}_{k=1}^{K}$ are i.i.d. then $\{F_{2k}\}_{k=1}^{K}$ are i.i.d. as well. The necessary conditions are valid for arbitrary fully connected layers as long as properly changing the subscripts. The detailed derivations are included in Appendix B and Appendix C. Moreover, we prove that the necessary conditions also hold for convolutional layers based on the connection between fully connected layers and convolutional layers in Appendix E.

Table 1: The statistical measures for examining the necessary conditions

| Layer | Activation expectation | Covariance | Weight expectation |
|---|---|---|---|
| $f_1$ | $E(X)$ | $Cov(A_n, A_{n'})$ | $E(A_n)$ |
| $f_2$ | $E(F_1)$ | $Cov(W_k, W_{k'})$ | $E(W_k)$ |
| $f_Y$ | $E(F_2)$ | $Cov(C_l, C_{l'})$ | $E(C_l)$ |

$$\{x_m\}_{m=1}^M \sim P(X), \{f_{1n}\}_{n=1}^N \sim P(F_1), \{f_{2k}\}_{k=1}^K \sim P(F_2)$$
$$A_n = \textstyle\sum_{m=1}^M A_{mn}, \text{where } \alpha_{mn} \sim P(A_{mn})$$
$$W_k = \textstyle\sum_{n=1}^N W_{nk}, \text{where } \beta_{nk} \sim P(W_{nk})$$
$$C_l = \textstyle\sum_{k=1}^K C_{kl}, \text{where } \gamma_{kl} \sim P(C_{kl})$$

It is noteworthy that the i.i.d. assumption for the parameters of DNNs indeed satisfy the necessary conditions for activations being i.i.d.. Nevertheless, since the i.i.d. prior is not an appropriate prior for the parameters of DNNs, it cannot guarantee activations of a hidden layer being i.i.d..

The theoretical necessary conditions indicate an applicable approach to examine whether activations of a hidden layer being i.i.d. given the assumption that the activations of the previous layer are i.i.d.. More specifically, since $\{F_{1n}\}_{n=1}^N$ are i.i.d., we can use the sample mean $\overline{f_1}$ to estimate $E(F_1)$. Because the samples of the training dataset $\mathcal{D}$ are also assumed to be i.i.d., we can train the MLP multiple times to draw multiple independent observations of $W_{nk}$ and $B_{2k}$, thus we can use the sample covariance to estimate $Cov(W_k, W_{k'})$, $Cov(W_k, B_{2k'})$, and etc.

Based on the empirical approach, we estimate the statistical measures (Table 1) to examine whether the activations of the three layers (i.e., $f_1$, $f_2$, and $f_Y$) being i.i.d. in the MLP for classifying the MNIST dataset. Without loss of generality, we restrict all the layers of the MLP from using bias to decrease computation complexity, thus the necessary conditions for $f_2$ can be simplified as

$$\forall (k, k') \in S_1 = \{(k, k') \in \mathbb{Z}^2 | k \neq k', 1 \leq k \leq K, 1 \leq k' \leq K\}, \text{ we must have}$$
$$Cov(G_{2k}, G_{2k'}) = Cov(W_k, W_{k'}) E^2(F_1) = 0 \tag{6}$$

$$E(F_1) E(W_k) = E(F_1) E(W_{k'}) \tag{7}$$

First, Figure 3 shows the correlation matrices, i.e., $A_{N \times N}$, $W_{K \times K}$, and $C_{L \times L}$, to check if their corresponding covariances, i.e, $Cov(A_n, A_{n'})$, $Cov(W_k, W_{k'})$, and $Cov(C_k, C_{k'})$, are close to zero. It shows that many correlations are far from zero, so their corresponding covariances are not close to zero either. Second, we use the sample mean, e.g., $\overline{x}$, to estimate the expectation, e.g., $E(X)$. Taking into account the estimation error, we can regard $E(X) = 0$ based on $\overline{x} = 0.131$. However, $\overline{f_1} = 0.436$ and $\overline{f_2} = 0.498$ imply $E(F_1) \neq 0$ and $E(F_2) \neq 0$ even considering the estimation error. Therefore, the activations of the last two hidden layer, i.e., $f_2$ and $f_Y$, cannot be independent because they cannot satisfy the first necessary condition (Equation 6). In addition, we show the necessary condition for activations being identically distributed also cannot be satisfied by the MLP in Appendix F.1. Furthermore, we demonstrate that the necessary conditions for activations being i.i.d. also cannot be satisfied in the CNN for classifying the CIFAR-10 dataset in Appendix F.2

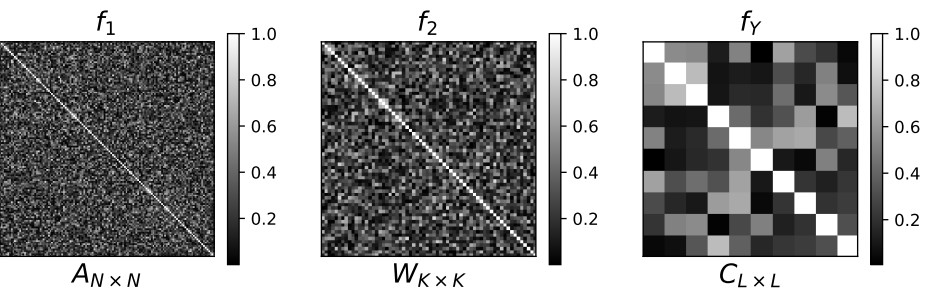

Figure 3: Three absolute correlation matrices for showing whether $Cov(A_n, A_{n'})$, $Cov(W_k, W_{k'})$, and $Cov(C_k, C_{k'})$ are close to zero.

Overall, we conclude that activations being i.i.d. is not valid for all the hidden layers of DNNs based on the theoretical necessary conditions and the experimental simulations on typical DNNs. In other words, we cannot use the CLT to establish the equivalence between GP and all the hidden layers of DNNs, thus GP can not correctly clarify all the hidden layers of DNNs. That necessitates a more general probabilistic representation for explaining the hidden layers of DNNs.

## 4.3   THE DISTRIBUTION OF A HIDDEN LAYER IS A GIBBS DISTRIBUTION

This section proves that the distribution of a hidden layer can be formulated as a Gibbs distribution and the statistical connection between two adjacent layers can be modeled as a PoE model.

Assuming the output layer in the MLP (Figure 1) is the softmax, its distribution can be expressed as

$$P(\boldsymbol{F_Y}) = \{P(f_{yl}) = \frac{1}{Z_{\boldsymbol{Y}}}\exp(f_{yl})\}_{l=1}^{L} \tag{8}$$

where $Z_{\boldsymbol{Y}} = \sum_{l=1}^{L}\exp(f_{yl})$ is the partition function. Since $f_{yl} = \sum_{k=1}^{K}\gamma_{kl}\cdot f_{2k} + b_{yl}$, we have

$$P(\boldsymbol{F_Y}) = \{P(f_{yl}) = \frac{1}{Z_{\boldsymbol{Y}}}\exp(\sum_{k=1}^{K}\gamma_{kl}\cdot f_{2k} + b_{yl})\}_{l=1}^{L} \tag{9}$$

Based on the properties of the exponential function, i.e., $\exp(a+b) = \exp(a)\cdot\exp(b)$ and $\exp(a\cdot b) = [\exp(b)]^{a}$, we can reformulate $P(\boldsymbol{F_Y})$ as

$$P(\boldsymbol{F_Y}) = \{P(f_{yl}) = \frac{1}{Z_{\boldsymbol{Y}}'}\prod_{k=1}^{K}[\exp(f_{2k})]^{\gamma_{kl}}\}_{l=1}^{L} \tag{10}$$

where $Z_{\boldsymbol{Y}}' = Z_{\boldsymbol{Y}}/\exp(b_{yl})$. Since $\{\exp(f_{2k})\}_{k=1}^{K}$ are scalar, we can introduce a new partition function $Z_{\boldsymbol{F_2}} = \sum_{k=1}^{K}\exp(f_{2k})$ such that $\{\frac{1}{Z_{\boldsymbol{F_2}}}\exp(f_{2k})\}_{k=1}^{K}$ become a probability measure. As a result, we can further reformulate $P(\boldsymbol{F_Y}) = \{P(f_{yl})\}_{l=1}^{L}$ as a Product of Expert (PoE) model as

$$P(\boldsymbol{F_Y}) = \{P(f_{yl}) = \frac{1}{Z_{\boldsymbol{Y}}''}\prod_{k=1}^{K}[\frac{1}{Z_{\boldsymbol{F_2}}}\exp(f_{2k})]^{\gamma_{kl}}\}_{l=1}^{L} \tag{11}$$

where $Z_{\boldsymbol{Y}}'' = Z_{\boldsymbol{Y}}/[\exp(b_{yl})\cdot\prod_{k=1}^{K}[Z_{\boldsymbol{F_2}}]^{\gamma_{kl}}]$ and each expert is defined as $\frac{1}{Z_{\boldsymbol{F_2}}}\exp(f_{2k})$.

It is noteworthy that all the experts $\{\frac{1}{Z_{\boldsymbol{F_2}}}\exp(f_{2k})\}_{k=1}^{K}$ form a probability measure and establish an exact one-to-one correspondence to all the neurons in the second hidden layer $\boldsymbol{f_2}$, i.e., $\{f_{2k}\}_{k=1}^{K}$. Therefore, the distribution of $\boldsymbol{f_2}$ can be expressed as

$$P(\boldsymbol{F_2}) = \{P(f_{2k}) = \frac{1}{Z_{\boldsymbol{F_2}}}\exp(f_{2k})\}_{k=1}^{K} \tag{12}$$

Based on the definition of Gibbs distribution, Equation 8 and 12 show that $\boldsymbol{f_Y}$ and $\boldsymbol{f_2}$ formulate two multivariate Gibbs distributions and their energy functions are equivalent to the negative of the output nodes $\{f_{yl}\}_{l=1}^{L}$ and the neurons $\{f_{2k}\}_{k=1}^{K}$, respectively. In addition, Equation 11 implies that the connection between $P(f_{yl})$ and $P(f_{2k})$ can be expressed as the PoE model, namely that $P(f_{yl})$ is equal to the product of all the experts $\{P(f_{2k})\}_{k=1}^{K}$ with their respective weights $\{\gamma_{kl}\}_{k=1}^{K}$.

Moreover, we prove that the probabilistic representation is valid for other hidden layers in the MLP. Since $\{f_{2k} = \sigma_2(\sum_{n=1}^{N}\beta_{nk}\cdot f_{1n} + b_{2k})\}_{k=1}^{K}$, $P(\boldsymbol{F_2})$ can be expressed as

$$P(\boldsymbol{F_2}) = \{P(f_{2k}) = \frac{1}{Z_{\boldsymbol{F_2}}}\exp[\sigma_2(\sum_{n=1}^{N}\beta_{nk}\cdot f_{1n} + b_{2k})]\}_{k=1}^{K} \tag{13}$$

Based on the equivalence between the gradient descent algorithm and the first order approximation (Battiti, 1992), we prove that $\sigma_2(\sum_{n=1}^{N}\beta_{nk}\cdot f_{1n} + b_{2k})]$ can be approximated as

$$\sigma_2(\sum_{n=1}^{N}\beta_{nk}\cdot f_{1n} + b_{2k})] \approx C_{21}\cdot[\sum_{n=1}^{N}\beta_{nk}\cdot f_{1n} + b_{2k}] + C_{22} \tag{14}$$

where $C_{21}$ and $C_{22}$ only depend on the activations $\{f_{1n}\}_{n=1}^{N}$ in the previous training iteration, thus they can be regarded as constants and absorbed by $\beta_{nk}$ and $b_{2k}$. The proof for the approximation is included in Appendix G. Therefore, $P(\boldsymbol{F_2})$ also can be modeled as a PoE model.

$$P(\boldsymbol{F_2}) = \{P(f_{2k}) \approx \frac{1}{Z''_{\boldsymbol{F_2}}} \prod_{n=1}^{N} [\frac{1}{Z_{\boldsymbol{F_1}}} \exp(f_{1n})]^{\beta_{nk}}\}_{k=1}^{K} \tag{15}$$

where $Z''_{\boldsymbol{F_2}} = Z_{\boldsymbol{F_2}} / [\exp(b_{2k}) \cdot \prod_{n=1}^{N} [Z_{\boldsymbol{F_1}}]^{\beta_{nk}}]$ and $Z_{\boldsymbol{F_1}} = \sum_{n=1}^{N} \exp(f_{1n})$.

Finally, the distribution of the first hidden layer $\boldsymbol{f_1}$ is formulated as

$$P(\boldsymbol{F_1}) = \{P(f_{1n}) = \frac{1}{Z_{\boldsymbol{F_1}}} \exp(f_{1n})\}_{n=1}^{N} \tag{16}$$

Overall, we introduce a novel probabilistic representation for fully connected layers in two aspects. First, a fully connected layer formulates a multivariate Gibbs distribution (Equation 8, 12, and 16) and the energy functions is equivalent to the negative activation. Second, the connection between two adjacent fully connected layers can be formulated as a PoE model (Equation 11 and 15).

The proposed probabilistic representation has two advantages over the GP explanations. First, the FoE explanation does not require any assumption, e.g., we formulate $P(f_{2k})$ as a PoE model, in which all the experts $\{\frac{1}{Z_{\boldsymbol{F_1}}} \exp(f_{1n})\}_{n=1}^{N}$ are defined by all the neurons $\{f_{1n}\}_{n=1}^{N}$. As a comparison, the GP explanation derives $P(f_{2k})$ as a Gaussian distribution in the limit of infinite neurons, i.e., $\{f_{1n}\}_{n=1}^{N \to \infty}$, under the assumption that $\{f_{1n}\}_{n=1}^{N}$ are i.i.d.. Second, the Gibbs distribution provides a more explicit explanation for the functionality of a fully connected layer. Specifically, a Gibbs distribution derives the probability of the multiple features defined by the neurons occurring in the input, but the covariance function of a GP cannot clearly explain the functionality of each neuron.

Moreover, we prove that a convolutional layer formulates a specific Gibbs distribution, namely the MRF model, in Appendix H. More specifically, a convolutional layer $\boldsymbol{f_i}$ with $K$ convolutional channels, i.e., $\boldsymbol{f_i} = \{\boldsymbol{f_i^k} = \sigma_i(\sum_{n=1}^{N} \boldsymbol{S_i^{(k,n)}} \circ \boldsymbol{f_{i-1}^n} + b_i^k \cdot \mathbf{1})\}_{k=1}^{K}$, formulate a MRF model as

$$P(\boldsymbol{F_i}) = \frac{1}{Z_{\boldsymbol{F_i}}} \prod_{k=1}^{K} \exp[\varphi_i^k(\boldsymbol{f_{i-1}})] = \frac{1}{Z_{\boldsymbol{F_i}}} \exp[\sum_{k=1}^{K} \varphi_i^k(\boldsymbol{f_{i-1}})] \tag{17}$$

where $\boldsymbol{f_{i-1}} = \{\boldsymbol{f_{i-1}^n}\}_{n=1}^{N}$ is the input, $\varphi_i^k(\boldsymbol{f_{i-1}}) = \sum_{n=1}^{N} \boldsymbol{S^{(k,n)}} \circ \boldsymbol{f_{i-1}^n} + b_i^k \cdot \mathbf{1})$ is the $k$th linear convolutional channel, $\boldsymbol{S^{(k,n)}}$ is the $k$th convolutional filter applying to the $n$th input channel, $b_i^k$ is the bias, $\sigma_i(\cdot)$ is an non-linear operator, and the partition function $Z_{\boldsymbol{F_i}} = \int \exp[\sum_{k=1}^{K} \varphi^k(\boldsymbol{f_{i-1}})] d\boldsymbol{f_{i-1}}$. The corresponding energy function $E_{\boldsymbol{F_i}} = -\sum_{k=1}^{K} \varphi^k(\boldsymbol{f_{i-1}})$.

In summary, the distribution of a hidden layer $\boldsymbol{f_i}$ can be formulated as a Gibbs distribution $P(\boldsymbol{F_i})$, especially the energy function $E_{\boldsymbol{F_i}}$ establish a connection between the deterministic function of the hidden layer and the Gibbs distribution $P(\boldsymbol{F_i})$. The correspondence between Gibbs distributions and hidden layers can be summarized by Table 2.

Table 2: The correspondence between hidden layers and Gibbs distributions

| Architecture | neuron $f_{1n}$ | fully connected layer $\boldsymbol{f_1} = \{f_{1n}\}_{n=1}^{N}$ | conv. channel $\boldsymbol{f_i^k}$ | conv. layer $\boldsymbol{f_i} = \{\boldsymbol{f_i^k}\}_{k=1}^{K}$ |
|---|---|---|---|---|
| Explanation | energy function $E_{F_{1n}} = -f_{1n}$ | discrete Gibbs $\{P(f_{1n})\}_{n=1}^{N}$ | energy function $E_{\boldsymbol{F_i}} = -\sum_{k=1}^{K} \varphi^k(\boldsymbol{f_{i-1}})$ | MRF $\frac{1}{Z_{\boldsymbol{F_i}}} \exp[\sum_{k=1}^{K} \varphi_i^k(\boldsymbol{f_{i-1}})]$ |

We use Equation 16 and Equation 17 as examples to illustrate the correspondence.

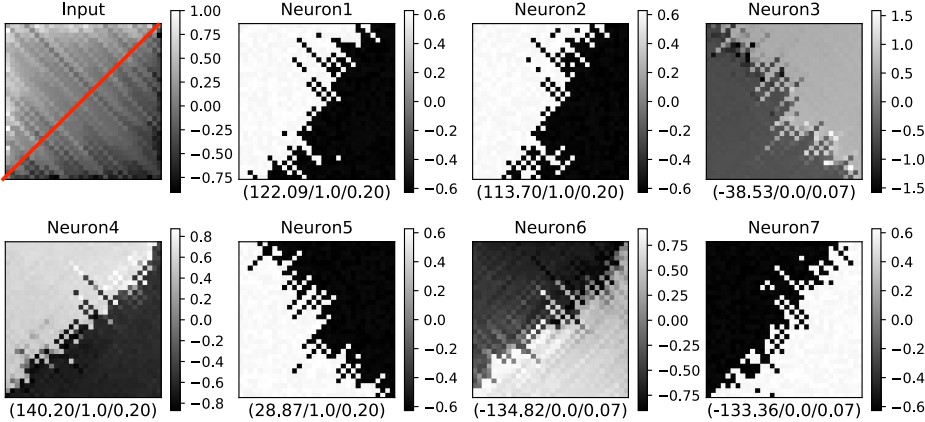

Figure 4: The top-left figure shows a $32 \times 32$ synthetic image as the input, which is sampled from the Gaussian distribution $\mathcal{N}(0, 1)$ and sorted in the primary diagonal direction by the descending order. The remaining figures visualize all the weights $\{\alpha_{mn}\}_{m=1}^{1024}$ of the seven neurons. In addition, the subscript in each figure, i.e., $(g_n(\boldsymbol{x}), f_n(\boldsymbol{x}), P(f_n))$, denotes the linear output $g_n(\boldsymbol{x})$, activation $f_n(\boldsymbol{x})$, and Gibbs probability $P(f_n)$.

## 5 SIMULATIONS

This section uses two synthetic datasets to demonstrate the proposed probabilistic representation: (i) the distribution of a fully connected layer is a multivariate discrete Gibbs distribution and (ii) the distribution a convolutional layer is a MRF model.

### 5.1 THE DISTRIBUTION OF A FULLY CONNECTED LAYER IS A MULTIVARIATE DISCRETE GIBBS DISTRIBUTION

A fully connected layer with $N$ neurons, i.e., $\boldsymbol{f} = \{f_n\}_{n=1}^N$, formulates a multivariate discrete Gibbs distribution to describe the probability of the $N$ features defined by their respective neurons occurring in the input. The energy function is equal to the negative activation, i.e., $E_{f_n} = -f_n(\boldsymbol{x})$.

$$P(\boldsymbol{F}) = \{P(f_n) = \frac{1}{Z_{\boldsymbol{F}}} \exp[f_n(\boldsymbol{x})]\}_{n=1}^N \tag{18}$$

where $f_n(\boldsymbol{x}) = \sigma[g_n(\boldsymbol{x})]$, $g_n(\boldsymbol{x}) = \sum_{m=1}^M \alpha_{mn} \cdot x_m + b_n$ is a linear filter with $\alpha_{mn}$ being the weights and $b_n$ being the bias, $\sigma(\cdot)$ is the activation function, and $Z_{\boldsymbol{F}} = \sum_{n=1}^N \exp[f_n(\boldsymbol{x})]$ is the partition function.

Since benchmark datasets consist of very complex features and typical DNNs have too complicated architecture, it is very hard to directly demonstrate the Gibbs distribution explanation. Alternatively, we choose the neural network for classifying the synthetic dataset in Appendix A.4, because the neural network only has a single fully connected hidden layer with seven neurons, i.e., $N = 7$, and the synthetic dataset only has four features. After the network is trained well, we choose a synthetic image as the input and show all the information of the learned hidden layer in Figure 4.

Since the input image is sorted in the primary diagonal direction, the input feature can be described as high values above the secondary diagonal (the red line) and low values below the red line. Figure 4 shows that Neuron 1, Neuron 2 and Neuron 4 correctly describe the input feature, thus they derive large positive outputs (i.e., $g_1(\boldsymbol{x}) = 122.09$, $g_2(\boldsymbol{x}) = 113.70$, and $g_4(\boldsymbol{x}) = 140.20$). However, Neuron 6 and Neuron 7 do not describe the input feature correctly, thus they derive large negative outputs (i.e., $g_6(\boldsymbol{x}) = -134.08$, $g_7(\boldsymbol{x}) = -133.36$). After applying the sigmoid function, we have $f_1(\boldsymbol{x}) = f_2(\boldsymbol{x}) = f_4(\boldsymbol{x}) = 1$ and $f_6(\boldsymbol{x}) = f_7(\boldsymbol{x}) = 0$. Finally, we derive the probability of different neurons, i.e., $P(f_1) = P(f_2) = P(f_4) = 0.20$ and $P(f_6) = P(f_7) = 0.07$, which indicates that the features defined by Neuron 1, Neuron 2 and Neuron 4 have high probability occurring in the input but the features defined by Neuron 6 and Neuron 7 have low probability occurring in the input.

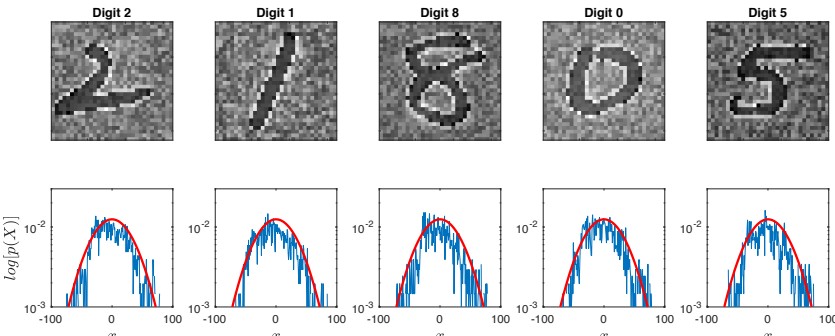

Figure 5: The first row shows five synthetic images of handwritten digits, the second row visualizes their respective histograms, and the red curve indicates the Gaussian distribution $\mathcal{N}(0, 1024)$.

## 5.2 THE DISTRIBUTION OF A CONVOLUTIONAL LAYER IS A MRF MODEL

Since the distributions of benchmark datasets are unknown, it is impossible to use them to verify the probabilistic explanation for convolutional layers. Alternatively, we generate a synthetic dataset obeying the Gaussian distribution $\mathcal{N}(0, 1024)$ based on the NIST dataset of handwritten digits. It consists of 20,000 $32 \times 32$ grayscale images in 10 classes (digits from 0 to 9), and each class has 1,000 training images and 1,000 testing images. Figure 5 shows five synthetic images and their perspective histograms. The method for generating the synthetic dataset is included in Appendix I. We design a CNN $= \{\boldsymbol{x}; \boldsymbol{f_1}; \boldsymbol{f_2}; \boldsymbol{f_3}; \boldsymbol{f_4}; \boldsymbol{f_Y}\}$ for classifying the synthetic dataset. The CNN has two convolutional layers with ReLU (i.e., $\boldsymbol{f_1}$ and $\boldsymbol{f_3}$), two max-pooling layers (i.e., $\boldsymbol{f_2}$ and $\boldsymbol{f_4}$) and a softmax output layer $\boldsymbol{f_Y}$. The detailed information about the CNN is included in Appendix I.

Based on the MRF explanation (Equation 17), the distribution of $\boldsymbol{f_1}$ can be formulated as

$$P(\boldsymbol{F_1}) = \frac{1}{Z_{\boldsymbol{F_1}}} \prod_{k=1}^{20} \exp[\varphi_1^k(\boldsymbol{x})] = \frac{1}{Z_{\boldsymbol{F_1}}} \exp[\sum_{k=1}^{20} \varphi_1^k(\boldsymbol{x})] \tag{19}$$

where $\varphi_1^k(\boldsymbol{x}) = \boldsymbol{S^k} \circ \boldsymbol{x} + b_1^k \cdot \boldsymbol{1}$ is the $k$th linear convolutional channel, $\boldsymbol{S^k}$ is the linear convolutional filter and $b_1^k$ is the bias. The energy function of $P(\boldsymbol{F_1})$ is equivalent to $\boldsymbol{E_{F_1}} = -\sum_{k=1}^{20} \boldsymbol{f_1^k}$.

In order to model a high dimensional dataset, e.g., the $32 \times 32$ synthetic image, we typically need a multivariate distribution with the same dimension, e.g., the dimension of the covariance function of a Gaussian process should be $1024 \times 1024$. However, most structured probabilistic graphical models, especially the MRF model, have two fundamental assumptions: stationary and Markov (Lyu & Simoncelli, 2007). The former means that the distribution is independent on the location, i.e., the distribution of pixels at different locations are identically distributed. The later indicates that the distribution of a single pixel is independent on the other pixels given its neighbors. The two assumptions allow us to use the histogram of samples to simulate the true distribution.

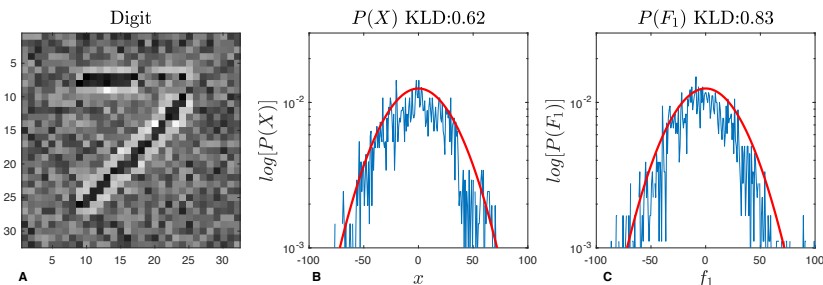

Figure 6: The red curve indicates the truly prior distribution $P(\boldsymbol{X}) = \mathcal{N}(0, 1024)$. The blue curves are different histograms. (A) the synthetic image $\boldsymbol{x}$ is the input. (B) the histogram of $\boldsymbol{x}$, i.e., $P(\boldsymbol{x})$, and $\mathrm{KL}[P(\boldsymbol{X})||P(\boldsymbol{x})] = 0.62$. (C) the histogram of $\boldsymbol{E_{F_1}}$ for estimating $P(\boldsymbol{F_1})$ and $\mathrm{KL}[P(\boldsymbol{X})||P(\boldsymbol{F_1})] = 0.83$.

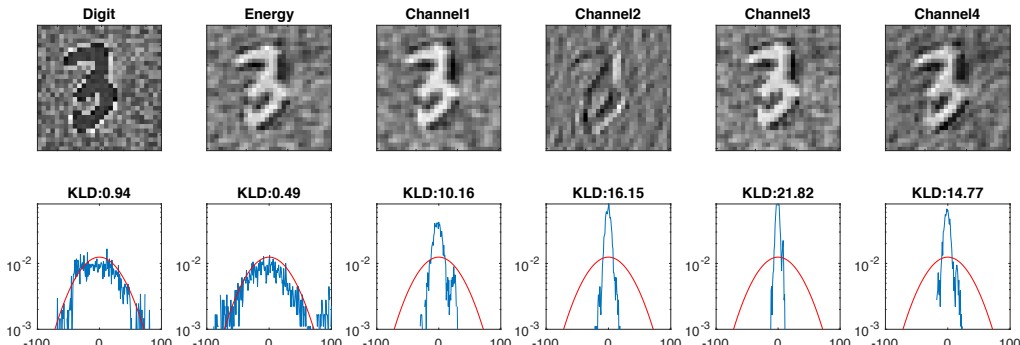

Figure 7: The firs row shows an synthetic image, the energy functions of $P(\boldsymbol{F_1})$ and five different experts, i.e., $\{P(\boldsymbol{F_1^n}) = \frac{1}{Z_{\boldsymbol{F_1^n}}}\exp(\boldsymbol{f_1^n})\}_{n=1}^5$. The second row shows their respective distributions, and the KLDs denote their distances to the true distribution, i.e., $\mathrm{KLD}[P(\boldsymbol{X})||P(\boldsymbol{F_1^n})]$.

After the CNN is well trained, we randomly choose a testing image $\boldsymbol{x}$ as the input to derive $P(\boldsymbol{F_1})$. Since $\boldsymbol{x} \sim P(\boldsymbol{X})$, we can use the histogram of the testing image, i.e., $P(\boldsymbol{x})$, to estimate $P(\boldsymbol{X})$. In addition, $\boldsymbol{E_{F_1}}$ is a sufficient statistics for $P(\boldsymbol{F_1})$ such that we can use the histogram of $\boldsymbol{E_{F_1}}$ to estimate $P(\boldsymbol{F_1})$. As a result, we can verify the probabilistic representation by calculating the distance between $P(\boldsymbol{F_1})$ and $P(\boldsymbol{X})$, i..e, $\mathrm{KL}[P(\boldsymbol{X})||P(\boldsymbol{F_1})]$. We visualize their distributions in Figure 6, which shows that $P(\boldsymbol{F_1})$ is very close to $P(\boldsymbol{X})$, i.e., $\mathrm{KL}(P(\boldsymbol{X})||q(\boldsymbol{F_1})) = 0.83$, which demonstrates the distribution of a convolutional layer can be modeled as a MRF model.

In addition, the experiment provides a new viewpoint to demonstrate the activations of different convolutional channels are not i.i.d.. Given another synthetic image, we visualize the output of five convolutional channels in $\boldsymbol{f_1}$ in Figure 7. It shows that the activations of different channels at the same position are correlated, i.e., they are not independent. We also estimate the distribution of each convolutional channel based on the histograms of their respective energy functions. It shows that the distributions of different channels are different, i.e., they are not identically distributed.

## 6 DISCUSSION

Based on the proposed probabilistic explanations for hidden layers, we discuss certain theoretical and practical topics of deep learning. First, we further confirm the equivalence between SGD and the variational inference. Second, we demonstrate that the entire architecture of DNNs can be explained as a Bayesian network, in which the hidden layers close to the input formulate prior distributions of the training dataset. Third, we propose a novel regularization method to improve the generalization performance of DNNs by pre-training the hidden layers corresponds to the prior distributions.

### 6.1 SGD CAN BE EXPLAINED AS A VARIATIONAL INFERENCE

Recent works prove that SGD works as a specific Bayesian posterior inference, i.e., the variational inference (Mandt et al., 2017; Chaudhari & Soatto, 2018). Here we further confirm the Bayesian explanation for SGD based on the proposed probabilistic representation. Briefly speaking, given a DNN $= \{\boldsymbol{x}; \boldsymbol{f_1}; ...; \boldsymbol{f_I}; \boldsymbol{f_Y}\}$, we prove that the conditional distribution of $\boldsymbol{f_Y}$ given $\boldsymbol{x}$, i.e., $P(\boldsymbol{F_Y}|\boldsymbol{X})$, is also a Gibbs distribution and its energy function is equivalent to the deterministic function formulated by the entire architecture of the DNN, i.e., the DNN describes a family of Gibbs distribution $\mathcal{Q}$. Based on the definition of variational inference, SGD aims to find a Gibbs distribution $P^*(\boldsymbol{F_Y}|\boldsymbol{X})$ such that it can minimize the distance between $P(\boldsymbol{F_Y}|\boldsymbol{X})$ and the true posterior distribution $P(\boldsymbol{Y}|\boldsymbol{X})$ is determined by the training dataset $\mathcal{D}$.

$$P^*(\boldsymbol{F_Y}|\boldsymbol{X}) = \underset{P\in\mathcal{Q}}{\arg\min}\,\mathrm{KL}[P(\boldsymbol{Y}|\boldsymbol{X})||P(\boldsymbol{F_Y}|\boldsymbol{X})] \tag{20}$$

Moreover, we prove that SGD is also equivalent to the energy minimization optimization, which in turn validates the Gibbs explanation for the hidden layer of DNNs. The proof is in Appendix J.

We justify the theoretical explanation for SGD based on the CNN classifying the synthetic dataset and the MLP classifying the MNIST dataset in Appendix K.1. In addition, we demonstrate that the learned parameters via SGD are not i.i.d. based on in Appendix K.2, which indirectly justifies that GP with the i.i.d. assumption for the parameters of DNNs cannot correctly clarify DNNs.

## 6.2 The entire architecture of DNNs can be explained as a Bayesian network

Since the output of the hidden layer $f_i$ in the DNN $= \{x; f_1; ...; f_I; f_Y\}$ is commonly the input of the next layer $f_{i+1}$, the DNN formulates a Markov chain $F_1 \rightarrow \cdots \rightarrow F_I \rightarrow F_Y$. As a result, the distribution of a hidden layer in the DNN should be regarded as a conditional distribution and the entire architecture of the DNN can be formulated as a joint distribution.

$$P(F_Y; F_I..; F_1|X) = P(F_Y|F_I) \cdot ... \cdot P(F_{i+1}|F_i) \cdot ... \cdot P(F_1|X) \tag{21}$$

Richard & Lippmann (1991) prove that $P(F_Y|X)$ is an estimation of the true posterior distribution $P(Y|X) \propto P(X|Y)P(Y)$ and we demonstrate that $P(F_1|X)$ is a prior distribution of the training dataset and the equivalence between SGD and the variational inference, thus the entire architecture of DNNs can be explained as a Bayesian network. That provides a probabilistic explanation for the hierarchy property of DNNs, which cannot be clarified by GP (Matthews et al., 2018).

## 6.3 A novel regularization algorithm for DNNs

Bayesian theory indicates that a better prior distribution corresponds to a better posterior distribution given the same likelihood distribution, thus pre-training the hidden layers corresponding to prior distributions should be an effective approach to improve the generalization performance of DNNs. Erhan et al. (2010) first demonstrate that the unsupervised pre-training is an effective regularization for DNNs through capturing the prior knowledge of the input. However, since they do not know which hidden layer corresponds to a prior distribution, they have to pre-training all the hidden layers, which restricts its wide application due to the computation complexity. In addition, Le et al. (2018) prove that incorporating the prior knowledge of the input results in uniform stability and provides a bound on generalization error for a specific DNN, i.e., the supervised auto-encoders.

Based on the proposed probabilistic representation, we first justify that a better prior distribution indeed has a positive correlation to the generalization performance in Appendix K.3. Subsequently, we propose a novel a novel regularization algorithm to improve the generalization performance of DNNs through pre-training the hidden layers only corresponding to prior distributions and show that it outperforms classical regularizations, e.g., dropout (Srivastava et al., 2014), for the image recognition task on the benchmark CIFAR-10 dataset in Appendix L.

## 7 Conclusion

In this paper, we demonstrate that the parameters of DNNs do not satisfy the i.i.d assumption and the activations being i.i.d. cannot be valid for all the hidden layers of DNNs. As a result, GP cannot correctly explain all the hidden layer of DNNs. Alternatively, we propose a novel probabilistic representation for DNNs in four aspects: (i) a hidden layer formulates a Gibbs distribution, of which the energy function is determined by the activations, (ii) the connection between two adjacent layers can be modeled by a PoE model, (iii) the entire architecture of DNNs can be explained as a Bayesian network, and (iv) the SGD can be explained as the Bayesian variational inference algorithm.

There are two general directions for future research. The theoretical direction is to examine the probabilistic representation in other popular DNNs, e.g., the residual networks, which can further verify the proposed probabilistic representation. The practical direction is to validate the proposed regularization in more complex DNNs and datasets.

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

Table 3: The architecture of the MLP for MNIST classification

| Layer | Description | Dimension | Filter dimension ($M \times N$) |
|:-:|:-:|:-:|:-:|
| $\boldsymbol{x}$ | Input | 784 | — |
| $\boldsymbol{f_1}$ | FC(sigmoid) | 128 | $128 \times 784$ |
| $\boldsymbol{f_2}$ | FC(sigmoid) | 64 | $64 \times 128$ |
| $\boldsymbol{f_Y}$ | Output(softmax) | 10 | $10 \times 64$ |

FC denotes the fully connected layer
Dimension: the number of nodes (i.e., neurons) in every layer.
Filter dimension ($M \times N$): a layer has $M$ filters with $N \times 1$ dimension

# A  THE PARAMETERS OF DNNS CANNOT SATISFY THE I.I.D. PRIOR

## A.1  A MLP ON THE MNIST DATASET

We design a MLP (Figure 1) to classify the benchmark MNIST dataset. The MLP has two hidden layers, each hidden layer is a fully connected layer with the sigmoid activation function, and the output layer is the softmax. Table 3 summarizes the architecture of the MLP.

After the MLP is well trained (its training and testing accuracy is $97.7\%$ and $97.1\%$, respectively), we quantify the correlation of different weights in the same layer based on Equation 22 and derive the correlation matrix for all the weights in every hidden layer.

$$r(\boldsymbol{\alpha}_n, \boldsymbol{\alpha}_{n'}) = \frac{\sum_{m=1}^{M}(\alpha_{mn} - \overline{\alpha}_n)(\alpha_{mn'} - \overline{\alpha}_{n'})}{\sqrt{\sum_{m=1}^{M}(\alpha_{mn} - \overline{\alpha}_n)^2 \sum_{m=1}^{M}(\alpha_{mn'} - \overline{\alpha}_{n'})^2}} \approx Corr(\boldsymbol{\alpha}_n, \boldsymbol{\alpha}_{n'}) \quad (22)$$

where $\overline{\alpha}_n = \frac{1}{M}\sum_{m=1}^{M}\alpha_{mn}$ is the sample mean of $\{\alpha_{mn}\}_{m=1}^{M}$.

To examine if different weights are correlated, we only need to check whether their correlation coefficients are close to zero, so we take the absolute value of the correlation matrix element-wisely. Figure 8 visualizes the correlation matrices of each hidden layer, i.e., $A'_{N \times N}$, $B'_{K \times K}$, and $C'_{L \times L}$.

We find that most weights in $\boldsymbol{f_1}$ are uncorrelated because their correlation coefficients are very close to zero, i.e., $|r(\boldsymbol{\alpha}_n, \boldsymbol{\alpha}_{n'})| \approx 0$, but many weights in $\boldsymbol{f_2}$ and $\boldsymbol{f_Y}$ are correlated to others, i.e., $|r(\boldsymbol{\beta}_k, \boldsymbol{\beta}_{k'})| \neq 0$ and $|r(\boldsymbol{\gamma}_l, \boldsymbol{\gamma}_{l'})| \neq 0$. In addition, we derive the average of the absolute correlation coefficients of all the weights in each layer and show them at the top of Figure 8, which further validates that the weights of some hidden layer are correlated.

Therefore, the parameters of the MLP cannot satisfy the i.i.d. prior.

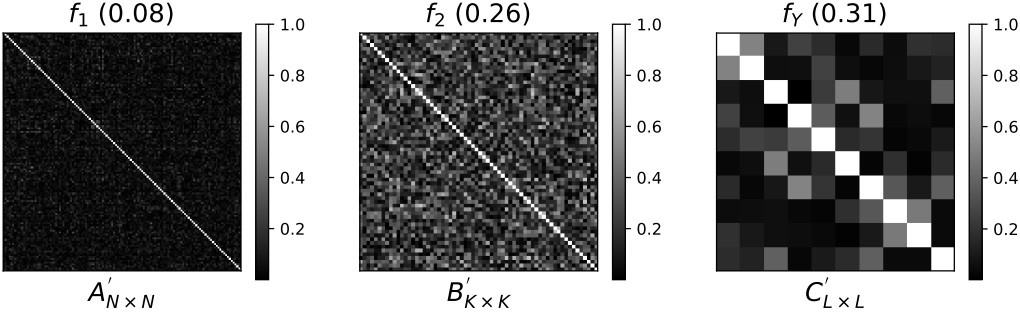

Figure 8: Three absolute correlation matrices for the weights in $\boldsymbol{f_1}$, $\boldsymbol{f_2}$, and $\boldsymbol{f_Y}$. In addition, 0.08, 0.26, 0.31 are the average of the absolute correlation coefficients for all the weights in $\boldsymbol{f_1}$, $\boldsymbol{f_2}$, and $\boldsymbol{f_Y}$, respectively.

Table 4: The architecture of the NN $= \{\boldsymbol{x}; \boldsymbol{f}; \boldsymbol{f_Y}\}$ for MNIST classification

| Layer | Description | Dimension | Filter dimension ($M \times N$) |
|---|---|---|---|
| $\boldsymbol{x}$ | Input | 784 | — |
| $\boldsymbol{f}$ | FC(sigmoid) | 128 | $128 \times 784$ |
| $\boldsymbol{f_Y}$ | Output(softmax) | 10 | $10 \times 128$ |

FC denotes the fully connected layer
Dimension: the number of nodes (i.e., neurons) in every layer.
Filter dimension ($M \times N$): a layer has $M$ filters with $1 \times N$ dimension

## A.2 A NEURAL NETWORK WITH A SINGLE HIDDEN LAYER ON THE MNIST DATASET

Based on the seminal work (Neal, 1994), we design a neural network with a single fully connected hidden layer (Figure 9) for classifying the benchmark MNIST dataset (LeCun et al., 1998). The input layer dimension is $1 \times 784$, the hidden layer has 128 neurons with the sigmoid activation function, and the output layer is the softmax with $1 \times 10$ dimension. Table 4 summarizes the architecture of the network. In addition, $\{\alpha_{mn}\}_{m=1}^{M}$ denote the weights of the $n$th linear filter in the fully connected hidden layer and $\{\beta_{nl}\}_{n=1}^{N}$ are the weights of the $l$th linear filter in the output layer. Therefore, the hidden layer has 128 linear filters with $1 \times 784$ dimension, i.e., $\{\alpha_{mn}\}_{m=1}^{784}$. The output layer has totally 10 linear filters and their dimension is $1 \times 128$, i.e., $\{\beta_{nl}\}_{n=1}^{128}$.

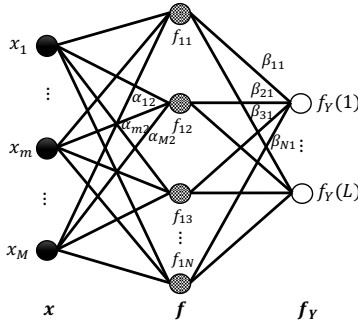

Figure 9: A neural network with a single hidden layer NN $= \{\boldsymbol{x}; \boldsymbol{f}; \boldsymbol{f_Y}\}$.

After the network is well trained (its training and testing accuracy is 97.4% and 96.9%, respectively), we calculate the absolute sample correlations, i.e., $|r(\boldsymbol{\alpha}_n, \boldsymbol{\alpha}_{n'})|$ and $|r(\boldsymbol{\beta}_k, \boldsymbol{\beta}_{k'})|$, and derive two absolute correlation matrices, i.e., $A'_{N \times N}$ and $B'_{L \times L}$, based on Equation 2. They are shown in Figure 10. We find that most $|r(\boldsymbol{\alpha}_n, \boldsymbol{\alpha}_{n'})|$ are close to zero, but $|r(\boldsymbol{\beta}_k, \boldsymbol{\beta}_{k'})|$ not. That indicates the parameters of the NN do not satisfy the necessary condition, thus they cannot satisfy the i.i.d. prior.

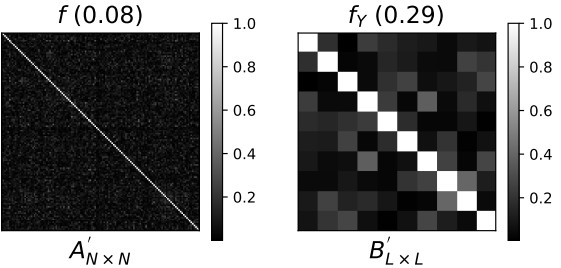

Figure 10: Two absolute correlation matrices for the weights in $\boldsymbol{f}$ and $\boldsymbol{f_Y}$. In addition, 0.08 and 0.29 are the average of the absolute correlation coefficients for all the weights in $\boldsymbol{f}$ and $\boldsymbol{f_Y}$, respectively.

Table 5: The architectures of the CNN for CIFAR-10 classification

| Layer | Description | Output dimension | Filters dimension | Correlation matrix |
|:---:|:---:|:---:|:---:|:---:|
| $x$ | Input | $32 \times 32 \times 3$ | — | — |
| $f_1$ | Conv $(3 \times 3)$ + Maxpool | $16 \times 16 \times 64$ | $3 \times 3 \times 3 \times 64$ | $A_{192 \times 192}$ |
| $f_2$ | Conv $(3 \times 3)$ + Maxpool | $8 \times 8 \times 128$ | $3 \times 3 \times 64 \times 128$ | $B_{8192 \times 8192}$ |
| $f_3$ | FC + Sigmoid | 1024 | $8192 \times 1024$ | $C_{1024 \times 1024}$ |
| $f_4$ | FC + Sigmoid | 256 | $1024 \times 256$ | $D_{256 \times 256}$ |
| $f_Y$ | Output(softmax) | 10 | $256 \times 10$ | $E_{10 \times 10}$ |

## A.3 A CNN ON THE CIFAR-10 DATASET

We design a CNN to classify the benchmark CIFAR-10 dataset (Krizhevsky, 2009). The CNN has two convolutional layers without activation function, two max-pooling layers for dimension reduction, and two fully connected layers with the sigmoid activation function. In addition, the output layer is the softmax. Table 5 summarizes the architecture of the CNN.

After we train the CNN well (its training and testing accuracy are $98.0\%$ and $69.4\%$, respectively). We quantify the correlation of different weights in the same layer based on Equation 2, thereby deriving the absolute correlation matrix for all the weights in every hidden layer.

Since $x$ has 3 channels and $f_1$ has 64 channels, $f_1$ has totally $3 \times 64 = 192$ filters. Hence, the dimension of the correlation matrix for $f_1$ is $192 \times 192$. In particular, we flatten the output of the second convolutional layer, i.e., reshaping $8 \times 8 \times 128$ to $1 \times 8192$, thus the filter dimension of $f_3$ is $8192 \times 1024$. Table 5 shows the dimension of the correlation matrices for all the layers in the CNN.

Figure 11 shows the absolute correlation matrices for every layer. We can find that most weights are correlated in $f_1$ and $f_2$, i.e., the weights in the convolutional layers are correlated. In addition, the weighs in the output layer are also shown to be correlated. However, the weights in $f_3$ and $f_4$ are not correlated, because their correlation coefficients are close to zero. In other words, the weights in the fully connected layers are uncorrelated. Moreover, we derive the average of the absolute correlation coefficients of all the weights in each layer and show them at the top of Figure 11, which further validates the correlation of the weights in each layer. Overall, the parameters of the CNN cannot satisfy the necessary condition. Therefore, they cannot satisfy the i.i.d. prior.

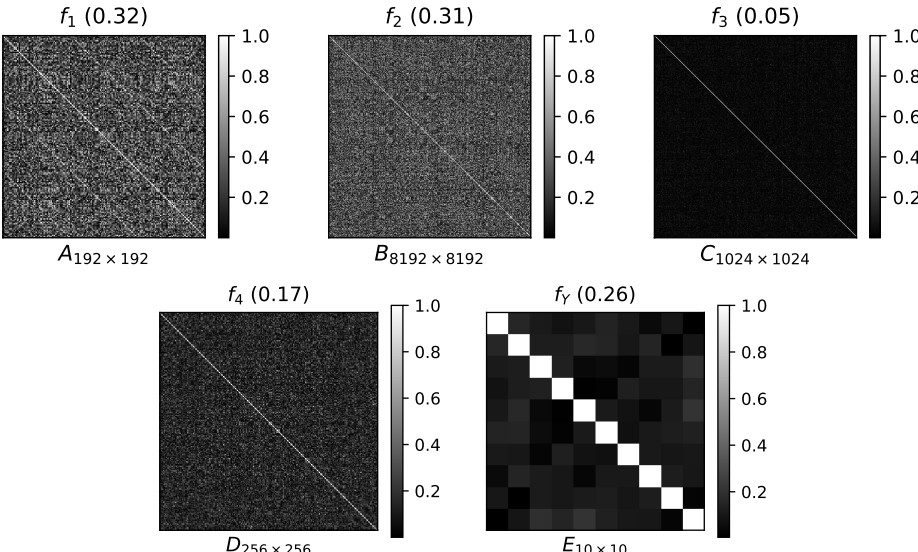

Figure 11: Five absolute correlation matrices for the weights in the layer of CNN. The number at the top of the figure is the average of the absolute correlation coefficients for all the weights of the layer

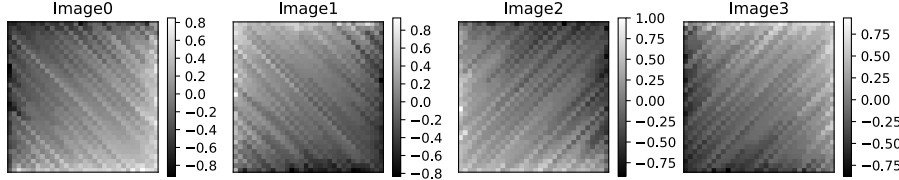

Figure 12: Four synthetic images with different features. All images are sampled from $\mathcal{N}(0,1)$. The only difference is that they are sorted in different diagonal directions by the ascending or descending orders. Image0 and Image1 are sorted in the primary diagonal direction by the ascending order and the descending order, respectively. Image2 and Image3 are sorted in the secondary diagonal direction by the ascending order and the descending order, respectively.

### A.4 Neural networks with a single hidden layer on a synthetic dataset

In order to further demonstrate that the parameters of DNNs cannot satisfy the i.i.d. assumption, we construct a neural network, i.e., NN $= \{x; f; f_Y\}$ shown in Figure 9, to classify a synthetic dataset. The synthetic dataset consists of 4,800 $32 \times 32$ grayscale images with 4 classes, and every synthetic image is sampled from the Gaussian distribution $\mathcal{N}(0,1)$. To make the synthetic image have certain features that can be learned by neural networks, we sort the synthetic image in the primary or secondary diagonal directions by the ascending or descending orders. Thus, the synthetic dataset only has four features, i.e., four classes, and their corresponding images are drawn above.

There are two important reasons why we choose the synthetic dataset. Above all, compared to benchmark datasets containing too complex features, the synthetic dataset only has four simple features, thus we can easily show the correlation of different parameters by simply visualizing them. In addition, since a single synthetic image only has a single feature, we can easily examine the connection between the parameters of networks and the synthetic image.

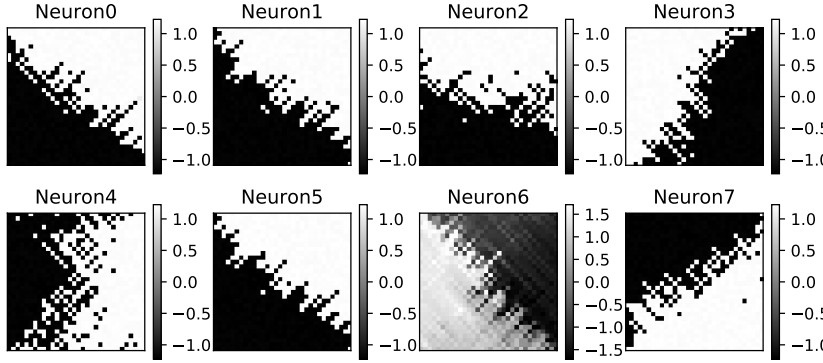

Figure 13: Eight neurons in the fully connected hidden layer. Their dimension is $32 \times 32$

Since the synthetic dataset is very simple, we set the hidden layer $f$ have only eights neurons and the output layer $f_Y$ have four output nodes. After training the network well, we show all the eight neurons in Figure 13. In contrast to the i.i.d. assumption, their weights show strong correlation in three aspects. First, weights show strong internal correlation within each neuron, i.e., if a weight has large magnitude, its neighbors have high probability to be large, and vice versa. Second, weights show strong external correlation between different neurons. For example, most weights of Neuron1 and Neuron5 at the same position show strong positive correlation, and most weights of Neuron3 and Neuron7 at the same position show strong negative correlation. Third, the weights show strong dependence on the training dataset. More specifically, since all the features of the synthetic dataset concentrate in the primary and secondary diagonal directions, most linear filters also demonstrate related features in the same directions.

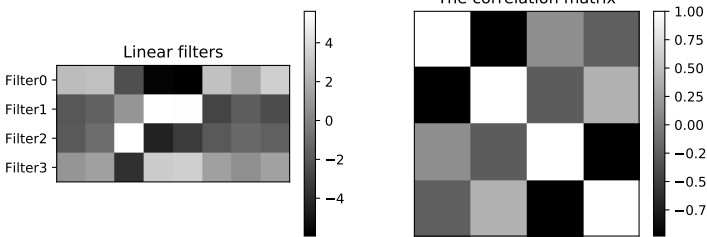

Figure 14: The left figure visualizes the four linear filters in the output layer. The right figure visualizes the correlation matrix of the four filters in the output layer.

Figure 14 visualizes the four linear filters in the output layer. Though they not show too strong internal correlation because their dimension is too small, their weights at the same position still show strong external correlation, e.g., the weights of Filter0 and Filter1 at the same position show strong negative correlation. Moreover, we calculate the correlation coefficients of all the four filters based on Equation 2 and derive the correlation matrix show in Figure 14. It shows that the correlation coefficient of Filter0 and Filter1 is close to $-1$, which further validates that they have strong negative correlation, hence the parameters of the output layer do not satisfy the i.i.d. prior.

Compared to the simulations using benchmark datasets, the synthetic dataset enable us to expose the correlation of different weights and their dependence on the training dataset in the more clearly way. Overall, we demonstrate that the parameters of neural networks are not i.i.d..

## B  PROOF: THE NECESSARY CONDITION FOR THE ACTIVATIONS OF HIDDEN LAYERS BEING INDEPENDENT

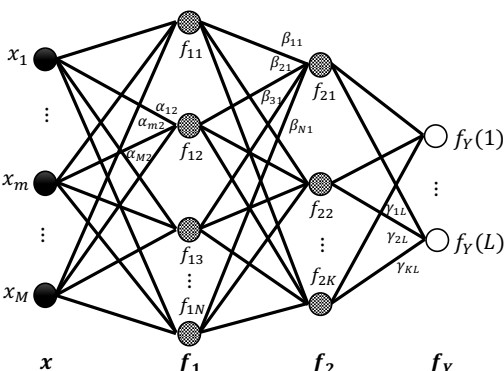

Figure 15: A MLP $= \{\boldsymbol{x}; \boldsymbol{f_1}; \boldsymbol{f_2}; \boldsymbol{f_Y}\}$, in which $\boldsymbol{f_1}$ has $N$ neurons, i.e., $\boldsymbol{f_1} = \{f_{1n} = \sigma_1[g_{1n}(\boldsymbol{x})]\}_{n=1}^N$, where $g_{1n}(\boldsymbol{x}) = \sum_{m=1}^M \alpha_{mn} \cdot x_m + b_{1n}$ is the $n$th linear filter, $\alpha_{mn}$ is the weight of the edge between $x_m$ and $f_{1n}$, $b_{1n}$ is the bias, and $\sigma_1$ is a non-linear activation function. Similarly, $\boldsymbol{f_2} = \{\sigma_2[g_{2k}(\boldsymbol{f_1})]\}_{k=1}^K$ where $g_{2k}(\boldsymbol{f_1}) = \sum_{n=1}^N \beta_{nk} \cdot f_{1n} + b_{2k}$. In addition, since $\boldsymbol{f_Y}$ is softmax, $\boldsymbol{f_Y} = \{\frac{1}{Z_Y} \exp[g_{yl}(\boldsymbol{f_2})]\}_{l=1}^L$ where $g_{yl}(\boldsymbol{f_2}) = \sum_{k=1}^K \gamma_{kl} \cdot f_{2k} + b_{yl}$, and the partition function is $Z_Y = \sum_{l=1}^L \exp[g_{yl}(\boldsymbol{f_2})]$.

In the context of Bayesian probability, all the weights and the biases in the MLP (Figure 15) have prior distributions, i.e., $\alpha_{mn} \sim P(A_{mn})$, $\beta_{nk} \sim P(W_{nk})$, $b_{1n} \sim P(B_{1n})$, and $b_{2k} \sim P(B_{2k})$, thus we regard $G_{1n} = \sum_{m=1}^M A_{mn} X_m + B_{1n}$ as the random variable of $g_{1n}(\boldsymbol{x}) = \sum_{m=1}^M \alpha_{mn} \cdot x_m + b_{1n}$, and $G_{2k} = \sum_{n=1}^N W_{nk} F_{1n} + B_{2k}$ as the random variable of $g_{2k}(\boldsymbol{f_1}) = \sum_{n=1}^N \beta_{nk} \cdot f_{1n} + b_{2k}$. Therefore, the random variables of arbitrary activations in $\boldsymbol{f_1}$ and $\boldsymbol{f_2}$, e.g., $f_{1n}$ and $f_{2k}$, can be expressed as $F_{1n} = \sigma_1(G_{1n})$ and $F_{2k} = \sigma_2(G_{2k})$, respectively.

Based on the theorem (Appendix D) that functions of independent random variables are independent, we can derive that if $\{F_{2k}\}_{k=1}^K$ are independent and the activation function $\sigma_1(\cdot)$ is invertible, then $\{G_{2k}\}_{k=1}^K$ are independent. In other words, in order to prove $\{F_{2k}\}_{k=1}^K$ are independent, we only need to prove $\{G_{2k}\}_{k=1}^K$ being independent as long as $\sigma_2(\cdot)$ is invertible.

The necessary condition for $\{G_{2k}\}_{k=1}^K$ being independent is $\forall (k, k') \in S_1 = \{(k, k') \in \mathbb{Z}^2 | k \neq k', 1 \leq k \leq K, 1 \leq k' \leq K\}$, the covariance $Cov(G_{2k}, G_{2k'}) = 0$, which can be formulated as

$$Cov(G_{2k}, G_{2k'}) = Cov(\sum_{n=1}^N W_{nk}F_{1n} + B_{2k}, \sum_{n'=1}^N W_{n'k'}F_{1n'} + B_{2k'}) = 0 \tag{23}$$

Based on $Cov(X + Y, W + Z) = Cov(X, W) + Cov(X, Z) + Cov(Y, W) + Cov(Y, Z)$, we can derive that

$$\begin{aligned}
Cov(G_{2k}, G_{2k'}) &= \sum_{n=1}^N \sum_{n'=1}^N Cov(W_{nk}F_{1n}, W_{n'k'}F_{1n'}) + \sum_{n=1}^N Cov(W_{nk}F_{1n}, B_{2k'}) \\
&+ \sum_{n'=1}^N Cov(W_{n'k'}F_{1n'}, B_{2k}) + Cov(B_{2k}, B_{2k'})
\end{aligned} \tag{24}$$

Based on the definition of covariance, i.e., $Cov(A, B) = E(AB) - E(A)E(B)$, we have

$$\begin{aligned}
Cov(W_{nk}F_{1n}, W_{n'k'}F_{1n'}) &= E(W_{nk}W_{n'k'} \cdot F_{1n}F_{1n'}) - E(W_{nk}F_{1n})E(W_{n'k'}F_{1n'}) \\
Cov(W_{nk}F_{1n}, B_{2k'}) &= E(W_{nk}B_{2k'}F_{1n}) - E(W_{nk}F_{1n})E(B_{2k'})
\end{aligned} \tag{25}$$

Based on the law of total expectation, i.e, $E(AB) = E_A(AE_{B|A}(B))$, we have

$$\begin{aligned}
E(W_{nk}W_{n'k'} \cdot F_{1n}F_{1n'}) &= E_{W_{nk}W_{n'k'}}(W_{nk}W_{n'k'} \cdot E_{F_{1n}F_{1n'}|W_{nk}W_{n'k'}}(F_{1n}F_{1n'})) \\
E(W_{nk}B_{2k'}F_{1n}) &= E_{W_{nk}B_{2k'}}(W_{nk}B_{2k'} \cdot E_{F_{1n}|W_{nk}B_{2k'}}(F_{1n})) \\
E(W_{nk}F_{1n}) &= E_{W_{nk}}(W_{nk} \cdot E_{F_{1n}|W_{nk}}(F_{1n}))
\end{aligned} \tag{26}$$

Since the statistical properties of the input $\{F_{1n}\}_{n=1}^N$ are unrelated to the weights $\beta_{nk}$ and biases $b_{2k}$, we have $E_{F_{1n}F_{1n'}|W_{nk}W_{n'k'}}(F_{1n}F_{1n'}) = E_{F_{1n}F_{1n'}}(F_{1n}F_{1n'})$, $E_{F_{1n}|W_{nk}B_{2k'}}(F_{1n}) = E_{F_{1n}}(F_{1n})$, and $E_{F_{1n}|W_{nk}}(F_{1n}) = E_{F_{1n}}(F_{1n})$. As a result, we can derive that

$$\begin{aligned}
E(W_{nk}W_{n'k'} \cdot F_{1n}F_{1n'}) &= E_{W_{nk}W_{n'k'}}(W_{nk}W_{n'k'})E_{F_{1n}F_{1n'}}(F_{1n}F_{1n'}) \\
E(W_{nk}B_{2k'}F_{1n}) &= E_{W_{nk}B_{2k'}}(W_{nk}B_{2k'})E_{F_{1n}}(F_{1n}) \\
E(W_{nk}F_{1n}) &= E_{W_{nk}}(W_{nk})E_{F_{1n}}(F_{1n})
\end{aligned} \tag{27}$$

Substituting Equation (27) for the corresponding expectations in Equation (25), we have

$$\begin{aligned}
Cov(W_{nk}F_{1n}, W_{n'k'}F_{1n'}) &= E(W_{nk}W_{n'k'})E(F_{1n}F_{1n'}) - E(W_{nk})E(W_{n'k'})E(F_{1n})E(F_{1n'}) \\
Cov(W_{nk}F_{1n}, B_{2k'}) &= E(W_{nk}B_{2k'})E(F_{1n}) - E(W_{nk})E(B_{2k'})E(F_{1n})
\end{aligned} \tag{28}$$

Assuming $\{F_{1n}\}_{n=1}^N$ are i.i.d. and $F_{1n} \sim P(F_1)$, we have $E(F_{1n}F_{1n'}) = E(F_{1n})E(F_{1n'}) = E^2(F_1)$. Hence, we can derive that

$$\begin{aligned}
Cov(W_{nk}F_{1n}, W_{n'k'}F_{1n'}) &= Cov(W_{nk}, W_{n'k'})E^2(F_1) \\
Cov(W_{nk}F_{1n}, B_{2k'}) &= Cov(W_{nk}, B_{2k'})E(F_1)
\end{aligned} \tag{29}$$

Substituting Equation (29) for the corresponding covariances in Equation (24), we have

$$\begin{aligned}
Cov(G_{2k}, G_{2k'}) &= \sum_{n=1}^N \sum_{n'=1}^N Cov(W_{nk}, W_{n'k'})E^2(F_1) + \sum_{n=1}^N Cov(W_{nk}, B_{2k'})E(F_1) \\
&+ \sum_{n'=1}^N Cov(W_{n'k'}, B_{2k})E(F_1) + Cov(B_{2k}, B_{2k'})
\end{aligned} \tag{30}$$

Based on $Cov(X + Y, W + Z) = Cov(X + W) + Cov(X + Z) + Cov(Y + W) + Cov(Y + Z)$, we can derive that

$$Cov(G_{2k}, G_{2k'}) = Cov(\sum_{n=1}^{N} W_{nk}, \sum_{n'=1}^{N} W_{n'k'})E^2(F_1) + Cov(\sum_{n=1}^{N} W_{nk}, B_{2k'})E(F_1)$$
$$+ Cov(\sum_{n'=1}^{N} W_{n'k'}, B_{2k})E(F_1) + Cov(B_{2k}, B_{2k'}) \tag{31}$$

Therefore, given $\sigma_2(\cdot)$ is invertible and $\{F_{1n}\}_{n=1}^{N}$ are i.i.d., the necessary condition for $\{F_{2k}\}_{k=1}^{K}$ being independent can be expressed as

$$Cov(G_{2k}, G_{2k'}) = Cov(W_k, W_{k'})E^2(F_1) + Cov(W_k, B_{2k'})E(F_1)$$
$$+ Cov(W_{k'}, B_{2k})E(F_1) + Cov(B_{2k}, B_{2k'}) = 0 \tag{32}$$

where $W_k = \sum_{n=1}^{N} W_{nk}$ and $W_{k'} = \sum_{n=1}^{N} W_{nk'}$.

Overall, in order to guarantee the activations of a hidden layer are independent with each other in the context of Bayesian probability, the prior distributions of the weights and biases of the hidden layer should satisfy the above necessary condition given the inputs of the hidden layer are i.i.d. and the activation functions are invertible.

## C PROOF: THE NECESSARY CONDITION FOR THE ACTIVATIONS OF HIDDEN LAYERS BEING IDENTICALLY DISTRIBUTED

Given the same DNN drawn in Figure 15, we assume $\sigma_2(\cdot)$ being strictly increasing and differentiable, so $\sigma_2(\cdot)$ is invertible and its inverse $\sigma_2^{-1}(\cdot)$ is also strictly increasing. The cumulative distribution function of $F_{2k}$ can be expressed as

$$\Phi_{F_{2k}}(f) = \phi(F_{2k} \leq f)$$
$$= \phi(\sigma_1(G_{2k}) \leq f)$$
$$= \phi(G_{2k} \leq \sigma_1^{-1}(f)) \tag{33}$$
$$= \Phi_{G_{2k}}(\sigma_1^{-1}(f))$$

where $\phi(F_{2k} \leq f)$ is the probability of $F_{2k}$ takes on a value less than or equal to $f$. Subsequently, we can obtain that

$$P_{F_{2k}}(f) = \frac{\partial \Phi_{F_{2k}}(f)}{\partial f} = \frac{\partial \Phi_{G_{2k}}(\sigma_2^{-1}(f))}{\partial f}$$
$$= P_{G_{2k}}(\sigma_2^{-1}(f))\frac{\partial \sigma_2^{-1}(f)}{\partial f} \tag{34}$$

Equation (34) means that if the activation function $\sigma_2(\cdot)$ is strictly increasing and differentiable, $\{F_{2k}\}_{k=1}^{K}$ being identically distributed implies $\{G_{2k}\}_{k=1}^{K}$ being identically distributed.

In the context of Bayesian probability, $G_{2k} = \sum_{n=1}^{N} W_{nk}F_{1n} + B_{2k}$, we can derive that

$$E(G_{2k}) = \sum_{n=1}^{N} E(W_{nk}F_{1n}) + E(B_{2k}) \tag{35}$$

Based on the law of total expectation and Equation 27, we have

$$E(W_{nk}F_{1n}) = E_{W_{nk}}(W_{nk})E_{F_{1n}}(F_{1n}) \tag{36}$$

Assuming $\{F_{1n}\}_{n=1}^{N}$ are i.i.d. and $F_{1n} \sim F_1$, we have $E(F_{1n}) = E(F_1)$. Hence, we can derive that

$$E(G_{2k}) = E(F_1)\sum_{n=1}^{N} E(W_{nk}) + E(B_{2k}) \tag{37}$$

Based on the property of expectation, i.e., $E(A) + E(B) = E(A + B)$, we have

$$E(G_{2k}) = E(F_1)E(\sum_{n=1}^{N} W_{nk}) + E(B_{2k}) \tag{38}$$

Therefore, given $\sigma_2(\cdot)$ is strictly increasing and differentiable and $\{F_{1n}\}_{n=1}^{N}$ are i.i.d., the necessary condition for $\{F_{2k}\}_{k=1}^{K}$ being identically distributed is that $\forall (n, n') \in S_1$, we have

$$E(F_1)E(W_k) + E(B_{2k}) = E(F_1)E(W_{k'}) + E(B_{2k'}) \tag{39}$$

where $W_k = \sum_{n=1}^{N} W_{nk}$ and $W_{k'} = \sum_{n=1}^{N} W_{nk'}$.

Overall, in order to guarantee the activations of a hidden layer are identically distributed, the prior distributions of the weights and the biases of the hidden layer should satisfy the above necessary condition in the context of Bayesian probability given the inputs of the hidden layer are i.i.d. and the activation functions are strictly increasing and differentiable.

## D THEOREM: FUNCTIONS OF INDEPENDENT RANDOM VARIABLES ARE INDEPENDENT

Let $X$ and $Y$ be independent random variables on a probability space $(\Omega, \mathcal{F}, \mathbb{P})$. Let $g$ and $h$ be real-valued functions defined on the codomains of $X$ and $Y$, respectively. Then $g(X)$ and $h(Y)$ are independent random variables.

**Proof:** Let $A \subseteq \mathbb{R}$ and $B \subseteq \mathbb{R}$ be the range of $g$ and $h$, the joint distribution between $g(X)$ and $h(Y)$ can be formulate as $\mathbb{P}(g(X) \in A, h(Y) \in B)$. Let $g^{-1}(A)$ and $h^{-1}(B)$ denote the preimages of $A$ and $B$, respectively, we have

$$\mathbb{P}(g(X) \in A, h(Y) \in B) = \mathbb{P}(X \in g^{-1}(A), Y \in h^{-1}(B)) \tag{40}$$

Based on the definition of independence, we can derive that

$$\begin{aligned} \mathbb{P}(g(X) \in A, h(Y) \in B) &= \mathbb{P}(X \in g^{-1}(A))\mathbb{P}(Y \in h^{-1}(B)) \\ &= \mathbb{P}(g(X) \in A)\mathbb{P}(h(Y) \in B) \end{aligned} \tag{41}$$

Based on the definition of preimage, we can derive that

$$\mathbb{P}(g(X) \in A, h(Y) \in B) = \mathbb{P}(g(X) \in A)\mathbb{P}(h(Y) \in B) \tag{42}$$

Therefore, $g(X)$ and $h(Y)$ are independent random variables.

## E THE NECESSARY CONDITIONS FOR ACTIVATIONS BEING I.I.D. ARE VALID FOR CONVOLUTIONAL LAYERS

Assuming a DNN $= \{x; f_1; ...; f_I; f_Y\}$ has two adjacent convolutional layers $f_i$ and $f_{i+1}$. The first convolutional layer has $N$ channels, i.e. $f_i = \{f_i^n\}_{n=1}^{N}$, and the second convolutional layer has $K$ channels, i.e. $f_{i+1} = \{f_{i+1}^k\}_{k=1}^{K}$. Therefore, the dimension of the convolutional filters $S_{i+1}$ in $f_{i+1}$ are $W \times H \times N \times K$, i.e., the dimension of a single filter connecting the $f_i^n$ channel and the $f_{i+1}^k$ channel are $D(S_{i+1}^{(k,n)}) = W \times H$. As a result, the $f_{i+1}^k$ channel can be formulated as

$$f_{i+1}^k = \sigma_{i+1}(\sum_{n=1}^{N} S_{i+1}^{(k,n)} \circ f_i^n + b_{i+1}^k \cdot \mathbf{1}) \tag{43}$$

where $\sigma_{i+1}$ is the activation function and $b_{i+1}$ is the bias for the $(i+1)$th channel.

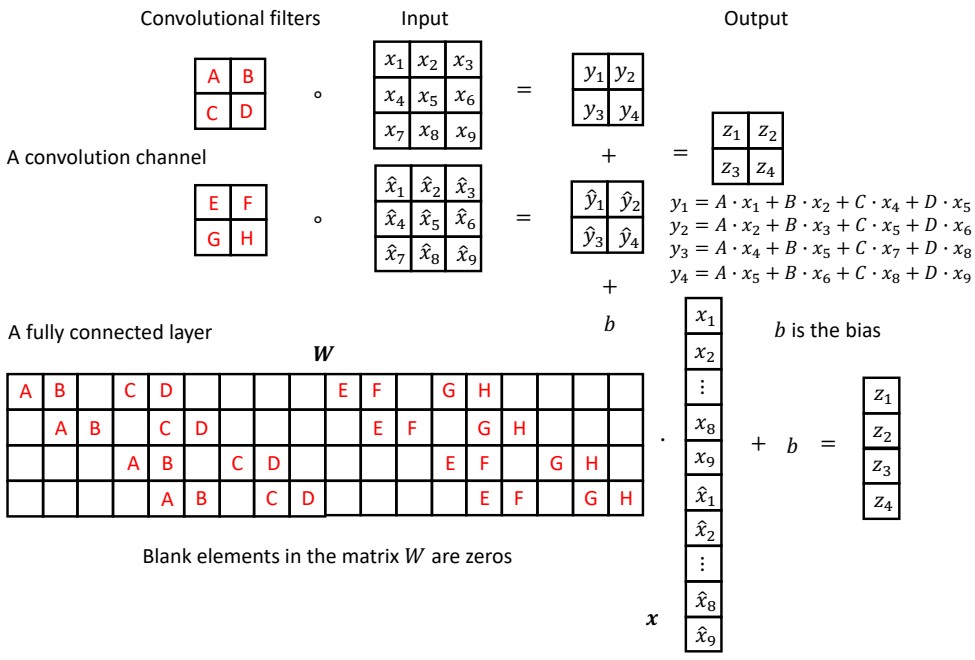

Figure 16: The equivalence between a convolution channel and a fully connected layer.

Since Garriga-Alonso et al. (2018) demonstrate the equivalence between a convolution operation and the matrix multiplication, we can regard the convolutional channel $f_{i+1}^k$ as a fully connected layer. Specifically, the input of the virtual fully connected layer is the same as the input of the convolutional channel, the output of the virtual fully connected layer is the vectorized output of the convolutional channel, and the weights of the virtual fully connected layer depend on the convolutional filter and the spatial location of the output of the convolutional channel.

Without considering the activation function, the equivalence between a convolutional channel and a fully connected layer can be explained in Figure 16, in which the first convolutional layer has two channels, i.e., $f_i = \{x; \hat{x}\}$, where $x = \{x_1, \cdots, x_9\}$ and $\hat{x} = \{\hat{x}_1, \cdots, \hat{x}_9\}$, and the $k$th channel of the second convolutional layer is $f_{i+1}^k = \{z_1, z_2, z_3, z_4\}$. In addition, the dimension of the convolutional filter $S_{i+1}$ is $2 \times 2 \times 2 \times 1$, where $S_{i+1}^{(k,1)} = [A, B; C, D]$ and $S_{i+1}^{(k,2)} = [E, F; G, H]$. Figure 16 shows that the output of $f_{i+1}^k$ can be expressed as a fully connected layer. The input of the fully connected layer is the vector $x$, the weights of each neuron is represented by each row of the matrix $W$, and the bias is $b$, thus $f_{i+1}^k = W \cdot x + b \cdot \mathbf{1}$.

If we consider the activation function and make a precise statement, $f_{i+1}^k$ can be formulated as

$$f_{i+1}^k = \sigma_{i+1}(W_{i+1}^k \cdot f_i + b_{i+1}^k \cdot \mathbf{1}) \tag{44}$$

where $W_{i+1}^k$ is the matrix corresponding to all the convolutional filters in the $k$th channel.

To derive the necessary conditions for two arbitrary different channels are i.i.d., we need to derive the necessary conditions for two arbitrary activations in two arbitrary different channels are i.i.d. For example, if the $m$th activation of the $k$ channel and the $m'$th activation of the $k'$ channel in $f_{i+1}$ are i.i.d., the necessary conditions are $Cov(f_{i+1}^k(m), f_{i+1}^{k'}(m')) = 0$, $E(f_{i+1}^k(m)) = E(f_{i+1}^{k'}(m'))$.

Since we demonstrate the equivalence between a convolutional channel and a fully connected layer, the necessary conditions for activations being i.i.d. in a fully connected layer are also valid for a convolutional layer. In other words, we can also use the Equation 3, Equation 4, and Equation 5 to examine whether activations being i.i.d. is valid in a convolutional layer under the assumption that all activations being i.i.d. in the previous layer.

Table 6: The statistical measures for examining the necessary conditions

| Layer | Activation expectation | Covariance | Weight expectation |
|---|---|---|---|
| $f_1$ | $E(X)$ | $Cov(A_n, A_{n'})$ | $E(A_n)$ |
| $f_2$ | $E(F_1)$ | $Cov(W_k, W_{k'})$ | $E(W_k)$ |
| $f_Y$ | $E(F_2)$ | $Cov(C_l, C_{l'})$ | $E(C_l)$ |

$$\{x_m\}_{m=1}^M \sim P(X), \{f_{1n}\}_{n=1}^N \sim P(F_1), \{f_{2k}\}_{k=1}^K \sim P(F_2)$$
$$A_n = \sum_{m=1}^M A_{mn}, \text{ where } \alpha_{mn} \sim P(A_{mn})$$
$$W_k = \sum_{n=1}^N W_{nk}, \text{ where } \beta_{nk} \sim P(W_{nk})$$
$$C_l = \sum_{k=1}^K C_{kl}, \text{ where } \gamma_{kl} \sim P(C_{kl})$$

## F  ACTIVATIONS BEING I.I.D. CANNOT BE VALID FOR ALL THE HIDDEN LAYERS OF DNNS

In this section, we demonstrate that activations being i.i.d. cannot be valid for all the hidden layers of DNNs through showing that the hidden layers of two typical DNNs, i.e, the MLP and the CNN, cannot satisfy the necessary conditions for activations being i.i.d..

### F.1  THE MLP ON THE MNIST DATASET

The architecture of the MLP for classifying the MNIST dataset is visualized in Figure 15, and its specific parameters are summarized in Table 3. Since the activation functions of all the hidden layers in the MLP are the sigmoid function, which is strictly increasing, differentiable, and invertible, we only need to examine the first two necessary conditions in the MLP.

Since we restrict all the hidden layers from using the bias, the first two necessary conditions for activations being i.i.d. can be simplified as

$$\forall (k, k') \in S_1 = \{(k, k') \in \mathbb{Z}^2 | k \neq k', 1 \leq k \leq K, 1 \leq k' \leq K\}, \text{ we must have}$$
$$Cov(G_{2k}, G_{2k'}) = Cov(W_k, W_{k'})E^2(F_1) = 0 \tag{45}$$

$$E(F_1)E(W_k) = E(F_1)E(W_{k'}) \tag{46}$$

Based on the architecture of the MLP, Table 6 summarizes all the statistical measures for each layer of the MLP. Therefore, the key to examine if the layers of the MLP satisfy the necessary conditions is to estimate the statistical measures.

To estimate the expectations $E(X)$, $E(F_1)$, and $E(F_2)$, we use the sample means $\overline{x}$, $\overline{f_1}$, and $\overline{f_2}$, respectively. The definition of the sample mean is

$$\overline{x} = \frac{1}{M} \sum_{m=1}^M x_m, \overline{f_1} = \frac{1}{N} \sum_{n=1}^N f_2(n), \overline{f_2} = \frac{1}{K} \sum_{k=1}^K f_2(k) \tag{47}$$

After the MLP is well-trained, we derive the sample means as $\overline{x} = 0.131, \overline{f_1} = 0.436,$ and $\overline{f_2} = 0.498$ given the benchmark MNIST testing dataset.

Taking into account the estimation error, we can regard $E(X) = 0$. As a result, we can derive that $\forall (n, n'), Cov(G_{1n}, G_{1n'}) = 0$ and $E(G_{1n}) = E(G_{1n'}) = 0$, thus $f_1$ satisfy the necessary conditions for activations being i.i.d..

However, $\overline{f_1} = 0.436$ and $\overline{f_2} = 0.498$ imply that $E(F_1) \neq 0$ and $E(F_2) \neq 0$ even considering the estimation error. Therefore, we need to estimate $Cov(W_k, W_{k'})$ and $E(W_k)$ to examine whether the activations $\{f_{2k}\}_{k=1}^K$ being i.i.d..

Since the training dataset is commonly viewed as i.i.d., we can use the same training dataset to train the MLP several times to derive multiple independent observation samples of $\{\beta_{nk}(t)\}_{t=1}^T$ to estimate the random variable $W_{nk}$, thereby estimating $Cov(W_k, W_{k'})$ and $E(W_k)$.

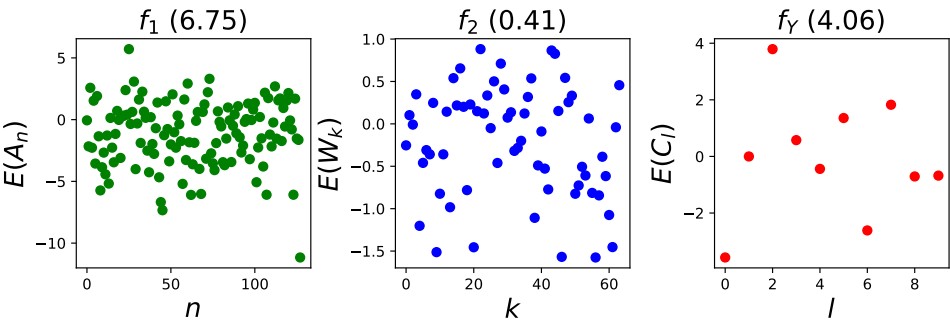

Figure 17: The expectation of the weights summation, i.e., $E(A_n)$, $E(W_k)$, and $E(C_l)$, in each layer. The number at the top of the figure indicates the variance of the points drawn in the figure.

More specifically, we use the sample correlation $r(\boldsymbol{\beta}_k, \boldsymbol{\beta}_{k'})$ to check whether $Cov(W_k, W_{k'})$ are close to zero and use the sample mean $\overline{\beta_k}$ to estimate $E(W_k)$.

$$r(\boldsymbol{\beta}_k, \boldsymbol{\beta}_{k'}) = \frac{\sum_{t=1}^{T}(\beta_k(t) - \overline{\beta_k})(w_{k'}(t) - \overline{\beta_{k'}})}{\sqrt{\sum_{t=1}^{T}(\beta_k(t) - \overline{\beta_k})^2 \sum_{t=1}^{T}(\beta_{k'}(t) - \overline{\beta_{k'}})^2}} \approx Corr(W_k, W_{k'}) \tag{48}$$

where $\overline{\beta_k} = \frac{1}{T}\sum_{t=1}^{T}\beta_k(t)$ is the sample mean of $\{\beta_k(t)\}_{t=1}^{T}$ and $\beta_k(t) = \sum_{n=1}^{N}\beta_{nk}(t)$.

In order to make a tradeoff between the estimation precision and computation complexity, we use the same training dataset to train the MLP 20 times to derive the samples, i.e., $\{\beta_{nk}(t)\}_{t=1}^{20}$. For an activation $f_{2k}$, it has 128 weights because its previous layer has 128 neurons. Hence, a single sample $\beta_k(t) = \sum_{n=1}^{128}\beta_{nk}(t)$. Based on the 20 samples, i.e., $\{\beta_k(t)\}_{t=1}^{20}$, we can calculate $r(\boldsymbol{\beta}_k, \boldsymbol{\beta}_{k'})$ to examine whether $Cov(W_k, W_{k'})$ are close to zero and derive the sample mean to estimate $E(W_k)$. We use the same method to estimate $Cov(A_n, A_{n'})$, $E(A_n)$, $Cov(C_l, C_{l'})$, and $E(C_l)$.

Finally, we shows the absolute correlation matrices i.e., $A_{N \times N}$, $W_{K \times K}$, and $C_{L \times L}$, in Figure 3. Since many elements in the three matrices are far from zero, we conclude that their corresponding covariances, i.e, $Cov(A_n, A_{n'})$, $Cov(W_k, W_{k'})$, and $Cov(C_k, C_{k'})$, are not close to zero based on connection between correlation and covariance

$$Corr(W_k, W_{k'}) = \frac{Cov(W_k, W_{k'})}{\sigma_{W_k}\sigma_{W_{k'}}} \tag{49}$$

where $\sigma_{W_k} > 0$ is the standard derivation of $W_k$.

Figure 17 visualizes the expectations, i.e., $E(A_n)$, $E(W_k)$, and $E(C_l)$, for each layer in the MLP. We find that some expectations show great variations, especially $\{E(A_n)\}_{n=1}^{N}$ and $\{E(C_l)\}_{l=1}^{L}$. In other words, the MLP cannot guarantee that all the expectations of the weights summation are equivalent even though we take into account the estimation error. Therefore, the MLP cannot satisfy the necessary condition for activations being identically distributed.

Overall, we conclude that not all the hidden layers of the MLP can satisfy the necessary conditions for activations being i.i.d. under the assumption that the activations of the previous layer are i.i.d.

## F.2 THE CNN ON THE CIFAR-10 DATASET

The architecture of the CNN for classifying the CIFAR-10 dataset is in Table 5. In particular, the two convolutional layers do not have activations function, i.e., their activation functions can be viewed as $y = x$, which is strictly increasing, differentiable, and invertible. The sigmoid activation function is also strictly increasing, differentiable, and invertible, thus we only need to examine the first two necessary conditions in the CNN. To simplify derivation and decrease computation complexity, we assume the max-pooling layers are special activation functions and satisfy the third necessary condition for activations being i.i.d.

Table 7: The statistical measures for examining the necessary conditions of each layer in the CNN

| | $f_1$ | $f_2$ | $f_3$ | $f_4$ | $f_Y$ |
|---|---|---|---|---|---|
| Description | Conv Maxpool | Conv Maxpool | FC Sigmoid | FC Sigmoid | Output Softmax |
| Activation Dim. | $16{\times}16{\times}64$ | $8{\times}8{\times}128$ | $1024{\times}1$ | $256{\times}1$ | $10{\times}1$ |
| Filter Dim. | $3{\times}3{\times}3{\times}64$ | $3{\times}3{\times}64{\times}128$ | $8192{\times}1024$ | $1024{\times}256$ | $256{\times}10$ |
| Weight Dim. | $M{\times}N$ $(27{\times}64)$ | $N{\times}K$ $(576{\times}128)$ | $K{\times}L$ $(8192{\times}1024)$ | $L{\times}Q$ $(1024{\times}256)$ | $Q{\times}R$ $(256{\times}10)$ |
| Weight PDF | $a_{mn}{\sim}P(A_{mn})$ | $b_{nk}{\sim}P(B_{nk})$ | $c_{kl}{\sim}P(C_{kl})$ | $d_{lq}{\sim}P(D_{lq})$ | $f_{qr}{\sim}P(F_{qr})$ |
| Sum. PDF | $A_n{=}\sum_{m=1}^{M}A_{mn}$ | $B_k{=}\sum_{n=1}^{N}B_{nk}$ | $C_l{=}\sum_{k=1}^{K}C_{kl}$ | $D_q{=}\sum_{l=1}^{L}D_{lq}$ | $F_r{=}\sum_{q=1}^{Q}F_{qr}$ |
| Covariance | $Cov(A_n,A_{n'})$ | $Cov(B_k,B_{k'})$ | $Cov(C_l,C_{l'})$ | $Cov(D_q,D_{q'})$ | $Cov(F_r,F_{r'})$ |
| Sum. Exp. | $E(A_n)$ | $E(B_k)$ | $E(C_l)$ | $E(D_q)$ | $E(F_r)$ |
| Activation Exp. | $E(X)$ | $E(F_1)$ | $E(F_2)$ | $E(F_3)$ | $E(F_4)$ |

Dim. is short for dimension. Sum. is short for Weights Summation. Exp. is short for Expectation.

Similar as the MLP, we also restrict all the layers of the CNN from using the bias, thus the first two necessary conditions for activations being i.i.d. can be simplified as

$$\forall (k,k') \in S_1 = \{(k,k') \in \mathbb{Z}^2 | k \neq k', 1 \leq k \leq K, 1 \leq k' \leq K\}, \text{ we must have}$$
$$Cov(G_{2k}, G_{2k'}) = Cov(W_k, W_{k'})E^2(F_1) = 0 \tag{50}$$

$$E(F_1)E(W_k) = E(F_1)E(W_{k'}) \tag{51}$$

Table 7 summarizes all the statistical measures for every layer of the CNN based on the architecture of the CNN in Table 5. The definitions of most random variables are similar as the MLP except the dimension of the weights. To check if two arbitrary convolutional channels are i.i.d., we need to know all the weights of two convolutional channels. Based on the equivalence between a convolutional channel and a fully connected layer (Appendix E), all the weights of a single convolutional channel consist of all the convolutional filters related to the convolutional channel. For example, the dimension of all the convolutional filters related to a convolutional channel in $f_1$ are $3 \times 3 \times 3$, thus a single convolutional channel in $f_1$ has 27 weights, so the weights dimension of $f_1$ is $27 \times 64$.

After training the CNN well (its training accuracy is $96.2\%$ and testing accuracy is $67.9\%$), we use the same methods as the MLP to estimate the above statistical measures in Table 7.

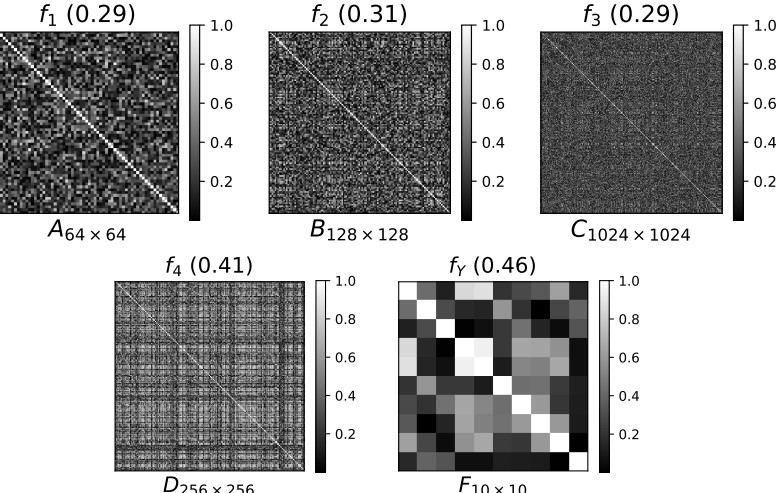

Figure 18: Five absolute correlation matrices for the weight summations in each layer. The number at the top is the average of the absolute correlation coefficients for all the weights summations of the layer

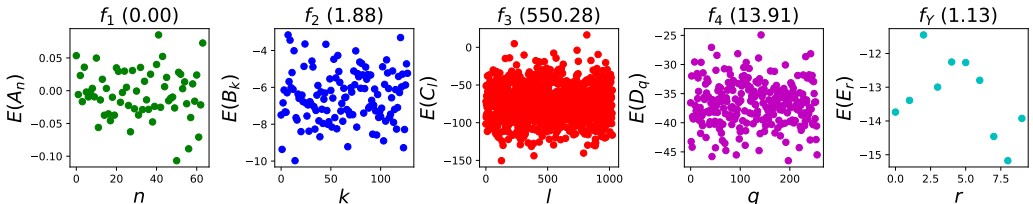

Figure 19: The expectation of the weights summation of each layer in the CNN. The number at the top of the figure indicates the variance of the points drawn in the figure..

For the activation expectations of each layer in the CNN, we have $\overline{x} = -0.023$, $\overline{f_1} = -707.074$, $\overline{f_2} = -6347.836$, $\overline{f_3} = 36.870$, $\overline{f_4} = 15.131$, thus all the expectations cannot be zero except the input layer, i.e., $E(X) = 0$, $E(F_1) \neq 0$, $E(F_2) \neq 0$, $E(F_3) \neq 0$, $E(F_4) \neq 0$.

For the covariances, we show all the correlation matrices of each layer of the CNN in Figure 18. We find that many correlation coefficients are far from zero in all the correlation matrices, thus their corresponding covariances are not close zero either. Based on the estimations for the activation expectation and the covariance between different weights summations in each layer, we can derive that activations being independent cannot be valid for all the layers of the CNN.

For the weights summation expectation, we visualize all the expectations for each layer in the CNN in Figure 19. We find that some expectations have great variations, especially $\{E(C_l)\}_{l=1}^{L}$ and $\{E(D_q)\}_{q=1}^{Q}$, which means that most expectations of weights summations of different neurons in $f_3$ and $f_4$ cannot be equivalent. As a result, the activations of the two fully connected layers in the CNN cannot be identically distributed.

Overall, we conclude that not all the hidden layers of the CNN can satisfy the necessary conditions for activations being i.i.d. under the assumption that the activations of the previous layer are i.i.d.

## G   THE EQUIVALENCE BETWEEN THE GRADIENT DESCENT ALGORITHM AND THE FIRST ORDER APPROXIMATION

In the context of deep learning, most learning algorithms belong to the gradient descent algorithm. Given a DNN $= \{x; f_1; ...; f_I; f_Y\}$ and a cost function $H[f_Y, P(Y|X)]$, where $P(Y|X)$ is the true distribution of the training label given the corresponding training dataset. Let $\theta$ be the parameters of the DNN, the gradient descent aims to optimize $\theta$ by minimizing $H[f_Y, P(Y|X)]$ (Rumelhart et al., 1986). We typically update $\theta$ iteratively to derive $\theta^*$, which can be expressed as

$$\theta_{t+1} = \theta_t - \alpha \nabla_{\theta_t} H[f_Y, P(Y|X)] \tag{52}$$

where $\nabla_{\theta_t} H[f_Y, P(Y|X)]$ is the Jacobian matrix of $H[f_Y, P(Y|X)]$ with respect to $\theta_t$ at the $t$th iteration, and $\alpha > 0$ denotes a constant indicating the learning rate. Since $P(Y|X)$ is constant, we denote $H(f_Y)$ as $H[f_Y, P(Y|X)]$ for simplifying the following derivation.

Since the functions of all hidden layers are differentiable and the output of a hidden layer is the input of its next layer, the Jacobian matrix of $H(f_Y)$ with respect to the parameters of the $i$th hidden layer, i.e., $\nabla_{\theta(i)} H(f_Y)$, can be expressed as follows based on the chain rule.

$$\nabla_{\theta(i)} H(f_Y) = \nabla_{f_Y} H(f_Y) \cdot \nabla_{f_I} f_Y \cdot \prod_{j=i+1}^{I} \nabla_{f_{j-1}} f_j \cdot \nabla_{\theta(i)} f_i \tag{53}$$

where $\theta(i)$ denote the parameters of the $i$th hidden layer.

Based on Equation 52 and Equation 53, $\theta(i)$ can be learned by the gradient descent method as

$$\theta_{t+1}(i) = \theta_t(i) - \alpha [\nabla_{\theta_t(i)} H(f_Y)] \tag{54}$$

Table 8 summarizes the backpropagation training procedure for the MLP in Figure 15.

Table 8: One iteration of the backpropagation training procedure for the MLP

| Layer | Gradients update $\nabla_{\boldsymbol{\theta}_t(i)} H(\boldsymbol{f_Y})$ | | Parameters and activations update | |
|---|---|---|---|---|
| $\boldsymbol{f_Y}$ | $\nabla_{\boldsymbol{f_Y}} H(\boldsymbol{f_Y}) \nabla_{\boldsymbol{\theta}_t(Y)} \boldsymbol{f_Y}$ | $\downarrow$ | $\boldsymbol{\theta}_{t+1}(Y) = \boldsymbol{\theta}_{t+1}(Y) - \alpha[\nabla_{\boldsymbol{\theta}_t(Y)} H(\boldsymbol{f_Y})], \boldsymbol{f_Y}(\boldsymbol{f_2}, \boldsymbol{\theta}_{t+1}(Y))$ | $\uparrow$ |
| $\boldsymbol{f_2}$ | $\nabla_{\boldsymbol{f_Y}} H(\boldsymbol{f_Y}) \nabla_{\boldsymbol{f_2}} \boldsymbol{f_Y} \nabla_{\boldsymbol{\theta}_t(2)} \boldsymbol{f_2}$ | $\downarrow$ | $\boldsymbol{\theta}_{t+1}(2) = \boldsymbol{\theta}_{t+1}(2) - \alpha[\nabla_{\boldsymbol{\theta}_t(2)} H(\boldsymbol{f_Y})], \boldsymbol{f_2}(\boldsymbol{f_1}, \boldsymbol{\theta}_{t+1}(2))$ | $\uparrow$ |
| $\boldsymbol{f_1}$ | $\nabla_{\boldsymbol{f_Y}} H(\boldsymbol{f_Y}) \nabla_{\boldsymbol{f_2}} \boldsymbol{f_Y} \nabla_{\boldsymbol{f_1}} \boldsymbol{f_2} \nabla_{\boldsymbol{\theta}_t(1)} \boldsymbol{f_1}$ | $\downarrow$ | $\boldsymbol{\theta}_{t+1}(1) = \boldsymbol{\theta}_{t+1}(1) - \alpha[\nabla_{\boldsymbol{\theta}_t(1)} H(\boldsymbol{f_Y})], \boldsymbol{f_1}(\boldsymbol{x}, \boldsymbol{\theta}_{t+1}(1))$ | $\uparrow$ |
| $\boldsymbol{x}$ | — | | — | |

The uparrow and downarrow indicate the order of gradients and parameters(activations) update, respectively.

If an arbitrary function $\boldsymbol{f}$ is differentiable at point $\boldsymbol{p}^*$ in $\mathbb{R}^N$ and its differential is represented by the Jacobian matrix $\nabla_{\boldsymbol{p}} \boldsymbol{f}$, the first order approximation of $\boldsymbol{f}$ near the point $\boldsymbol{p}$ can be formulated as

$$\boldsymbol{f}(\boldsymbol{p}) - \boldsymbol{f}(\boldsymbol{p}^*) = (\nabla_{\boldsymbol{p}^*} \boldsymbol{f}) \cdot (\boldsymbol{p} - \boldsymbol{p}^*) + o(||\boldsymbol{p} - \boldsymbol{p}^*||) \tag{55}$$

where $o(||\boldsymbol{p} - \boldsymbol{p}^*||)$ is a quantity that approaches zero much faster than $||\boldsymbol{p} - \boldsymbol{p}^*||$ approaches zero.

Based on the equivalence between the gradient descent method and the first order approximation (Battiti, 1992), updating the activations of the hidden layers of the MLP in Figure 15 during the backpropagation training procedure can be approximated as

$$\boldsymbol{f_2}[\boldsymbol{f_1}, \boldsymbol{\theta}_{t+1}(2)] \approx \boldsymbol{f_2}[\boldsymbol{f_1}, \boldsymbol{\theta}_t(2)] + (\nabla_{\boldsymbol{\theta}_t(2)} \boldsymbol{f_2}) \cdot [\boldsymbol{\theta}_{t+1}(2) - \boldsymbol{\theta}_t(2)]$$
$$\boldsymbol{f_1}[\boldsymbol{x}, \boldsymbol{\theta}_{t+1}(1)] \approx \boldsymbol{f_1}[\boldsymbol{x}, \boldsymbol{\theta}_t(1)] + (\nabla_{\boldsymbol{\theta}_t(1)} \boldsymbol{f_1}) \cdot [\boldsymbol{\theta}_{t+1}(1) - \boldsymbol{\theta}_t(1)] \tag{56}$$

where $\boldsymbol{f_2}[\boldsymbol{f_1}, \boldsymbol{\theta}_t(2)]$ denote the activations of the second hidden layer based on the parameters learned in the $t$th iteration, i.e., $\boldsymbol{\theta}_t(2)$, given the activations of the first hidden layer, i.e., $\boldsymbol{f_1}$. The definitions of $\boldsymbol{f_2}[\boldsymbol{f_1}, \boldsymbol{\theta}_{t+1}(2)]$ and $\boldsymbol{f_1}(\boldsymbol{x}, \boldsymbol{\theta}_t(1))$ are the same as $\boldsymbol{f_2}[\boldsymbol{f_1}, \boldsymbol{\theta}_t(2)]$.

Because $\boldsymbol{f_2}$ has $K$ neurons, i.e., $\boldsymbol{f_2} = \{f_{2k} = \sigma_2(\sum_{n=1}^N \beta_{nk} \cdot f_{1n} + b_{2k})\}_{k=1}^K$, and each neuron has $N+1$ parameters, i.e., $\boldsymbol{\theta}(2) = \{[\beta_{1k}; \cdots; \beta_{Nk}; b_{2k}]\}_{k=1}^K$, the dimension of $\nabla_{\boldsymbol{\theta}_t(2)} \boldsymbol{f_2}$ is equal to $K \times (N+1)$ and $\nabla_{\boldsymbol{\theta}_t(2)} \boldsymbol{f_2}$ can be expressed as

$$\nabla_{\boldsymbol{\theta}_t(2)} \boldsymbol{f_2} = (\nabla_{\sigma_2} \boldsymbol{f_2}) \cdot [\boldsymbol{f_1}; 1]^T \tag{57}$$

where $\nabla_{\sigma_2} \boldsymbol{f_2} = \frac{\partial \boldsymbol{f_2}[\boldsymbol{f_1}, \boldsymbol{\theta}_t(2)]}{\partial \sigma_2}$.

Substituting $(\nabla_{\sigma_2} \boldsymbol{f_2}) \cdot [\boldsymbol{f_1}; 1]^T$ for $\nabla_{\boldsymbol{\theta}_t(2)} \boldsymbol{f_2}$ in Equation 56, we can derive

$$\boldsymbol{f_2}[\boldsymbol{f_1}, \boldsymbol{\theta}_{t+1}(2)] \approx \boldsymbol{f_2}[\boldsymbol{f_1}, \boldsymbol{\theta}_t(2)] + (\nabla_{\sigma_2} \boldsymbol{f_2}) \cdot [\boldsymbol{f_1}; 1]^T \cdot \boldsymbol{\theta}_{t+1}(2) - (\nabla_{\sigma_2} \boldsymbol{f_2}) \cdot [\boldsymbol{f_1}; 1]^T \cdot \boldsymbol{\theta}_t(2) \tag{58}$$

If we only consider a single neuron, such as $f_{2k}$, in $\boldsymbol{f_2}$, $\boldsymbol{\theta}_{t+1}(2k) = [\beta_{1k}; \cdots; \beta_{Nk}; b_{2k}]$ and $\boldsymbol{\theta}_t(2k) = [\beta_{1k}^*; \cdots; \beta_{Nk}^*; b_{2k}^*]$, thus $[\boldsymbol{f_1}; 1]^T \cdot \boldsymbol{\theta}_{t+1}(2k) = \sum_{n=1}^N \beta_{nk} \cdot f_{1n} + b_{2k}$. As a result, Equation 58 with respect to a single neuron can be expressed as

$$f_{2k}[\boldsymbol{f_1}, \boldsymbol{\theta}_{t+1}(2k)] \approx (\nabla_{\sigma_2} f_{2k}) \cdot [\sum_{n=1}^N \beta_{nk} \cdot f_{1n} + b_{2k}] \text{ (First order approximation)}$$
$$+ f_{2k}[\boldsymbol{f_1}, \boldsymbol{\theta}_t(2k)] - (\nabla_{\sigma_2} f_{2k}) \cdot [\sum_{n=1}^N \beta_{nk}^* \cdot f_{1n} + b_{2k}^*] \text{ (Error)} \tag{59}$$

Equation 59 indicates that $f_{2k}[\boldsymbol{f_1}, \boldsymbol{\theta}_{t+1}(2k)]$ can be expressed as its first order approximation with an error component based on the activation in the previous iteration, i.e., $f_{2k}[\boldsymbol{f_1}, \boldsymbol{\theta}_t(2k)]$. Since $\nabla_{\sigma_2} f_{2k} = \frac{\partial f_{2k}[\boldsymbol{f_1}, \boldsymbol{\theta}_t(2)]}{\partial \sigma_2}$ is only related to $\boldsymbol{f_1}$ and the parameters in the $t$th training iteration, i.e, $\boldsymbol{\theta}_t(2)$, it can be regarded as a constant. Also note the error component not contains any parameters in the $(t+1)$th training iteration. In summary, $f_{2k}(\boldsymbol{f_1}, \boldsymbol{\theta}_{t+1}(2k))$ can be reformulated as

$$f_{2k}(\boldsymbol{f_1}, \boldsymbol{\theta}_{t+1}(2k)) \approx C_1 \cdot [\sum_{n=1}^N \beta_{nk} \cdot f_{1n} + b_{2k}] + C_2 \tag{60}$$

where $C_1 = \nabla_{\sigma_2} f_{2k}$ and $C2 = f_{2k}(\boldsymbol{f_1}, \boldsymbol{\theta}_t(2k)) - (\nabla_{\sigma_2} f_{2k}) \cdot [\sum_{n=1}^N \beta_{nk}^* \cdot f_{1n} + b_{2k}^*]$. Similarly, the activations in the first hidden layer, i.e, $\boldsymbol{f_1}$, also can be formulated as the first order approximation in the context of the gradient descent learning algorithm.

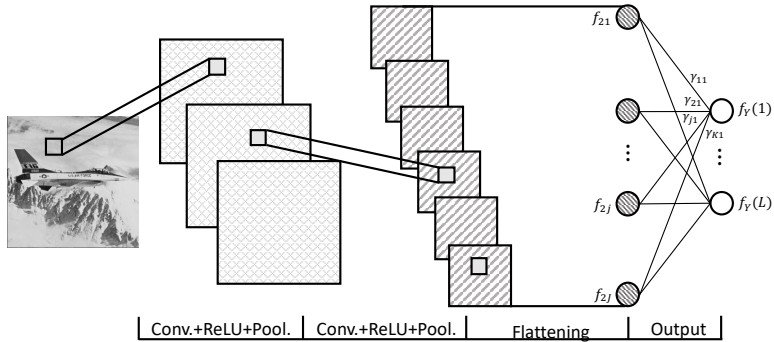

Figure 20: The above CNN has two convolutional layers, i.e., $\boldsymbol{f_1}$, $\boldsymbol{f_2}$, and one softmax output layer, i.e., $\boldsymbol{f_Y}$. We flatten the output of $\boldsymbol{f_2}$ as the input of $\boldsymbol{f_Y}$.

## H THE PROPOSED PROBABILISTIC REPRESENTATION HOLDS FOR CONVOLUTIONAL LAYERS

This section proves that the proposed probabilistic representation is valid for convolutional layers. In the above CNN, the input $\boldsymbol{x}$ has $Q$ channels $\boldsymbol{x} = \{\boldsymbol{x}^q\}_{q=1}^Q$ (e.g., $Q = 3$ if $\boldsymbol{x}$ are color images), $\boldsymbol{f_1}$ has $N$ convolutional channels $\boldsymbol{f_1} = \{\boldsymbol{f_1^n}\}_{n=1}^N$, $\boldsymbol{f_2}$ has $K$ convolutional channels $\boldsymbol{f_2} = \{\boldsymbol{f_2^k}\}_{k=1}^K$. The output layer $\boldsymbol{f_Y}$ is the softmax with $L$ nodes, thus its distribution can be formulated as

$$P(\boldsymbol{F_Y}) = \{P(f_{yl}) = \frac{1}{Z_{\boldsymbol{F_Y}}}\exp(f_{yl})\}_{l=1}^L \tag{61}$$

where $Z_{\boldsymbol{F_Y}} = \sum_{l=1}^L \exp(f_{yl})$ denotes the partition function. Since $\{f_{yl} = \sum_{j=1}^J \gamma_{jl} \cdot f_{2j} + b_{yl}\}_{l=1}^L$, we can derive

$$P(\boldsymbol{F_Y}) = \{P(f_{yl}) = \frac{1}{Z_{\boldsymbol{F_Y}}}\exp(\sum_{j=1}^J \gamma_{jl} \cdot f_{2j} + b_{yl})\}_{l=1}^L \tag{62}$$

where $\{f_{2j}\}_{j=1}^J$ is the flattened output of $\boldsymbol{f_2}$, $\gamma_{jl}$ is the weight of the edge between $f_{2j}$ and $f_{yl}$, and $b_{yl}$ denotes the bias. Since $\boldsymbol{f_2}$ is a convolutional layer with $K$ convolutional channels, it can be reformulated as $\{f_{2j}\}_{j=1}^J = \{\boldsymbol{f_2^k}\}_{k=1}^K$. As a result, $P(\boldsymbol{F_Y})$ can be reformulated as

$$P(\boldsymbol{F_Y}) = \{P(f_{yl}) = \frac{1}{Z_{\boldsymbol{F_Y}}}\exp(\sum_{k=1}^K \boldsymbol{\gamma}_{kl}^T \cdot \boldsymbol{f_2^k} + b_{yl})\}_{l=1}^L \tag{63}$$

where $\boldsymbol{\gamma}_{kl}$ is a subset of all the parameters $\{\gamma_{jl}\}_{j=1}^J$ such that $\sum_{k=1}^K \boldsymbol{\gamma}_{kl}^T \cdot \boldsymbol{f}_2^k = \sum_{j=1}^J \gamma_{jl} \cdot f_{2j}$, and $\boldsymbol{\gamma}_{kl}^T$ is the transpose of $\boldsymbol{\gamma}_{kl}$. Therefore, $P(\boldsymbol{F_Y})$ can be reformulated as

$$P(\boldsymbol{F_Y}) = \{P(f_{yl}) = \frac{1}{Z'_{\boldsymbol{F_Y}}}\prod_{k=1}^K \exp(\boldsymbol{\gamma}_{kl}^T \cdot \boldsymbol{f_2^k})\}_{l=1}^L \tag{64}$$

where $Z'_{\boldsymbol{F_Y}} = Z_{\boldsymbol{F_Y}}/[\exp(b_{yl})]$. Recall the element-wise matrix power, e.g., $\exp(\boldsymbol{a})^{\boldsymbol{b}}$, where $\boldsymbol{a} = [a_1; a_2; a_3]$ and $\boldsymbol{b} = [b_1; b_2; b_3]$, we can derive that $\exp(\boldsymbol{a}) = [\exp(a_1); \exp(a_2); \exp(a_3)]$ and $\exp(\boldsymbol{a})^{\boldsymbol{b}} = [\exp(a_1)^{b_1}; \exp(a_2)^{b_2}; \exp(a_3)^{b_3}] = [\exp(a_1 b_1); \exp(a_2 b_2); \exp(a_3 b_3)]$. As a result, $\exp(\boldsymbol{a}\boldsymbol{b}) = \exp(a_1 b_1 + a_2 b_2 + a_3 b_3) = \prod_{|a|} \exp(\boldsymbol{a})^{\boldsymbol{b}}$, where $|\boldsymbol{a}|$ is the element numbers of $\boldsymbol{a}$.

Based on the element-wise matrix power, we can reformulate $P(\boldsymbol{F_Y})$ as

$$P(\boldsymbol{F_Y}) = \{P(f_{yl}) \approx \frac{1}{Z'_{\boldsymbol{F_Y}}}\prod_{k=1}^K \exp(\boldsymbol{\gamma}_{kl}^T \cdot \boldsymbol{f_2^k}) = \frac{1}{Z'_{\boldsymbol{F_Y}}}\prod_{k=1}^K \prod_{|\boldsymbol{f_2^k}|}[\exp(\boldsymbol{f_2^k})]^{\gamma_{kl}}\}_{l=1}^L \tag{65}$$

Moreover, we introduce a new partition function $Z_{F_2^k} = \sum_{f_2^k} \exp(f_2^k)$ to guarantee that $\frac{1}{Z_{F_2^k}} \exp(f_2^k)$ is a probability measure. As a result, $P(F_Y)$ can be reformulated as

$$P(F_Y) = \{P(f_{yl}) = \frac{1}{Z''_{F_Y}} \prod_{k=1}^K \prod_{|f_2^k|} [\frac{1}{Z_{F_2^k}} \exp(f_2^k)]^{\gamma_{kl}}\}_{l=1}^L \tag{66}$$

where $Z''_{F_Y} = Z'_{F_Y} / \prod_{k=1}^K [Z_{F_2^k}]^{\gamma_{kl} \cdot |f_2^k|}$. Overall, $P(F_Y)$ can be reformulated as a PoE model, in which each expert is defined by every convolutional channel in $f_2$

$$P(f_2^k) = \frac{1}{Z_{F_2^k}} \exp(f_2^k) \tag{67}$$

Since $f_2$ has $K$ convolutional channels, i.e., $f_2 = \{f_2^k\}_{k=1}^K$, and each channel can be formulated as the summation of all the convolutional channels in $f_1$, i.e.,

$$f_2 = \{f_2^k = \sigma_2(\sum_{n=1}^N S_2^{(k,n)} \circ f_1^n + b_2^k \cdot \mathbf{1})\}_{k=1}^K \tag{68}$$

where $S_2^{(k,n)} \circ f_1^n$ is the output of the $k$th convolutional filter applying into the $n$th channel of $f_1$, $b_2^k$ is the bias, and $\sigma_2(\cdot)$ is the activation function. Therefore, we have

$$P(f_2^k) = \frac{1}{Z_{F_2^k}} \exp[\sigma_2(\sum_{n=1}^N S_2^{(k,n)} \circ f_1^n + b_2^k \cdot \mathbf{1})] \tag{69}$$

Based on the equivalence between the gradient descent learning and the first order approximation (Appendix G), Equation 69 can be approximated as

$$P(f_2^k) \approx \frac{1}{Z_{F_2^k}} \exp(\sum_{n=1}^N S_2^{(k,n)} \circ f_1^n + b_2^k \cdot \mathbf{1}) \tag{70}$$

Let us define a linear function as $\varphi_2^k(f_1) = \sum_{n=1}^N S_2^{(k,n)} \circ f_1^n + b_2^k \cdot \mathbf{1}$, thus the distribution of the $k$th convolutional channel in $f_2$, i.e., $P(f_2^k)$, can be expressed as

$$P(f_2^k) \approx \frac{1}{Z_{F_2^k}} \exp[\varphi_2^k(f_1)] \tag{71}$$

If we regard all the linear filters $\{\varphi_2^k(f_1)\}_{k=1}^K$ as potential functions for modeling signal structures of $f_1$, we can formulate $P(F_2)$ as a specific Gibbs distribution, i.e., the Markov Random Fields (MRFs), which can be expressed as

$$P(F_2) = \frac{1}{Z_{F_2}} \prod_{k=1}^K P(f_2^k) = \frac{1}{Z_{F_2}} \exp[\sum_{k=1}^K \varphi_2^k(f_1)] \tag{72}$$

where $Z_{F_2} = \sum_{f_1} \exp[\sum_{k=1}^K \varphi_2^k(f_1)]$ is the partition function for $P(F_2)$. In summary, we prove that the convolutional layer $f_2$ formulates a MRF model to describe the statistical property of $f_1$.

Here we prove that the convolutional layer $f_1$ also formulates a MRF model to describe the statistical property of the input $x$. Based on Equation 70, we have

$$P(f_2^k) \approx \frac{1}{Z'_{F_2^k}} \exp(\sum_{n=1}^N S_2^{(k,n)} \circ f_1^n) \tag{73}$$

where $Z'_{F_2^k} = Z_{F_2^k} / \exp(b_2^k)$.

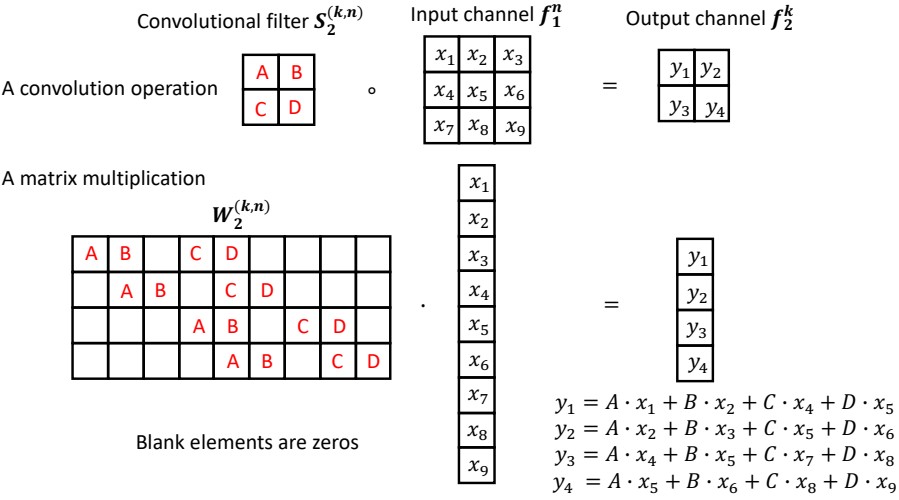

Figure 21: The equivalence between a convolutional operation and a full connected layer.

Based on the equivalence between a convolutional channel and a fully connected layer proven in Appendix E, we can regard a convolutional operation as a matrix multiplication, which is visualized in Figure 21. As a result, we have

$$P(\boldsymbol{f_2^k}) \approx \frac{1}{Z'_{\boldsymbol{F_2^k}}} \exp(\sum_{n=1}^{N} \boldsymbol{S_2^{(k,n)}} \circ \boldsymbol{f_1^n}) = \frac{1}{Z'_{\boldsymbol{F_2^k}}} \exp(\sum_{n=1}^{N} \boldsymbol{W_2^{(k,n)}} \cdot \boldsymbol{f_1^n}) \qquad (74)$$

where $\boldsymbol{W_2^{(k,n)}}$ is a matrix corresponding to the convolution filter $\boldsymbol{S_2^{(k,n)}}$.

Based on the element-wise matrix power, $P(\boldsymbol{f_2^k})$ can be reformulated as

$$P(\boldsymbol{f_2^k}) \approx \frac{1}{Z'_{\boldsymbol{F_2^k}}} \prod_{n=1}^{N} \exp(\boldsymbol{W_2^{(k,n)}} \cdot \boldsymbol{f_1^n}) = \frac{1}{Z'_{\boldsymbol{F_2^k}}} \prod_{n=1}^{N} \exp(\boldsymbol{f_1^n})^{\boldsymbol{W_2^{(k,n)}}} \qquad (75)$$

Similarly, we introduce a new partition function $Z_{\boldsymbol{F_1^n}} = \sum_{\boldsymbol{f_1^n}} \exp(\boldsymbol{f_1^n})$ to guarantee that $\frac{1}{Z_{\boldsymbol{F_1^n}}} \exp(\boldsymbol{f_1^n})$ is a probability measure. As a result, $P(\boldsymbol{f_2^k})$ can be reformulated as

$$P(\boldsymbol{f_2^k}) = P(f_{yl}) \approx \frac{1}{Z''_{\boldsymbol{F_2^k}}} \prod_{n=1}^{N} [\frac{1}{Z_{\boldsymbol{F_1^n}}} \exp(\boldsymbol{f_1^n})]^{\boldsymbol{W_2^{(k,n)}}} \qquad (76)$$

where $Z''_{\boldsymbol{F_2^k}} = Z'_{\boldsymbol{F_2^k}} / \prod_{n=1}^{N} [Z_{\boldsymbol{F_1^n}}]^{\boldsymbol{W_2^{(k,n)}}}$. Therefore, $P(\boldsymbol{f_2^k})$ can be reformulated as a PoE model, in which all the experts are defined as $P(\boldsymbol{f_1^n}) = \frac{1}{Z_{\boldsymbol{F_1^n}}} \exp(\boldsymbol{f_1^n})$

Since $\boldsymbol{f_1}$ has $N$ convolutional channels

$$\boldsymbol{f_1} = \{\boldsymbol{f_1^n} = \sigma_1(\sum_{q=1}^{Q} \boldsymbol{S_1^{(n,q)}} \circ \boldsymbol{x^q} + b_1^n \cdot \boldsymbol{1})\}_{n=1}^{N} \qquad (77)$$

Based on the same derivation as Equations 70, 71, 72, $P(\boldsymbol{F_1})$ also can be expressed as a MRF model

$$P(\boldsymbol{F_1}) = \frac{1}{Z_{\boldsymbol{F_1}}} \prod_{n=1}^{N} P(\boldsymbol{f_1^n}) = \frac{1}{Z_{\boldsymbol{F_1}}} \exp[\sum_{n=1}^{N} \varphi_1^n(\boldsymbol{x})] \qquad (78)$$

where $\varphi_1^n(\boldsymbol{x}) = \sum_{q=1}^{Q} \boldsymbol{S_1^{(n,q)}} \circ \boldsymbol{x^q} + b_1^n \cdot \boldsymbol{1})$ and the partition function $Z_{\boldsymbol{F_1}} = \sum_{\boldsymbol{x}} \exp[\sum_{n=1}^{N} \varphi_1^n(\boldsymbol{x})]$. Overall, the distribution of a convolutional layer can be formulated as a MRF model.

---

**Algorithm 1** The algorithm for generating the synthetic dataset

---

**input:** NIST dataset of handwritten digits by class
1: **repeat**
2:     binarizing an image of NIST to obtain $z$
3:     extracting the central part of $z$ to obtain $z_c$ with dimension $64 \times 64$
4:     downsampling $z_c$ to obtain $z_{cd}$ with dimension $32 \times 32$
5:     extracting the edge of $z_{cd}$ to obtain the mask image $m_{cd}$
6:     decomposing $m_{cd}$ into four parts, i.e., $m_{\text{outside}}$, $m_{\text{outside-boundary}}$, $m_{\text{inside-boundary}}$, and $m_{\text{inside}}$.
7:     sampling $\mathcal{N}(0, 1024)$ to derive a random vector $x \in \mathbb{R}^{1024 \times 1}$
8:     sorting $x$ in the descending order to derive $\hat{x}$
9:     decomposing $\hat{x}$ into four parts, i.e., $\hat{x} = \{\hat{x}_{\text{outside}}, \hat{x}_{\text{outside-boundary}}, \hat{x}_{\text{inside-boundary}}, \hat{x}_{\text{inside}}\}$
10:     placing each pixel of $\hat{x} = \{\hat{x}_{\text{outside}}, \hat{x}_{\text{outside-boundary}}, \hat{x}_{\text{inside-boundary}}, \hat{x}_{\text{inside}}\}$ into the above masks.
11: **until** (20,000 synthetic images are generated)
**output:** The synthetic dataset

---

## I   THE APPROACH TO GENERATE THE SYNTHETIC DATASET BASED ON NIST AND THE ARCHITECTURE OF THE CNN

In this section, we present a novel approach to generate a synthetic dataset obeying the Gaussian distribution based on the NIST [1] dataset of handwritten digits by class. The synthetic dataset has similar characteristics as the benchmark MNIST dataset. It consists of 20,000 $32 \times 32$ grayscale images in 10 classes (digits from 0 to 9), and each class has 1,000 training and 1,000 testing images. More specifically, the approach to generate the synthetic dataset is summarized in Algorithm 1, and Figure 22 visualize the spatial relation of $\hat{x} = \{\hat{x}_{\text{outside}}, \hat{x}_{\text{outside-boundary}}, \hat{x}_{\text{inside-boundary}}, \hat{x}_{\text{inside}}\}$.

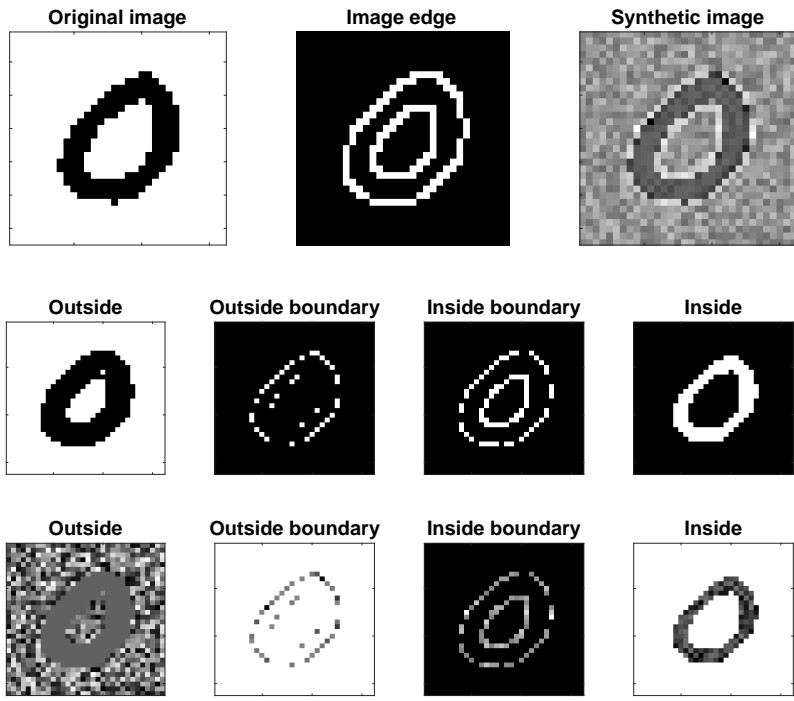

Figure 22:   The first row shows an original image, its edge, and the corresponding synthetic image based on the original one. The second row uses white pixels to show the spatial position of the four mask parts, i.e., $m_{\text{outside}}$, $m_{\text{outside-boundary}}$, $m_{\text{inside-boundary}}$, and $m_{\text{inside}}$. The third row shows the synthetic image corresponding to each mask part, i.e., $\hat{x} = \{\hat{x}_{\text{outside}}, \hat{x}_{\text{outside-boundary}}, \hat{x}_{\text{inside-boundary}}, \hat{x}_{\text{inside}}\}$.

---

[1]https://www.nist.gov/srd/nist-special-database-19

Table 9: The architectures of the CNN for synthetic image classification

| R.V. | Layer | Description | CNN |
|---|---|---|---|
| $\boldsymbol{X}$ | $\boldsymbol{x}$ | Input | $32 \times 32 \times 1$ |
| $\boldsymbol{F_1}$ | $\boldsymbol{f_1}$ | Conv $(3 \times 3)$ + ReLU | $30 \times 30 \times 20$ |
| | $\boldsymbol{f_2}$ | Maxpool | $15 \times 15 \times 20$ |
| $\boldsymbol{F_2}$ | $\boldsymbol{f_3}$ | Conv $(5 \times 5)$ + ReLU | $11 \times 11 \times 60$ |
| | $\boldsymbol{f_4}$ | Maxpool | $5 \times 5 \times 60$ |
| $\boldsymbol{F_Y}$ | $\boldsymbol{f_Y}$ | Output(softmax) | $1 \times 1 \times 10$ |

R.V. is the random variable of the hidden layer(s).

The architecture of the CNN for classifying the synthetic dataset is summarized in Table 9.

## J    THE EQUIVALENCE BETWEEN SGD AND VARIATIONAL INFERENCE

### J.1    THE DISTRIBUTION OF THE OUTPUT LAYER GIVEN THE INPUT IS A GIBBS DISTRIBUTION

In Section 6.2, we demonstrate the entire architecture of a DNN $= \{\boldsymbol{x}; \boldsymbol{f_1}; ...; \boldsymbol{f_I}; \boldsymbol{f_Y}\}$ corresponds to a joint distribution $P(\boldsymbol{F_Y}; \boldsymbol{F_I}..; \boldsymbol{F_1}|\boldsymbol{X})$. As a result, the conditional distribution of $\boldsymbol{f_Y}$ given $\boldsymbol{x}$, i.e., $P(\boldsymbol{F_Y}|\boldsymbol{X})$, can be formulated as

$$P(\boldsymbol{F_Y}|\boldsymbol{X}) = \int \cdots \int P(\boldsymbol{F_Y}; \boldsymbol{F_I}..; \boldsymbol{F_1}|\boldsymbol{X})d\boldsymbol{F_I} \cdots d\boldsymbol{F_1} \tag{79}$$

Since $P(\boldsymbol{F_Y}; \boldsymbol{F_I}..; \boldsymbol{F_1}|\boldsymbol{X}) = P(\boldsymbol{F_Y}|\boldsymbol{F_I}) \cdot ... \cdot P(\boldsymbol{F_{i+1}}|\boldsymbol{F_i}) \cdot ... \cdot P(\boldsymbol{F_1}|\boldsymbol{X})$ and only $\boldsymbol{F_2}$ depends on $\boldsymbol{F_1}$, we can derive

$$P(\boldsymbol{F_Y}|\boldsymbol{X}) = \int \cdots \int P(\boldsymbol{F_Y}|\boldsymbol{F_I}) \cdot ... \cdot P(\boldsymbol{F_3}|\boldsymbol{F_2})d\boldsymbol{F_I} \cdots d\boldsymbol{F_2} \cdot \int P(\boldsymbol{F_2}|\boldsymbol{F_1})P(\boldsymbol{F_1}|\boldsymbol{X})d\boldsymbol{F_1} \tag{80}$$

Since $\int P(\boldsymbol{F_2}|\boldsymbol{F_1})P(\boldsymbol{F_1}|\boldsymbol{X})d\boldsymbol{F_1} = \int P(\boldsymbol{F_2}; \boldsymbol{F_1}|\boldsymbol{X})d\boldsymbol{F_1} = P[\boldsymbol{F_2}|\boldsymbol{f_1}(\boldsymbol{X})]$, where $\boldsymbol{f_1}(\cdot)$ is the deterministic function defined by the corresponding hidden layer, $P(\boldsymbol{F_Y}|\boldsymbol{X})$ can be simplified as

$$P(\boldsymbol{F_Y}|\boldsymbol{X}) = \int \cdots \int P(\boldsymbol{F_Y}|\boldsymbol{F_I}) \cdot ... \cdot P(\boldsymbol{F_3}|\boldsymbol{F_2})P[\boldsymbol{F_2}|\boldsymbol{f_1}(\boldsymbol{X})]d\boldsymbol{F_I} \cdots d\boldsymbol{F_2} \tag{81}$$

Similarly, since only $\boldsymbol{F_3}$ depends on $\boldsymbol{F_2}$, we can obtain

$$P(\boldsymbol{F_Y}|\boldsymbol{X}) = \int \cdots \int P(\boldsymbol{F_Y}|\boldsymbol{F_I}) \cdot ... \cdot P(\boldsymbol{F_3}|\boldsymbol{f_2}(\boldsymbol{f_1}(\boldsymbol{X})))d\boldsymbol{F_I} \cdots d\boldsymbol{F_3} \tag{82}$$

If we iteratively apply the integral with respect to $\boldsymbol{F_3}$ until $\boldsymbol{F_I}$, we can obtain

$$P(\boldsymbol{F_Y}|\boldsymbol{X}) = P[\boldsymbol{F_Y}|\boldsymbol{f_I}(\cdots \boldsymbol{f_2}(\boldsymbol{f_1}(\boldsymbol{X}))\cdots)] \tag{83}$$

Since the output layer $\boldsymbol{f_Y}$ is commonly a fully connected layer. we can formulate $P(\boldsymbol{F_Y}|\boldsymbol{X})$ as a multivariate discrete Gibbs distribution based on Equation 8.

$$P(\boldsymbol{F_Y}|\boldsymbol{X}) = \frac{1}{Z_{\mathrm{DNN}}(\boldsymbol{x})}\exp[\boldsymbol{f_Y}(\boldsymbol{f_I}(\cdots \boldsymbol{f_2}(\boldsymbol{f_1}(\boldsymbol{x}))\cdots))] \tag{84}$$

It is important to note that the energy function $E_{\mathrm{DNN}}(\boldsymbol{f_Y}|\boldsymbol{x}) = -\boldsymbol{f_Y}(\boldsymbol{f_I}(\cdots \boldsymbol{f_2}(\boldsymbol{f_1}(\boldsymbol{x}))\cdots))$, i.e., $E_{\mathrm{DNN}}(\cdot)$ is determined by the entire architecture of DNNs. In addition, since $P(\boldsymbol{F_Y}|\boldsymbol{X})$ is a conditional distribution of $\boldsymbol{F_Y}$ given $\boldsymbol{X}$, the partition function $Z_{\mathrm{DNN}}(\boldsymbol{x})$ should integrate all possible activations of the output layer $\boldsymbol{f_Y}$ to guarantee $P(\boldsymbol{F_Y}|\boldsymbol{X})$ being a valid conditional probability, i.e., $Z_{\mathrm{DNN}}(\boldsymbol{x}) = \int \exp[\boldsymbol{f_Y}(\boldsymbol{f_I}(\cdots \boldsymbol{f_2}(\boldsymbol{f_1}(\boldsymbol{x}))\cdots))]d\boldsymbol{f_Y}(\boldsymbol{f_I}(\cdots \boldsymbol{f_2}(\boldsymbol{f_1}(\boldsymbol{x}))\cdots))$.

For clarity, here we only consider integral no matter a Gibbs distribution is discrete or continuous, because $\sum_{\boldsymbol{f_1}\cdots \boldsymbol{f_I}} \exp[-E_{\mathrm{DNN}}(\boldsymbol{f_Y}|\boldsymbol{x})]$ is a special case of integral. All functions involving $\boldsymbol{f_i}$ should include parameters $\boldsymbol{\theta_{f_i}}$ in the conditioning set, but we omit for the same reason.

## J.2 SGD CAN BE EXPLAINED AS A VARIATIONAL INFERENCE ALGORITHM

As a dominant paradigm for Bayesian posterior inference $P(H|E) \propto P(E|H)P(H)$, the variational inference converts the inference problem into an optimization problem, where the prior distribution $P(H)$ is the probability of arbitrary hypothesis $H$ with respect to the observation $E$, and $P(E|H)$ is the likelihood distribution. Specifically, the variational inference posits a family of approximate distributions $\mathcal{Q}$ and aims to find a distribution $Q^*(H)$ that minimizes the Kullback-Leibler (KL) divergence between $P(H|E)$ and $Q(H)$ (Blei et al., 2017).

$$Q^*(H) = \arg\min_{Q \in \mathcal{Q}} \text{KL}(P(H|E)\|Q(H)) \tag{85}$$

Notably, recent works demonstrate that SGD performs variational inference during training DNNs (Mandt et al., 2017; Chaudhari & Soatto, 2018). Following these works, we further confirm the Bayesian explanation for SGD based on the proposed probabilistic for DNNs.

Given a DNN $= \{\boldsymbol{x}; \boldsymbol{f_1}; ...; \boldsymbol{f_I}; \boldsymbol{f_Y}\}$, we choose the cross entropy $H[P(\boldsymbol{Y}|\boldsymbol{X}), P(\boldsymbol{F_Y}|\boldsymbol{X})]$ as the loss function, where $P(\boldsymbol{Y}|\boldsymbol{X})$ is the true posterior distribution determined by the training dataset $\mathcal{D}$ and $P(\boldsymbol{F_Y}|\boldsymbol{X})$ is the conditional distribution of $\boldsymbol{f_Y}$ given $\boldsymbol{x}$ derived from the DNN.

$$H[P(\boldsymbol{Y}|\boldsymbol{X}), P(\boldsymbol{F_Y}|\boldsymbol{X})] = H[P(\boldsymbol{Y}|\boldsymbol{X})] + \text{KL}[P(\boldsymbol{Y}|\boldsymbol{X}) \parallel P(\boldsymbol{F_Y}|\boldsymbol{X})]$$
$$= -\sum_{l=1}^{L} P[\boldsymbol{y}(l)|\boldsymbol{x}]log[P(\boldsymbol{f_Y}(l)|\boldsymbol{x})] \tag{86}$$

where $H[P(\boldsymbol{Y}|\boldsymbol{X})]$ is the entropy of $P(\boldsymbol{Y}|\boldsymbol{X})$ and $P(\boldsymbol{y}(l)|\boldsymbol{x})$ is the true posterior distribution given the training data $\boldsymbol{x}$. For example, if $L = 4$ and $P(\boldsymbol{y}|\boldsymbol{x})$ is $[0, 1, 0, 0]$, so $P[\boldsymbol{y}(2)|\boldsymbol{x}] = 1$ and $P[\boldsymbol{y}(3)|\boldsymbol{x}] = 0$. $P(\boldsymbol{f_Y}|\boldsymbol{x})$ is the learned conditional distribution of $\boldsymbol{f_Y}$ given the same $\boldsymbol{x}$, e.g., $P(\boldsymbol{f_Y}|\boldsymbol{x})$ could be $[0.2, 0.7, 0.1, 0]$.

In the context of deep learning, SGD aims to find the $P^*(\boldsymbol{F_Y}|\boldsymbol{X})$ such that it can minimize the loss function $H[P(\boldsymbol{Y}|\boldsymbol{X}), P(\boldsymbol{F_Y}|\boldsymbol{X})]$ given the training dataset $\mathcal{D}$.

$$P^*(\boldsymbol{F_Y}|\boldsymbol{X}) = \arg\min_{P(\boldsymbol{F_Y}|\boldsymbol{X}) \in \mathcal{Q}} H[P(\boldsymbol{Y}|\boldsymbol{X}), P(\boldsymbol{F_Y}|\boldsymbol{X})] \tag{87}$$

Given the training dataset $\mathcal{D}$, $P(\boldsymbol{Y}|\boldsymbol{X})$ is a constant, i.e., $H[P(\boldsymbol{Y}|\boldsymbol{X})] = 0$, thus we can obtain

$$P^*(\boldsymbol{F_Y}|\boldsymbol{X}) = \arg\min_{P(\boldsymbol{F_Y}|\boldsymbol{X}) \in \mathcal{Q}} \text{KL}[P(\boldsymbol{Y}|\boldsymbol{X}) \parallel P(\boldsymbol{F_Y}|\boldsymbol{X})] \tag{88}$$

Comparing the Equation 88 and 85, we can confirm the equivalence between SGD and variational inference based on the proposed probabilistic explanations for DNNs. More specifically, the DNN defines a family of conditional Gibbs distribution $P(\boldsymbol{F_Y}|\boldsymbol{X})$ (Equation 84) and SGD aims to find $P^*(\boldsymbol{F_Y}|\boldsymbol{X})$ such that it can minimize the distance to the true posterior distribution $P(\boldsymbol{Y}|\boldsymbol{X})$.

## J.3 SGD IS EQUIVALENT TO THE ENERGY MINIMIZATION OPTIMIZATION

In most deep learning applications, $P(\boldsymbol{y}|\boldsymbol{x})$ is commonly an one-hot vector, i.e., $P(\boldsymbol{y}|\boldsymbol{x})$ only has a single non-zero element. If we assume that $l^*$ is the index of the non-zero element of $P(\boldsymbol{y}|\boldsymbol{x})$, i.e., $P[\boldsymbol{y}(l^*)|\boldsymbol{x}] = 1$, we can simplify $\text{KL}[P(\boldsymbol{Y}|\boldsymbol{X}) \parallel P(\boldsymbol{F_Y}|\boldsymbol{X})] = -log[P(\boldsymbol{f_Y}(l^*)|\boldsymbol{x})]$.

Based on the definition of $P(\boldsymbol{F_Y}|\boldsymbol{X})$ (Equation 84), we can obtain

$$\text{KL}[P(\boldsymbol{Y}|\boldsymbol{X}) \parallel P(\boldsymbol{F_Y}|\boldsymbol{X})] = -log[P(\boldsymbol{f_Y}(l^*)|\boldsymbol{x})]$$
$$= -f_{yl^*}(\boldsymbol{f_I}(\cdots \boldsymbol{f_2}(\boldsymbol{f_1}(\boldsymbol{x})) \cdots)) + log[Z_{\text{DNN}}(\boldsymbol{x})] \tag{89}$$
$$= E_{DNN}(f_{yl^*}|\boldsymbol{x}) + log[Z_{\text{DNN}}(\boldsymbol{x})]$$

where $f_{yl^*}$ is the $l^*$th output node of the output layer $\boldsymbol{f_Y}$. In particular, since $log[Z_{\text{DNN}}(\boldsymbol{x})]$ is not related to the output $\boldsymbol{f_Y}$, it can be viewed as a constant with respect to the gradient calculation indicated by Table 8. Finally, combining Equation 88 and 89, we can obtain

$$P^*(\boldsymbol{F_Y}|\boldsymbol{X}) = \arg\min_{P(\boldsymbol{F_Y}|\boldsymbol{X}) \in \mathcal{Q}} E_{DNN}(f_{yl^*}|\boldsymbol{x}) \tag{90}$$

The above equation indicates that SGD is equivalent to the energy minimization optimization, which in turn validates the Gibbs explanation for the hidden layer of DNNs.

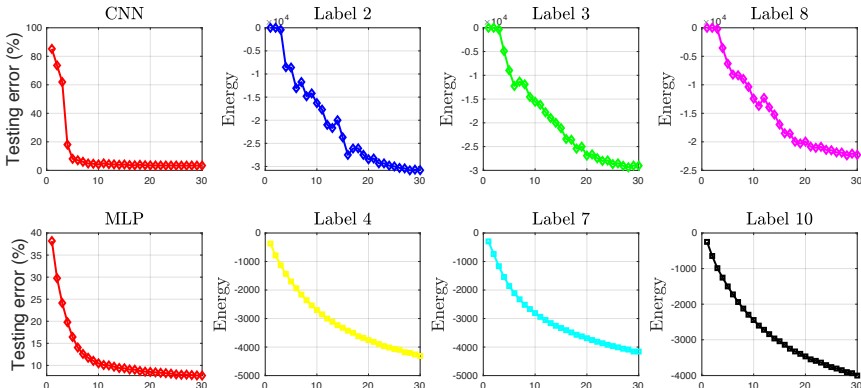

Figure 23: The first row shows the testing error of the CNN and the variation of $E_{CNN}(f_{yl^*}|\boldsymbol{x})$ given three testing datasets with different labels over 30 training epochs. The second row shows testing error of the MLP and the variation of $E_{MLP}(f_{yl^*}|\boldsymbol{x})$ given three testing datasets with different labels over 30 training epochs.

# K  EXPERIMENTS VALIDATING THE THEORETICAL EXPLANATION FOR SGD

## K.1  EXPERIMENT: SGD IS EQUIVALENT TO THE ENERGY MINIMIZATION OPTIMIZATION

To validate the proof that SGD learns a $P(\boldsymbol{F_Y}|\boldsymbol{X})$ with the minimal energy from the family of Gibbs distribution determined by a DNN, we use SGD to train the CNN defined in Table 9 for classifying the synthetic dataset shown in Figure 5. Specifically, we use the entire training dataset to train the CNN over 30 epochs. During each training epoch, we calculate the energy function of $P(\boldsymbol{F_Y}|\boldsymbol{X})$ given a testing dataset with the same label. Since the testing dataset has the same label, we have the same index of the single non-zero element of $P[\boldsymbol{y}|\boldsymbol{x}]$. As a result, we can easily examine if the corresponding energy function $E_{CNN}(f_{yl^*}|\boldsymbol{x})$ is decreasing during the training procedure.

Based on the architecture of CNN in Table 9, $E_{CNN}(f_{yl^*}|\boldsymbol{x})$ can be formulated as

$$E_{CNN}(f_{yl^*}|\boldsymbol{x}) = -f_{yl^*}(\boldsymbol{f_4}(\boldsymbol{f_3}(\boldsymbol{f_2}(\boldsymbol{f_1}(\boldsymbol{x}))))) \tag{91}$$

In Figure 23, the first row visualizes the variation of $E_{CNN}(f_{yl^*}|\boldsymbol{x})$ over 30 training epochs. Though there are certain fluctuations, $E_{CNN}(f_{yl^*}|\boldsymbol{x})$ indeed shows a decreasing trend. Due to limited space, we only visualize the energy minimization procedure given three labels, but the procedure is valid for any label. In addition, we demonstrate the equivalence based on the MLP (Table 3) for classifying the MNIST dataset. The energy function of $P(\boldsymbol{F_Y}|\boldsymbol{X})$ corresponding to the MLP is defined as

$$E_{MLP}(f_{yl^*}|\boldsymbol{x}) = -f_{yl^*}(\boldsymbol{f_2}(\boldsymbol{f_1}(\boldsymbol{x}))) \tag{92}$$

In Figure 23, the second row shows the same energy decreasing trend during training the MLP. Overall, we demonstrate the equivalence between SGD and the energy minimization optimization, which in turn validates the Gibbs explanation for DNNs.

## K.2  EXPERIMENT: THE PARAMETERS LEARNED BY SGD ARE NOT I.I.D.

To validate the parameters learned by SGD not i.i.d., we use SGD to train the MLP defined in Table 3 and the CNN defined in Table 5. Following the commonly used initialization approach, we initialize all the weights of the two DNNs by Gaussian random numbers, i.e., all the parameters can be viewed as i.i.d. before training. We use the absolute value of the sample correlation defined by Equation 2 to quantify the dependence between different weights during the training procedure.

Figure 24 visualizes the the variation of the average correlation between different weights of the three layers in the MLP. Initially, the average correlation for $\boldsymbol{f_1}$ and $\boldsymbol{f_2}$ are very small, because parameters are Gaussian random values. As the training procedure continues, we find that all the correlation values are increasing. Notably, the increase for $\boldsymbol{f_2}$ is huge, i.e., the weights of $\boldsymbol{f_2}$ cannot be i.i.d. anymore. However, the increase for $\boldsymbol{f_1}$ is very small, i.e., all the weights of $\boldsymbol{f_1}$ still could be i.i.d.. Therefore, not all the learned parameters in the MLP are i.i.d..

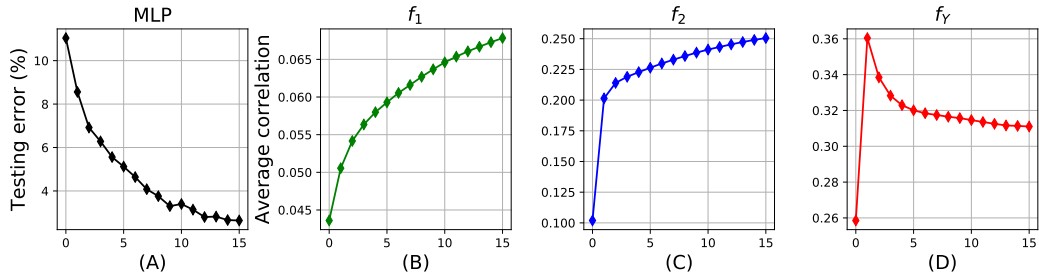

Figure 24: (A) shows the testing error over 15 training epochs. (B), (C), and (D) show the average correlation between different weights of three layers in the MLP over 15 training epochs.

Figure 25 visualizes the variation of the average correlation between different weights of the five layers in the CNN. Similar as the weights in the MLP, the average correlation for $f_1$ and $f_2$ initially are very small, because they are initialized as Gaussian random values. As the training procedure continues, they have a sharp increasing and quickly achieve stationary, thus the parameters of $f_1$ and $f_2$ are not i.i.d. anymore. In addition, the average correlation for $f_3$ and $f_4$ also have an increasing trend, but their increase are very small, i.e., the parameters of $f_3$ and $f_4$ still could be i.i.d.. Overall, we conclude that not all the learned parameters in the CNN are i.i.d..

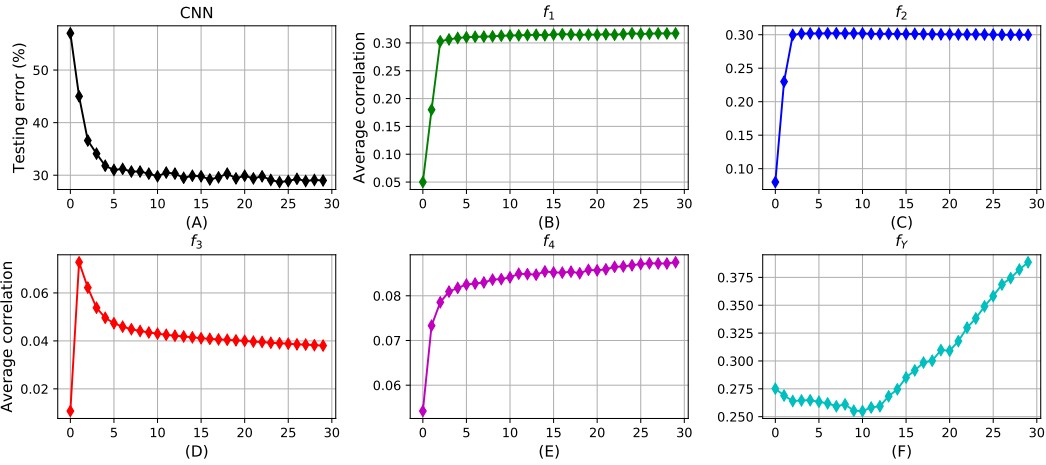

Figure 25: (A) shows the testing error over 30 training epochs. (B), (C), (D), (E) and (F) show the average correlation between different weights of five layers in the CNN over 30 training epochs.

### K.3 EXPERIMENT: A BETTER PRIOR DISTRIBUTION INDICATES A BETTER GENERALIZATION PERFORMANCE

We design three CNNs $= \{x; f_1; f_2; f_3; f_4; f_5; f_Y\}$ in Table 10 to classify the synthetic dataset shown in Figure 5. We use the entire training dataset to train the CNNs over 30 epochs and use the entire testing dataset to examine the generalization performance of the CNNs.

Based on the MRF explanation for convolutional layers, the first convolutional layer $f_1$ formulates a MRF model as the prior distribution $P(F_1)$, which can be expressed as

$$P(F_1) = \frac{1}{Z_{F_1}} \prod_{k=1}^{K} \exp[\varphi_1^k(x)] = \frac{1}{Z_{F_1}} \exp[\sum_{k=1}^{K} \varphi_1^k(x)] \tag{93}$$

where $\varphi_1^k(x) = S^k \circ x + b_1^k \cdot 1$ is the $k$th linear convolutional channel, $S^k$ is the linear convolutional filter and $b_1^k$ is the bias.

Table 10: The architectures of CNNs for classifying the synthetic dataset

| R.V. | Layer | Description | CNN1 | CNN2 | CNN3 |
|------|-------|-------------|------|------|------|
| $X$ | $x$ | Input | $32 \times 32 \times 1$ | $32 \times 32 \times 1$ | $32 \times 32 \times 1$ |
| $F_1$ | $f_1$ | Conv $(3 \times 3)$ +ReLU | $30 \times 30 \times 4$ | $30 \times 30 \times 12$ | $30 \times 30 \times 36$ |
|  | $f_2$ | Maxpool | $15 \times 15 \times 4$ | $15 \times 15 \times 12$ | $15 \times 15 \times 36$ |
| $F_2$ | $f_3$ | Conv $(5 \times 5)$ + ReLU | $11 \times 11 \times 20$ | $11 \times 11 \times 20$ | $11 \times 11 \times 20$ |
|  | $f_4$ | Maxpool | $5 \times 5 \times 20$ | $5 \times 5 \times 20$ | $5 \times 5 \times 20$ |
| $F_3$ | $f_5$ | Fully connected | $1 \times 1 \times 20$ | $1 \times 1 \times 20$ | $1 \times 1 \times 20$ |
| $F_Y$ | $f_Y$ | Output(softmax) | $1 \times 1 \times 10$ | $1 \times 1 \times 10$ | $1 \times 1 \times 10$ |

R.V. is the random variable of the hidden layer(s).

Since the synthetic dataset obeys the Gaussian distribution $P(\boldsymbol{X})$, we can measure the quality of the prior distribution by averaging $\mathrm{KLD}[P(\boldsymbol{X})\|P(\boldsymbol{F_1})]$ over all the testing images for every training epoch. In addition, we use the testing error to measure the generalization performance. Also note that the only difference between the tree CNNs is they have different $K$ numbers of convolutional channels in $\boldsymbol{f_1}$, i.e., different CNNs have different prior distributions. Therefore, we can examine the effect of the prior distribution by checking the generalization performance of the three CNNs.

Figure 26 shows the learned prior distributions and the corresponding testing error over 30 training epochs. We can find that there is a positive correlation between $\mathrm{KLD}[P(\boldsymbol{X})\|P(\boldsymbol{F_1})]$ and the testing error. Specifically, the testing error decreases as the learned prior distribution $P(\boldsymbol{F_1})$ approaching to the truly prior distribution $P(\boldsymbol{X})$, and the testing error becomes stable when $P(\boldsymbol{F_1})$ achieves stationary for all the three CNNs.

Moreover, we find that the CNN with more convolutional channels achieves the better generalization performance. In terms of Bayesian theory, that is because a convolutional layer with more convolutional channels can formulate a more powerful prior distribution, such that it can convey more prior knowledge of the dataset, thereby rendering better generalization performance. Overall, we justify that a better prior distribution indicates a better generalization performance.

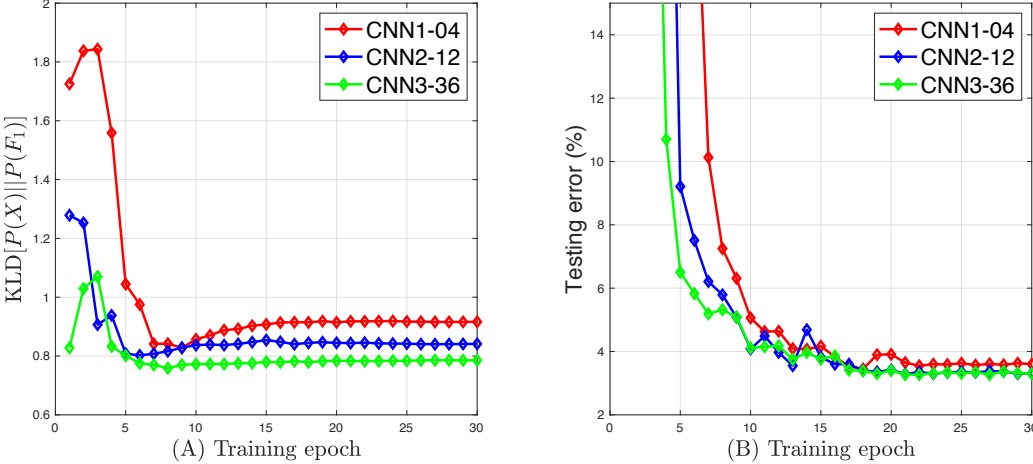

Figure 26: (A) shows the average $\mathrm{KL}[P(\boldsymbol{X})\|P(\boldsymbol{F_1})]$ overall all testing images. (B) shows the generalization performance of the three CNNs represented by the testing error. The number behind the three CNNs in the legend denotes the number of convolutional channels of $\boldsymbol{f_1}$.

## L    A REGULARIZATION ALGORITHM BASED ON THE PROBABILISTIC REPRESENTATION

### L.1    THE PROPOSED REGULARIZATION ALGORITHM

Based on the probabilistic representation, the first hidden layer formulates a MRF model as a prior distribution of the training dataset and other hidden layers recursively serve as prior distributions for their respective next layer in a CNN. However, the backpropagation calculates the gradient in the backward direction, i.e., updating the gradient of a hidden layer corresponding to a prior distribution must go over all hidden layers behind it. In addition, the degradation problem could makes the learned prior distribution not precise enough to regularize DNNs and results in overfitting, especially in very deep neural networks (He et al., 2016).

In this section, we propose to a new regularization learning algorithm for DNNs, which includes two steps: (i) we learn the prior distributions of DNNs directly from the training dataset $\{\boldsymbol{x}^j\}_{j=1}^J$ as the blue arrow shown in Figure 27 and (ii) we use the backpropagation to train the initialized DNNs by the learned prior distributions as the red arrow shown in Figure 27.

Since directly learning prior distribution from $\{\boldsymbol{x}^j\}_{j=1}^J$ is less complicated than the backpropagation learning algorithm, we can relieve the effects of degradation problem and derive prior distributions more precisely, thereby achieving the better generalization performance of DNNs.

An important question need to answer is how many hidden layers of DNNs need to be pre-trained in the first step. Since most prior information is stored in the training dataset $\{\boldsymbol{x}_j\}_{j=1}^J$, the hidden layer closest to the input of DNNs, i.e., the first hidden layer $\boldsymbol{f_1}$, should learn the most prior information from $\{\boldsymbol{x}^j\}_{j=1}^J$ as long as $\boldsymbol{f_1}$ has powerful representation ability through including sufficient hidden units, e.g., convolutional filters. Therefore, the proposed regularization method only pre-train the first hidden layer $\boldsymbol{f_1}$ in order to decrease the computation complexity. Pre-training more hidden layers could obtain better generalization performance, which can be a direction for future research.

We choose the Field of Experts (FoE) model to directly learn a prior distribution of $\{\boldsymbol{x}^j\}_{j=1}^J$ because of two reasons. Above all, FoE achieves great performance on learning prior knowledge of given datasets on various applications, e.g., image denoising (Roth & Black, 2005), image restoration Roth & Black (2008), and image inpainting (Schmidt et al., 2011). More importantly, FoE can be regarded as a Convolutional Neural Networks (CNN) with a single convolutional layer, hence we can easily use the learned FoE model to initialize the first hidden layer of CNNs as long as they have the same architecture. A FoE model can be formulated as

$$p_{\text{FoE}}(\boldsymbol{x}) = \frac{1}{Z_{\text{FoE}}} \prod_{k=1}^K \{f_k^{\text{expert}}[f_k(\boldsymbol{x})]\}, \tag{94}$$

where $f_k(\cdot)$ is a linear filter, $f_k^{\text{expert}}$ is an expert function, and $Z_{\text{FoE}} = \int_{\boldsymbol{x}} \prod_{k=1}^K \{f_k^{\text{expert}}[f_k(\boldsymbol{x})]\} d\boldsymbol{x}$ is the partition function. We use the contrastive divergence learning method to train $p_{\text{FoE}}(\boldsymbol{x})$ under the Kullback-Leibler divergence (KLD) criterion (Carreira-Perpinan & Hinton, 2005). More detail of the training procedure is in (Roth & Black, 2005; 2008). After deriving a well-trained FoE model, we use the learned linear filters $\{f_k\}_{k=1}^K$ to initialize the first convolutional layer of CNNs. Finally, we use the backpropagation algorithm to train the initialized CNN for a specific application, such as image recognition. We summarize the proposed regularization learning algorithm in Algorithm 2.

$$\{\boldsymbol{x}^j\}_{j=1}^J \quad P(\boldsymbol{F_1}) \cdot \cdots P(\boldsymbol{F_{i+1}}|\boldsymbol{F_i}) \cdots P(\boldsymbol{F_Y}|\boldsymbol{F_I}) \quad \{\boldsymbol{y}^j\}_{j=1}^J$$

Figure 27: The training procedure of a DNN $= \{\boldsymbol{x}; \boldsymbol{f_1}; \cdots; \boldsymbol{f_Y}\}$. The red arrow indicates the traditional training algorithm, e.g. backpropagaton, which derives an output of the DNN given training dataset $\{\boldsymbol{x}^j\}_{j=1}^J$ in the forward direction and calculate the gradient of parameters with respect to the training labels $\boldsymbol{Y}$ for updating the parameters in the backward direction.

---

**Algorithm 2** Regularization learning algorithm

---

**input:** $\mathcal{D} = \{(\boldsymbol{x}_n, \boldsymbol{y}_n)\}_{n=1}^N$, DNN $= \{\boldsymbol{x}; \boldsymbol{f_1}; ...; \boldsymbol{f_I}; \boldsymbol{f_Y}\}$, and $p_{\text{FoE}}(\boldsymbol{x})$
  1: **setup**
  2:     initialize iteration $k = 0$, and training epochs $K$
  3:     specify the architecture of the DNN and $p_{\text{FoE}}(\boldsymbol{x})$
  4: **regularization learning**
  5:     train $p_{\text{FoE}}(\boldsymbol{x})$ given $\{\boldsymbol{x}_n\}_{n=1}^N$
  6:     initialize $\boldsymbol{f_1}$ by the well-trained $p_{\text{FoE}}(\boldsymbol{x})$
  7: **repeat**
  8:     train the DNN given $\mathcal{D}$
  9: **until** $(k = K)$
**output:** the well-trained DNN for image recognition

---

### L.2 SIMULATIONS

In this section, we validate the proposed regularization learning algorithm for image recognition application on the CIFAR-10 benchmark dataset, which includes 70,000 $32 \times 32$ color images in 10 classes, and each class has 6,000 training images and 1,000 testing images. Specifically, we randomly choose 2,000 training images of each class as a new training dataset and use the same testing dataset. Smaller training dataset means overfitting being more likely to occur, which can help us validate the proposed regularization learning method more convincingly. In addition, we covert color images to grayscale to simplify computation.

In order to directly learn regularization from $\{\boldsymbol{x}^j\}_{j=1}^{20000}$, we design a FoE model with 12 Gaussian Scale Mixture (GSM) experts, thus it can be expressed as

$$p_{\text{FoE}}(\boldsymbol{x}) = \frac{1}{Z_{\text{FoE}}} \prod_{k=1}^{12} \{f_k^{\text{GSM}}[f_k(\boldsymbol{x})]\}, \tag{95}$$

where $f_k^{\text{GSM}}[f_k(\boldsymbol{x})] = \sum_{i=1}^{11} \pi_{ki} \cdot \mathcal{N}(f_k(\boldsymbol{x}); 0, \frac{\sigma_b^2}{\delta(i)})$, $\sigma_b^2$ is a fixed base variance, $\boldsymbol{\delta} = \{\delta(1), \cdots, \delta(11)\}$ is a range of scales, $f_k(\cdot)$ is a $3 \times 3$ convolutional filter, and $\pi_{ki}$ denotes the weight of each Gaussian distribution. The parameters of the FoE model is $\boldsymbol{\theta} = \{\boldsymbol{f}, \boldsymbol{\pi}\}$, where $\boldsymbol{f} = [f_1, ..., f_{12}]$ and $\boldsymbol{\pi} = \{\pi_{ki}\}_{k=1,i=1}^{k=12,i=11}$. The architecture of the CNN for image recognition is summarized in Table 11. It is noteworthy that the number of convolutional filters in $\boldsymbol{f_1}$ and the filter dimension are the same as the above FoE model, thus we can use the well-trained FoE to initialize $\boldsymbol{f_1}$ as a regularization for the CNN classifying the CIFAR-10 dataset based on Algorithm 2.

In order to make a comprehensive comparison, we choose two commonly used regularizations, i.e., the dropout (drop rate is 0.3) and the $L2$ norm, as references. We use the testing error to exam the generalization performances of different regularizations. Figure 28 visualizes that the proposed regularization method (abbr. Bayes Prior (BP)) successfully decreases the testing error. In particular, it outperforms dropout (32.26%) and achieve similar performance as $L2$ norm (29.58%). Therefore, this experiment validates the effectiveness of the proposed regularization learning method.

Table 11: The architectures of CNN for CIFAR-10 classification

| Layer | Description | Output |
|:---:|:---:|:---:|
| $\boldsymbol{x}$ | Input | $32 \times 32 \times 1$ |
| $\boldsymbol{f_1}$ | Conv ($3 \times 3$) | $30 \times 30 \times 12$ |
| $\boldsymbol{f_2}$ | ReLU | $30 \times 30 \times 12$ |
| $\boldsymbol{f_3}$ | Conv ($5 \times 5$) | $26 \times 26 \times 32$ |
| $\boldsymbol{f_4}$ | ReLU + Maxpool | $13 \times 13 \times 32$ |
| $\boldsymbol{f_5}$ | Conv ($5 \times 5$) | $9 \times 9 \times 128$ |
| $\boldsymbol{f_6}$ | ReLU + Maxpool | $4 \times 4 \times 128$ |
| $\boldsymbol{f_7}$ | Fully connected | $1 \times 1 \times 512$ |
| $\boldsymbol{f_8}$ | ReLU | $1 \times 1 \times 512$ |
| $\boldsymbol{f_9}$ | Fully connected | $1 \times 1 \times 32$ |
| $\boldsymbol{f_{10}}$ | ReLU | $1 \times 1 \times 32$ |
| $\boldsymbol{f_Y}$ | Output(softmax) | $1 \times 1 \times 10$ |

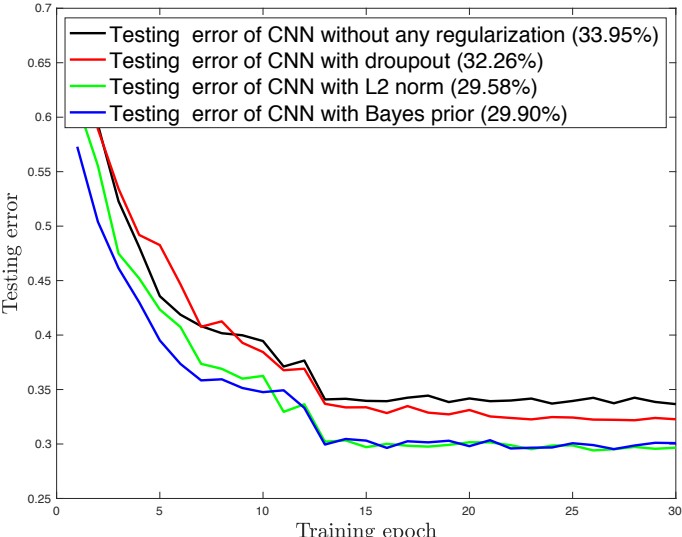

Figure 28: The average testing error of the CNN on the last 5 training epochs based on various regularizations. Bayes prior indicates the proposed regularization and its testing error is 29.90%.

Moreover, the proposed regularization method can be easily combined with other regularizations methods to improve the generalization performance further. We combine two arbitrary aforementioned regularization methods together and examine their generalization performances under the same experimental conditions as above. Figure 29 shows that the combination of two regularizations together achieves better generalization performance than any single one. In particular, the combination of BP and L2 norm achieves the lowest testing error (28.60%). Also note that if we combine three regularizations together, we achieve the best generalization performance (27.69%). This experiment further validate the proposed regularization learning algorithm.

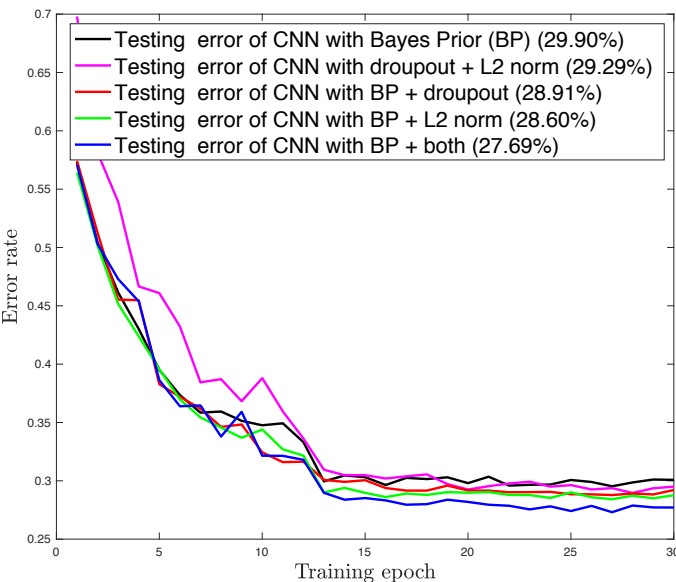

Figure 29: The average testing error of the CNN on the last 5 training epochs based on various combined regularizations. Bayes prior + both indicates that we combine all three regularizations together.

