# OpenReview forum: "Probabilistic modeling the hidden layers of deep neural networks"
_ICLR.cc/2020/Conference — Reject_

### Official Review · AnonReviewer2 · 2019-10-18
**Official Blind Review #1**

**Rating:** 6

**Review:**

Summary of the Paper:

The authors claim the that i.i.d hypothesis that is often used in the prior when looking for the equivalence between neural networks and GP is not valid. Then, they propose a new interpretation of neural networks as Gibbs distributions (in the case of fully connected layers) and a MRF in the case of convolutional layers. Some simulations are done to verify this.


Detailed comments:

Overall I believe that the writing of the paper is very sloppy and difficult to read and follow. It is not clear that the GP interpretation cannot be valid simply because the activations and weights are not i.i.d. I.I.D is a sufficient condition but not required in the central limit theorem. For example, the sum of correlated Gaussian variables also tends to a Gaussian distribution. The same for non-Gaussian variables. Therefore, I do not think that the GP interpretation of the NN is wrong simply for that. This simply shows that the i.i.d. prior may be suboptimal.

Summing up, I think that this paper needs more work. Currently, I do not think I can extract anything useful from it.

The prior is subjective and can be chosen by the user. So if an i.i.d. prior is actually chosen, the corresponding Bayesian  neural network will converge to a GP.

An i.i.d. prior may be sub-optimal. However, it can be used to interpret neural networks as GP. There is no problem with that.

It is well known that the sum of random variables can also converge to a Gaussian distribution even though they are not independent. This questions the claims of the paper.

The witting of the paper needs to be improved. There are several expressions that do not sound well. E.g., ", GP with i.i.d."

Z'y in Eq. (9) should depend on l.

It is not clear what is the distribution of a hidden layer (activations weights etc..).

"...and f Y is an estimation of the true distribution P (Y |X)" what do you mean by that?

Eq. (8) seems to be a prob. distribution for the random variable Fy. However, the authors give an expression for f_yl, which does not make sense.


**Experience Assessment:**

I do not know much about this area.

**Review Assessment: Checking Correctness Of Derivations And Theory:**

I did not assess the derivations or theory.

**Review Assessment: Checking Correctness Of Experiments:**

I did not assess the experiments.

**Review Assessment: Thoroughness In Paper Reading:**

I made a quick assessment of this paper.

---

> ### Author Response · Authors · 2019-11-10
> **Response to Review #2**
>
> Thank you for your detailed comments! We post a single comment ‘Why should we care about the i.i.d. assumption?’. We hope that can clarify our idea.
>
> Below are our replies to your specific comments.
>
> >>>I.I.D is a sufficient condition but not required in the central limit theorem. For example, the sum of correlated Gaussian variables also tends to a Gaussian distribution. The same for non-Gaussian variables. Therefore, I do not think that the GP interpretation of the NN is wrong simply for that.
>
> As we presented in the comment ‘Why should we care about the i.i.d. assumption’, we can find that I.I.D. is the prerequisite of CLT.
>
> We agree that the sum of correlated Gaussian variables tends to be Gaussian. However, that is not valid for non-Gaussian variables unless they are i.i.d. In other words, if we want to use CLT to derive a Gaussian distribution, we must guarantee all the random variables are i.i.d., otherwise, the result derived from CLT is not correct.
>
> >>>It is well known that the sum of random variables can also converge to a Gaussian distribution even though they are not independent. This questions the claims of the paper.
>
> To the best of our knowledge, only the summation of I.I.D. random variables converges to a Gaussian distribution. The Lyapunov CLT could relax the I.I.D premise to be independence only under certain conditions.
>
> >>>The prior is subjective and can be chosen by the user. So if an i.i.d. prior is actually chosen, the corresponding Bayesian neural network will converge to a GP. An i.i.d. prior may be sub-optimal. However, it can be used to interpret neural networks as GP. There is no problem with that.
>
> I agree that the prior is subjective. We have the freedom to choose any prior. However, if we want the posterior distribution derived from the chosen prior to precisely explain something of interest, we must choose a prior distribution that conveys correct prior knowledge. For example, assuming some dataset obeys a Laplace distribution, it would be very hard to use a Gaussian distribution to precisely model the dataset, right?
>
> Back to the GP explanation for DNNs, when we find GP cannot explain some important properties of DNNs, we should consider if he GP explanation is reasonable or correct and try to propose a better explanation. Are you agree?
>
> >>>There are several expressions that do not sound well. E.g., ", GP with i.i.d."
>
> GP with i.i.d. prior means one use GP to explain the distribution of hidden layers and assume all the parameters of DNNs obey i.i.d. Gaussian prior distribution. We already refine the paper to make it clear. Thank you!
>
> >>>Z'y in Eq. (9) should depend on l. It is not clear what is the distribution of a hidden layer (activations weights etc..).
>
> Z_Y is not dependent on the index l. As we written in the paper, Z_Y = \sum_{l=1}^L exp(f_yl) is the partition function to guarantee P(F_Y) is a valid probability measure.
>
> >>>"...and f Y is an estimation of the true distribution P (Y |X)" what do you mean by that?
>
> The previous work [1] proves that the output of DNNs estimates a Bayesian posterior distribution. Following this explanation, f_Y is the output of DNNs and P(Y|X) is the true posterior distribution determined by the training dataset.
> We explain all the notations in the preliminaries section.
>
> [1] M.D. Richard and R.P. Lippmann. Neural network classifiers estimate Bayesian a posteriori probabilities. Neural Computation, pages 461–483, 1991.
>
> >>>Eq. (8) seems to be a prob. distribution for the random variable Fy. However, the authors give an expression for f_yl, which does not make sense.
>
> Since the output layer f_Y has L nodes, i.e., f_Y = {f_y1, f_y2, …, f_yL}. As a result, F_Y is a random vector with L random variables corresponding to the L output nodes.
>
> We appreciate your detailed comments. We have already refined our paper based on your comments. Please let us know if you have more questions about our paper. Thank you!

---

### Official Review · AnonReviewer3 · 2019-10-23
**Official Blind Review #27**

**Rating:** 6

**Review:**


Main contribution of the paper
- The paper argues that the base assumption, the i.i.d. of the activated elements (activations) in the hidden layers, the existing methods (lee.et.al 2018) hold is not convincible.
- Instead, the author proposes a new way to probabilistically model the hidden layers, activations, and layer/layer connections.
- Based on the probabilistic model, the paper proposes a new regularizer.

Methods
- The author argues that the activation is not iid by empirically showing that the trained MLP (in most cases) does not un-correlated.
- The author proposes a new probabilistic model for MLP, and CNN assuming the Gibbs distribution to each activation and also assuming the product of expert (poE) model to explain the layer/layer relationship.
- And according to their model, CNN will be explained by the MRF model.
- The author proposes a regularization term regarding layer/layer connection.
- They argue that the SGD training can be seen as a first-order approximation of the inference of the hidden activations in MLP.

Questions
- See the Concerns

Strongpoints
- The probabilistic explanation of the MLP and the CNN seems novel and was interesting to the reviewer
- The proposed explanation assumes a weaker condition compared to the existing methods.

Concerns
- The main concern is that the reviewer cannot fully convince that i.i.d. assumption is wrong.
Even though the trained MLP does not support the i.i.d. condition, one can suppose that the reason would be the typical training method (SGD), just finding the local minima in a deterministic way.
Maybe the proof in Appendix.G. supports the argument of the author, but the reviewer failed to clearly agree with the argument.
A clear explanation regarding the issue would be required.
- As far as the author understands, the paper proposes a probabilistic (Bayesian) model for explaining MLP, but it seems that they just used SGD for training the model.
In that case, the reviewer is little suspicious of the role of the proposed regularization in that the regularization comes from Bayesian formulation, but the model was trained in a deterministic way.
The reviewer wants to ask the author that
(1) is it possible to infer the model in a Bayesian manner such as sampling?
(2) Is there any justification for using SGD when conducting the experiments regarding the regularization? If it is related to Appendix.G, clearer explanation would be appreciated.
- As far as the reviewer understands, the regularization deals with the practical part of the paper. It would be better to see the effect of the regularization of widely used networks such as small-layered ResNet or others.
If the proposed formulation has other practical strongpoints, it would be nice to clarify them.
- The explanation using Gibbs distribution and PoE looks similar to RBM. The reviewer strongly wants a clear explanation of the difference and the strongpoints compared to RBM.

Conclusion
- The author proposed a new probabilistic explanation of the neural network, which seems novel and worth reporting.
- However, the reviewer failed to fully agree on some steps in the process of the paper.
Therefore, the reviewer temporary rates the paper as weak-reject, but this can be adjusted after seeing the answers of the author.

Inquiries
- See the concerns parts.

**Experience Assessment:**

I have published one or two papers in this area.

**Review Assessment: Checking Correctness Of Derivations And Theory:**

I carefully checked the derivations and theory.

**Review Assessment: Checking Correctness Of Experiments:**

I assessed the sensibility of the experiments.

**Review Assessment: Thoroughness In Paper Reading:**

I read the paper thoroughly.

---

> ### Author Response · Authors · 2019-11-10
> **Response to Review #3**
>
> Thank you for your insightful and detailed comments.
>
> >>>>Above all, we understand your main concern that we cannot justify the Bayesian explanation for DNNs unless we prove the training procedure, i.e., SGD, works in a Bayesian manner as well. In order to answer this question well, we add Appendix J and Appendix K.1 to the paper.
>
> Briefly speaking, we prove that SGD can be explained as a Bayesian variational inference by three steps.
> *Step1, we prove the distribution of the output layer given the input, i.e., P(F_Y|X), is also a Gibbs distribution and its energy function is determined by the entire architecture of DNNs (Appendix J.1).
>
> *Step2, following previous works [1,2] proving the equivalence between SGD and variational inference, we further confirm the equivalence based on the proposed Gibbs explanation (Appendix J.2). In a word, SGD aims to find an optimal P*(F_Y|X) from a family of Gibbs distribution defined by DNNs, such that it can minimize the distance between P(Y|X) and P(F_Y|X), i.e.,
> P*(F_Y|X) = argmin KL[P(Y|X) || P(F_Y|X)]
>
> *Step3, we prove the equivalence between SGD and energy minimization optimization. Since the later is commonly used to optimize Gibbs distribution (a.k.a. the energy-based model), which in turn validates the Gibbs explanation for DNNs (Appendix J.3).
>
> In addition, new simulations on MLP and CNNs validate the equivalence between SGD and the energy minimization optimization (Appendix K.1). Overall, we propose probabilistic explanations for both the architecture of DNNs and the training procedure of DNNs. Therefore, we are safe to explain the i.i.d. assumption and the regularization in a statistical way.
>
> >>>>Below are our replies to your specific comments.
>
> >>>The main concern is that the reviewer cannot fully convince that i.i.d. assumption is wrong.
>
> We post a single comment ‘Why should we care about the i.i.d. assumption?’. Hope that can clarify our idea.  Please let us know if you have more questions about i.i.d.
>
> >>> is it possible to infer the model in a Bayesian manner such as sampling?
> Yes, we prove SGD is equivalent to the variational inference, which is a dominant paradigm for Bayesian posterior inference (Appendix J.2)
>
> >>> Is there any justification for using SGD when conducting the experiments regarding the regularization?
>
> Theoretically, a regularization corresponds to a prior distribution based on Bayesian theory [3].
> Experimentally, we demonstrate that SGD indeed learns a prior distribution and the prior distribution has a positive relation to the generalization performance, i.e., if SGD learns a better prior distribution, the testing error becomes smaller. (Appendix K.2)
> Do you agree with that answer?
>
> >>>It would be better to see the effect of the regularization of widely used networks such as small-layered ResNet or others. If the proposed formulation has other practical strongpoints, it would be nice to clarify them.
>
> Extending the regularization to ResNet is our research direction now. Different neural networks need to find different methods to pre-train the hidden layers corresponding to prior distributions, which requires lots of time. We will try our best to post the simulation results on ResNet in the final version. Since we explain DNNs in a Bayesian way, our research could be helpful to the neural architecture search based on Bayesian optimization.
>
> >>> The explanation using Gibbs distribution and PoE looks similar to RBM. The reviewer strongly wants a clear explanation of the difference and the strongpoints compared to RBM.
>
> Gibbs distribution is equivalent to Boltzmann distribution. Actually, they have different names in different backgrounds. In machine learning, Gibbs distribution is named as the energy-based model. In statistical physics, Gibbs distribution is named as renormalization group.
>
> RBM is a special case of Gibbs distribution. However, RBM only considers the connections between visible units x and hidden units h but overlooks the internal connections within x and h, thus it cannot explain complicated DNNs, e.g., CNNs. We refine the background section and provide more information about Gibbs distribution, the comparison between Gibbs and RBM, and previous works evolving deep learning and Gibbs distribution.
>
> Thanks again for your insightful comments! We have already refined the paper, please check the latest version. Hope we answer your questions correctly and look forward to your feedback again!
>
> [1] Stephan Mandt, Matthew D Hoffman, and David M Blei. Stochastic gradient descent as approximate Bayesian inference. The Journal of Machine Learning Research, 18(1):4873–4907, 2017.
> [2] Pratik Chaudhari and Stefano Soatto. Stochastic gradient descent performs variational inference, converges to limit cycles for deep networks. In 2018 Information Theory and Applications Workshop (ITA), pp. 1–10. IEEE, 2018.
> [3] Harald Steck and Tommi S. Jaakkola. On the dirichlet prior and Bayesian regularization. In NeurIPS, 2003.

---

> > ### Author Response · Authors · 2019-11-12
> > **More explanations for SGD and i.i.d. assumption**
> >
> > Dear reviewer,
> >
> > Thanks again for your comments. Do you think our answers resolve your concern?  Here we make more explanations for SGD and the i.i.d. assumption.
> >
> > First, we demonstrate that the parameters learned by SGD are not I.I.D. To validate that point, we use SGD to train the MLP defined in Table 3 and the CNN defined in Table 5. Following the commonly used initialization approach, we initialize the weights of the MLP by Gaussian random matrices, i.e., all parameters can be viewed as I.I.D.. before training.  We use the sample correlation define by Equation 2 to indicate the dependence between the different neurons. We visualize the variation of average correlation for each layer over all the training epochs.
> >
> > Our simulations show that the correlation value initially is very low, because we initialize the weights as Gaussian random numbers. However, as the training procedure continues, the correlation values of some hidden layers have a huge increase, and finally achieve a stationary status. It means that the learned parameters cannot keep the I.I.D. status during the whole training procedure. The detailed simulations are included in Appendix K.2, please check the new version of our paper.
> >
> > We think the I.I.D assumption contradicts the hierarchical property of DNNs. It is known that DNNs combine some low-level features together to derive high-level features [1]. Since high-level features are generated from low-level features, these high-level features should have certain correlations, which can be validated by the parameter's correlation. However, the I.I.D assumption contradicts the hierarchical property.
> > In particular, the previous work has already demonstrated that the Gaussian process is a flat representation [2].
> >
> > In addition, the reason we use SGD to train DNNs is that SGD is the most popular and fundamental algorithm to train DNNs. In other words, using SGD to train DNNs can help us to derive a general probabilistic conclusion about the I.I.D assumption.
> >
> > If you have questions about our paper, please feel free to let us know. Thank you!
> >
> > [1] LeCun, Y., Bengio, Y. & Hinton, G. Deep learning. Nature 521, 436–444 (2015)
> > [2] Alexander Matthews, Mark Rowland, Jiri Hron, Richard E. Turner, and Zoubin Ghahramani. Gaussian process behavior in wide deep neural networks. In ICLR, 2018.

---

> > ### Author Response · Authors · 2019-11-15
> > **Extra justification for the proposed regularization algorithm**
> >
> > Dear reviewer,
> >
> > Thanks again for your professional and helpful comments.
> > Here we show extra justification for the proposed regularization algorithm.
> >
> > >>>First, [1] demonstrates that the unsupervised pre-training is an effective regularization for DNNs through capturing the prior knowledge of the input. However, since they do not know which hidden layer corresponds to a prior distribution, they have to pre-training all the hidden layers, which restricts its wide application due to the computation complexity. Moreover, [2] proves that incorporating the prior knowledge of the input results in uniform stability and provides a bound on generalization error for a specific DNN, i.e., the supervised auto-encoders. The two previous work provides theoretical and empirical justification for the proposed regularization algorithm by pre-training the hidden layers serving as prior distributions.
> >
> > >>>Second, Appendix K.3 demonstrates that a better prior distribution indeed indicates a better generalization performance based on CNNs for classifying the synthetic dataset.
> >
> > We have already updated the paper, please check the latest version. If you have more questions, please let us know. Thanks again for your comments making our paper better!
> >
> > [1] Dumitru Erhan, Yoshua Bengio, Aaron Courville, Pierre-Antoine Manzagol, Pascal Vincent, and Samy Bengio. Why does unsupervised pre-training help deep learning? Journal of Machine Learning Research, 11(Feb):625–660, 2010.
> > [2]  Lei Le, Andrew Patterson, and Martha White. Supervised autoencoders: Improving generalization performance with unsupervised regularizers. In Advances in Neural Information Processing Systems, pp. 107–117, 2018.

---

### Official Review · AnonReviewer1 · 2019-10-24
**Official Blind Review #1**

**Rating:** 8

**Review:**

The authors show that parameters of a DNN do not satisfy the i.i.d. prior assumption and that neural layer activations considered as i.i.d. are not valid assumptions for all hidden layers of the network. One can therefore not rightfully use GPs to describe the network’s hidden layers. The authors suggest formulating the neurons per layer as energy functions thereby rendering a hidden layer as a Gibbs distribution and the connection between adjacent hidden layers as a PoE model.

The paper is well written and well postulated.

> Fig 4: What is the information presented by each neuron? How would this have looked with the i.i.d. prior in place.

> There are places in the paper where one must refer to the supplementary, for example sections H and J with the simulations. Do consider moving these crucial sections to the main paper.

> One recurring thought I had when the authors bring up Bayesian Hierarchical model, is that most of the BHMs rely on i.i.d assumptions both in the prior space and with the observations. How would you stand by your claim of explaining a DNN's layers to be modelled as a BHM?


**Experience Assessment:**

I have published one or two papers in this area.

**Review Assessment: Checking Correctness Of Derivations And Theory:**

I assessed the sensibility of the derivations and theory.

**Review Assessment: Checking Correctness Of Experiments:**

I assessed the sensibility of the experiments.

**Review Assessment: Thoroughness In Paper Reading:**

I read the paper at least twice and used my best judgement in assessing the paper.

---

> ### Author Response · Authors · 2019-11-10
> **Response to Review #1**
>
> Thank you for your encouraging review! Please find our replies to your specific comments below.
>
> >>> Fig 4: What is the information presented by each neuron? How would this have looked with the i.i.d. prior in place.
>
> Fig 4 visualizes the feature defined by the weights of each neuron in the fully connected hidden layer. Since the input is a vectorized image with dimension 1024×1, thus the dimension of the weights for each neuron is also 1024×1. In order to clearly show the feature defined by the weights of each neuron, we reshape the dimension of weights as 32×32 in Fig 4. Since the input image is sorted in the primary diagonal direction, the input feature can be described as high values above the secondary diagonal (the red line) and low values below the red line. Figure 4 shows that Neuron1, Neuron 2 and Neuron 4 correctly describe the input feature, thus they derive large positive outputs (i.e., g1(x) = 122.09, g2(x) = 113.70, and g4(x) = 140.20).  The probability of corresponding neurons, i.e., P (f1) = P (f2) = P (f4) = 0.20, indicates that the features defined by Neuron 1, Neuron 2 and Neuron 4 have high probability occurring in the input. (We have already refined the corresponding section to make it more clear)
>
> Previous works assume that the weights of neural networks are i.i.d., the experiment shows the i.i.d. assumption is not valid in two aspects. First, most weights in a single neuron are correlated to their neighbors, e.g., if a weight with high value at the top left position in Neuron 1, its neighbors have high probability be high value, and vice versa. Second, the weights in different neurons are correlated, e.g., the weights at the same position in Neuron 1 and Neuron 2 are positively correlated and the weights at the same position in Neuron 1 and Neuron 7 are negatively correlated. (Appendix A.4 provides more information about the weights of neurons are not i.i.d..)
>
> >>> There are places in the paper where one must refer to the supplementary, for example, sections H and J with the simulations. Do consider moving these crucial sections to the main paper.
>
> The page limitation of the main paper is the main reason why we put these parts in the Appendix. We will refine our paper to make it better based on your comments. Thank you!
>
> >>> How would you stand by your claim of explaining a DNN's layers to be modeled as a BHM?
>
> That is a very fundamental and important question for deep learning. Thanks for asking!
>
> Theoretically, since the output of a hidden layer is commonly the input of the next layer, we can form a Markov chain to describe the entire architecture of DNNs. Based on the Markov chain, we can derive two conclusions: (1) a hidden layer corresponds to a random variable (the definition of a hidden layer here is general, e.g., a linear convolutional layer and a non-linear layer like ReLU is viewed as a single layer) and (2) the entire architecture of DNNs corresponds a joint distribution.
>
> Given the first conclusion, i.e., a hidden layer corresponds to a random variable, we explain the random variable as a Gibbs distribution. Based on the Gibbs explanation, we experimentally find that the first hidden layer formulates a distribution of the dataset, i.e., a prior distribution. In addition, previous work [1] proves that DNNs estimate a posterior distribution P(Y|X), thus the output layer could be explained as a likelihood distribution. The two points trigger our interests to explain DNNs in a Bayesian way.
>
> Beyond your question, here is a more understanding of deep learning in a Bayesian way.
>
> The entire architecture of DNNs can be explained as joint distribution, the hidden layers close to the input correspond to prior distributions, the output layer defines a likelihood distribution given the prior distributions, e.g., DNN = {x, f_1, f_2, f_Y}, the joint distribution is P(F_Y, F_2, F_1|X) = P(F_Y|F_2) P(F_Y|F_1) P(F_1|X)
>
> The SGD aims to derive a posterior inference from the joint distribution, i.e., P(F_Y|X) = \int  P(F_Y, F_2, F_1|X)dF_1dF_2, where \int denotes integral.
>
> Thanks again for your comments! We have already refined the paper. Look forward to your feedback!
>
> [1] M.D. Richard and R.P. Lippmann. Neural network classifiers estimate Bayesian a posteriori probabilities. Neural Computation, pages 461–483, 1991.

---

> > ### Author Response · Authors · 2019-11-14
> > **Clarifying the probabilistic explanation for the entire architecture of DNNs**
> >
> > Dear reviewer,
> >
> > Thanks again for your professional and encouraging comments. Here we want to plus more explanations for the entire architecture of DNNs.
> >
> > >>> Above all, we think the entire architecture of DNNs cannot be explained as a Bayesian hierarchical model (BHM) after deliberately thoughts and discussion with our coauthors. Notably, BHM forms a hierarchical structure by defining a series of prior distributions for the parameters of the likelihood distribution. However, the proposed Gibbs distribution is actually a distribution of the activations of a hidden layer. Though the activations entirely depend on the parameters of the hidden layer given the input, we cannot say activations are entirely equivalent to the parameters, thus we could not use BHM to explain the entire architecture of DNNs. We apologize for the inappropriate explanation.
> >
> > >>> Second, we relax our explanation, i.e., the entire architecture of DNNs can be explained as a Bayesian network because of two reasons.  (1) The entire architecture of DNNs corresponds to a joint distribution, and (2) we use the Bayesian inference method to infer the true posterior distribution P(Y|X). In particular, we provide rigorous proof (Appendix J) and intense simulations (Appendix K) for the two points.
> >
> > >>> Third, the Bayesian network's explanation for the entire architecture of DNNs does not affect the main part of the paper. We can still derive that GP cannot explain DNNs correctly because the I.I.D assumption for parameters and activations is not satisfied. Bayesian networks still can explain the hierarchical property of DNNs.  Pre-training the hidden layers corresponding to prior distributions still can improve the generalization performance of DNNs since the previous work [1] has already demonstrated that pre-training an effective regularization to capture more features of the training dataset.
> >
> > We have already updated the paper, please check the latest version. If you have more questions, please let us know. Thanks again for your comments making our paper better!
> >
> > [1] Dumitru Erhan, Yoshua Bengio, Aaron Courville, Pierre-Antoine Manzagol, Pascal Vincent, and Samy Bengio. Why does unsupervised pre-training help deep learning? Journal of Machine Learning Research, 11(Feb):625–660, 2010.

---

### Public Comment · ~Xinshao_Wang1 · 2019-09-30
**modified: 27 Sep 2019**

Hi, how did you make it to modify your submission on 27 Sep 2019? Thanks.

---

> ### Author Response · Authors · 2019-09-30
> **The revision was enabled somehow at that time**
>
> We found the revision was enabled somehow then, so we refined the submission a little bit.

---

### Author Response · Authors · 2019-11-10
**Why should we care about the i.i.d. assumption when explaining DNNs as a Gaussian process?**

Let us review the central limit theorem and use a simple example to clarify the Gaussian Process (GP) explanation for deep learning.

(1) The Central Limit Theorem (CLT)
Given a sequence of $\textbf{i.i.d.}$ random variables $\{x_1, \cdots, x_n\}$ with expectation $\mu$ and variance $\sigma^2$, we can derive the average of these random variables, i.e., $S_n = \frac{1}{n}(x_1 + \cdots + x_n)$. CLT proves that the distribution of the random variable $\sqrt{n}(S_n - \mu)$ approaches to a Gaussian distribution as n goes to infinity.
$$\sqrt{n}(S_n - \mu) \rightarrow N(0, \sigma^2) \text{ as } n \rightarrow \infty$$
To some extent, we can say the summation of i.i.d. random variables obeys a Gaussian distribution.
It is important to note that $\{x_1, \cdots, x_n\} \text{ being } \textbf{i.i.d.}$ is the prerequisite of CLT.

Also note that CLT is the statistical foundation of using GP to explain the distribution of hidden layers of DNNs.
Therefore, I.I.D is very important for GP explanation. However, to the best of our knowledge, none of previous works thoroughly study if the activations of DNNs satisfy I.I.D. That is an important reason for we write the paper.

(2) Now let us use a simple neural network to clarify the GP explanation.
***A toy example***
Assuming we have a linear neural network with an input layer {$x_1, x_2$},  a hidden layer {$f_{11}, f_{12}$}, and an output layer {$f_{21}, f_{22}$}. Here we do not consider bias and non-linear activation function for simplicity.

The activations of f_1 can be formulated as
$$f_{11} = a_{11} \cdot x_1 + a_{21} \cdot x_2$$
$$f_{12} = a_{12} \cdot x_1 + a_{22} \cdot x_2$$
where $a_{11}$, $a_{21}$, $a_{12}$, and $a_{22}$ are parameters.

The activations of f_2 can be formulated as
$$f_{21} = b_{11} \cdot f_{11} + b_{21} \cdot f_{12} $$
$$f_{22} = b_{12} \cdot f_{11} + b_{22 } \cdot f_{12} $$
where $b_{11}$, $b_{21}$, $b_{12}$, and $b_{22}$ are parameters.

If we want to derive $P(f_{21})$ is a Gaussian distribution based on CLT, the necessary condition is $f_{11}$ and  $f_{12}$ being i.i.d.. To satisfy the necessary condition,  all the previous works simply assume all the parameters are i.i.d.. Specifically, if $a_{11}$, $a_{21}$, $a_{12}$, and $a_{22}$ are i.i.d., then $f_{11}$ and $f_{12}$ are i.i.d..
As a result, $P(f_{21})$ would approach a Gaussian distribution as the number of neurons in $f_1$ goes to infinity based on CLT.

***Problems****
The i.i.d. assumption could be correct for simple neural networks. However, considering the extremely complex architecture of DNNs, it is necessary to derive rigorous proof and implement intensive experiments to examine if the i.i.d. assumption is valid for DNNs. We think the rigorous logic of using GP to explain DNNs can be summarized as the following three steps. That is also the structure of our paper.

Step1. Check if the parameters of DNNs are i.i.d.. If they are i.i.d., we are safe to directly use GP to explain DNNs, this is the most ideal situation. However, we demonstrate that the parameters are not i.i.d..

Step2. Though the parameters of DNNs are not i.i.d., the activations of hidden layers still could be i.i.d.. If the activations of hidden layers are i.i.d., we are also safe to use GP to explain DNNs, the only thing we need to modify is choosing another prior distribution for the parameters of DNNs.
However, we demonstrate that activations being i.i.d. is not valid for all the hidden layer of DNNs.

Step3. Since neither the parameters of DNNs nor the activations of hidden layers are i.i.d., we cannot use GP to model the distribution of all the hidden layers of DNNs.  We need to propose an alternative explanation.

In addition, previous works [1,2] have already demonstrated that GP is sensitive to the curse of dimensionality and cannot explain the hierarchical representation, an essential of deep learning.


[1] G. Hinton, L. Deng, D. Yu, G. Dahl, A. Mohamed, N. Jaitly, A. Senior, V. Vanhoucke, P. Nguyen, T. Sainath, and B. Kingsbury. Deep neural networks for acoustic modeling in speech recognition. IEEE Signal Processing Magazine, 2012.
[2] Alexander Matthews, Mark Rowland, Jiri Hron, Richard E. Turner, and Zoubin Ghahramani. Gaussian process behavior in wide deep neural networks. In ICLR, 2018.

---

### Author Response · Authors · 2019-11-15
**Updated version of the paper**

We thank all reviewers for their comments. They are extremely insightful and help us to make our paper better. To address their comments, we have been actively refining our paper since the review is released. A new version is uploaded with lots of new results.

Here is the summary of the latest version of the paper.
(1) Refine the discussion part (Section 6)
           Section 6.1 demonstrates the equivalence between SGD and the variational inference.
           Section 6.2 demonstrates that the entire architecture of DNNs can be explained as a Bayesian network.
           Section 6.3 provides more justifications for the proposed regularization algorithm.
(2) Refine Appendix J
            Appendix J.1 proves P(F_Y|X) is also a Gibbs distribution and its energy function is determined by the entire architecture of DNNs.
            Appendix J.2 proves the equivalence between SGD and the variational inference.
            Appendix J.3 proves the equivalence between SGD and the energy minimization optimization.
(3) Refine Appendix K
             Appendix K.1 demonstrates the equivalence between SGD and energy minimization optimization.
             Appendix K.2 demonstrates that the parameters of DNNs are not i.i.d. during the training procedure.
             Appendix K.3 demonstrates that a better prior distribution indicates a better generalization performance.
(4) Refine the background part (Section 3) for providing more information about Gibbs distribution.
(5) Refine Section 4.3 and Appendix H to make the derivation and proof more clear.
(6) Refine Section 5.1 to make the simulation easily to understand.
(7) Accordingly, refine the abstract, introduction, and conclusion part.

---

### Decision · Program_Chairs · 2019-12-19

**Decision:**

Reject

**Comment:**

This paper makes a claim that the iid assumption for NN parameters does not hold. The paper then expresses the joint distribution as a Gibbs distribution and PoE. Finally, there are some results on SGD as VI. Reviewers have mixed opinion about the paper and it is clear that the starting point of the paper (regarding iid assumption) is unclear. I myself read through the paper and discussed this with the reviewer, and it is clear that there are many issues with this paper.

Here are my concerns:
- The parameters of DNN are not iid *after* training. They are not supposed to be. So the empirical results where the correlation matrix is shown does not make the point that the paper is trying to make.
- I agree with R2 that the prior is subjective and can be anything, and it is true that the "trained" NN may not correspond to a GP. This is actually well known which is why it is difficult to match the performance of a trained GP and trained NN.
- The whole contribution about connection to Gibbs distribution and PoE is not insightful. These things are already known, so I don't know why this is a contribution.
- Regarding connection between SGD and VI, they do *not* really prove anything. The derivation is *wrong*. In eq 85 in Appendix J2, the VI problem is written as KL(P||Q), but it should be KL(Q||P). Then this is argued to be the same as Eq. 88 obtained with SGD. This is not correct.

Given these issues and based on reviewers' reaction to the content, I recommend to reject this paper.